# Proteomic compensation by paralogs preserves protein interaction networks after gene loss in cancer

Anjan Venkatesh [1,2,3], Niall Quinn[2,3], Swathi Ramachandra Upadhya[1,3], Barbara De Kegel[2,3], Alfonso Bolado Carrancio[4], Thomas Lefeivre [1,2,3], Olivier Dennler [1,3], Kieran Wynne[1,2,5], Alexander von Kriegsheim[4] & Colm J Ryan [1,2,3,5]✉

## Abstract

**Proteins operate within dense interconnected networks, with interactions necessary both for stabilising proteins and enabling them to execute their molecular functions. Remarkably, protein–protein interaction networks operating within tumour cells continue to function despite widespread genetic perturbations. Previous work has demonstrated that tumour cells tolerate perturbations of paralogs better than perturbations of singleton genes, but the underlying mechanisms remain poorly understood. Here, we systematically profile the proteomic response of tumours and cell lines to gene loss. We find many examples of proteomic compensation, where loss of one gene causes increased abundance of a paralog, and collateral loss, where gene loss causes reduced paralog abundance. Compensation is enriched among paralog pairs that are central in the protein–protein interaction network and whose interaction partners perform essential functions. Compensation is also significantly more likely to be observed between synthetic lethal pairs. Our results support a model whereby loss of one gene results in increased protein abundance of its paralog, stabilising the protein–protein interaction network. Consequently, tumour cells may become dependent on the paralog for survival, creating potentially targetable vulnerabilities.**

**Keywords** Cancer; Gene Loss; Paralog Compensation; Synthetic Lethality; Protein-protein Interactions
**Subject Categories** Cancer; Proteomics

## Introduction

Tumour cells tolerate enormous amounts of genetic perturbations, including loss of function mutations and deletions of protein-coding genes (Zack et al, 2013; Martincorena et al, 2017; Zapata et al, 2018; De Kegel and Ryan, 2023). We, and others, have demonstrated that duplicate genes (paralogs) contribute

significantly to this genetic robustness—tumour cells are more tolerant to loss of paralogs than singleton genes, resulting in deletions of paralogs being more frequently observed in tumour genomes (De Kegel and Ryan, 2019, 2023). Perhaps the simplest explanation for the increased dispensability of paralogs is that pairs of paralogs can compensate for each other's loss because of their shared functions. Direct evidence for this model comes from double perturbation screens in cancer cell lines—often either paralog in a pair can be lost individually with relatively little fitness consequence, but their combined inhibition causes cell death, a phenomenon termed synthetic lethality (Dede et al, 2020; Gonatopoulos-Pournatzis et al, 2020; Parrish et al, 2021; Ito et al, 2021; Thompson et al, 2021). The increased dispensability of paralog genes, combined with evidence of frequent synthetic lethality between paralog pairs, supports a model whereby tumours can tolerate the loss of an individual paralog because its counterpart can compensate for its loss. This provides a clear model for the tolerance of tumour cells for genetic perturbation but provides no insight into the mechanisms by which this compensation takes place. In particular, it is entirely unclear how the molecular interaction networks operating within tumour cells adapt to the loss of paralog genes. Is there a need for increased production of the compensating paralog in order for it to carry out the functions of the lost paralog? How do protein complexes, which often rely on balance between different subunits, adapt to the loss of paralogous subunits?

To our knowledge, no systematic effort to understand the mechanisms behind paralog compensation has been undertaken in the context of cancer or human cells. However, there have been a number of systematic efforts to do so in the budding yeast *Saccharomyces cerevisiae*. Early work identified that some pairs of paralogs exhibit *transcriptional reprogramming* such that loss of one paralog was associated with increased transcription of another (Kafri et al, 2005). A subsequent systematic study in yeast assessed the proteomic response of paralogs to deletion of their counterpart and found that, of 202 paralogous gene pairs tested, just over 10% exhibit '*need based upregulation*' such that deletion of one paralog caused increased protein abundance of the other (DeLuna et al, 2010). This 'need-based upregulation' means that the compensatory response only occurs in environmental conditions where the

[1]Conway Institute of Biomolecular and Biomedical Research, University College Dublin, Dublin, Ireland. [2]Systems Biology Ireland, University College Dublin, Dublin, Ireland. [3]School of Computer Science, University College Dublin, Dublin, Ireland. [4]Edinburgh Cancer Research UK Centre, University of Edinburgh, Edinburgh, UK. [5]School of Medicine, University College Dublin, Dublin, Ireland. ✉E-mail: colm.ryan@ucd.ie

function of the gene is required for optimal growth or survival, and disappears in conditions where the function becomes dispensable. Intriguingly, the pairs that exhibited this *upregulation* were enriched for pairs known to be synthetic lethal, suggesting a relationship between phenotypic compensation (i.e. cases where the presence of a paralog enables cells to survive the deletion of an essential gene) and molecular compensation (i.e. cases where the paralogous protein is upregulated in the context of essential gene loss). The authors speculated that these relationships may be due to stoichiometric requirements of protein complexes containing both paralogs. A subsequent systematic effort to understand how loss of one paralog alters the protein–protein interactions of its counterpart revealed examples of both compensation and the opposite effect, with the two relationships occurring at similar frequencies (Diss et al, 2017). In the case of compensation, loss of one paralog resulted in increased interaction between the compensatory paralog and the protein–protein interaction partners of the lost paralog. In the case of the opposite effect, deletion of one paralog resulted in the other paralog losing protein–protein interaction partners. Further analysis suggested that such pairs are enriched among paralogs that form heteromers and that these pairs may stabilise each other through this interaction (Dandage and Landry, 2019).

We note that there has been a diversity of terms used in the literature to refer to cases where the abundance of one paralog is increased in response to the deletion of another—need-based upregulation, transcriptional reprogramming, active compensation —and here we simply use the term 'proteomic compensation' (Kafri et al, 2005; DeLuna et al, 2010; Diss et al, 2014). The opposite effect, where the abundance of one paralog is decreased in response to deletion of another has also been observed (Diss et al, 2017; Dandage and Landry, 2019), and reported using distinct terms, including *negative responsiveness*, and *dependency*. We here use the term *collateral loss* to indicate that a reduction in the protein abundance of one paralog happens in response to the genetic perturbation of another (i.e. the protein is not itself the target of the genetic event, but nonetheless it displays decreased abundance).

While no systematic study of the molecular consequences of paralog loss has been performed in cancer, there have been systematic efforts to understand how tumour cells respond to copy number variation in general (Gonçalves et al, 2017; Sousa et al, 2019; Cheng et al, 2022). These studies have made use of matched genomic, transcriptomic, and proteomic profiles of tumours to understand how genetic variation influences transcriptomic and proteomic variation. A particularly striking finding has been that copy number changes at the genomic level are often attenuated at the protein level and that this is especially evident for genes that encode protein complex subunits, suggesting that regulatory mechanisms operate within tumour cells to maintain protein complex stoichiometry (Stingele et al, 2012; Gonçalves et al, 2017; Sousa et al, 2019; Schukken and Sheltzer, 2022). An additional finding has been that many protein–protein interaction partners control each other's abundance, such that loss of one gene was associated with reduction of another (Gonçalves et al, 2017). There is some evidence that the regulation of protein complex stoichiometry and the controlling relationships between paralog pairs occurs via proteasome-mediated degradation of excess subunits (Gonçalves et al, 2017; Sousa et al, 2019; Schukken and Sheltzer, 2022).

Here we sought to directly address the question of how tumour cells respond to loss of paralog genes. First, we investigated the proteomic consequences of knocking out a set of 34 paralogous genes using CRISPR-Cas9 gene editing in a single genetic background. Consistent with results from yeast, we identified examples of both collateral loss and protein compensation (DeLuna et al, 2010; Diss et al, 2017). We then performed a larger scale analysis of tumour samples with matched genomes, transcriptomics, and proteomes. This allowed us to assess the regulatory relationships between thousands of paralog pairs and revealed hundreds of examples of collateral loss and protein compensation. The set of compensation pairs identified was enriched in paralogs that were central in the protein–protein interaction network, belonged to small paralog families, and were members of essential protein complexes. In contrast, paralogs that exhibit collateral loss relationships were more peripheral on the protein–protein interaction network and less likely to be involved in essential functions. Compensation pairs were also highly enriched among known synthetic lethal pairs. Overall, our results suggest a model where gene loss-induced disruptions to the protein–protein interaction network are buffered by increased protein abundance of paralogous proteins, upon which tumour cells then become dependent for survival. These proteomic compensation events are widespread in cancer, enabling cancer cells to thrive despite frequent gene loss.

## Results

### Mass spectrometry profiling of isogenic knockouts reveals compensation and collateral loss events

We first modelled the proteomic consequences of paralog loss using isogenic cell lines, allowing us to causally link an introduced mutation to changes in protein abundance. Previous systematic efforts to identify associations between the loss of one protein and the abundance of a paralog had been carried out in budding yeast and were performed by mating GFP-tagged strains (where fluorescence was used as a proxy for protein abundance) with gene deletion strains (DeLuna et al, 2010). Since this approach does not readily translate to mammalian cells, we used an alternative approach—CRISPR perturbation of individual paralogs followed by quantification of the proteome by mass spectrometry. We used the HAP1 model for this work—HAP1 is a near-haploid cell line, allowing for reliable generation of homozygous mutations. We selected 34 paralogous genes to mutate, based on a number of criteria—firstly the genes had to be detected in the HAP1 transcriptome but be non-essential for growth (Blomen et al, 2015). Within this subset, we prioritised genes that had a previously reported or suspected synthetic lethal interaction with at least one of their paralogs, on the grounds that these pairs exhibit phenotypic compensation and therefore might be more likely to display proteomic compensation (DeLuna et al, 2010; De Kegel et al, 2021; Gonatopoulos-Pournatzis et al, 2020; Parrish et al, 2021; Thompson et al, 2021; Ito et al, 2021; Dede et al, 2020). We obtained knockout HAP1 cell lines from Horizon discovery and performed whole-cell lysate proteomic profiling of 34 HAP1-derived clones, each of which had a frameshift insertion or deletion of a single paralog, and the parental HAP1 strain for comparison (Fig. 1A, Methods). Each clone was grown in four separate cultures, constituting four

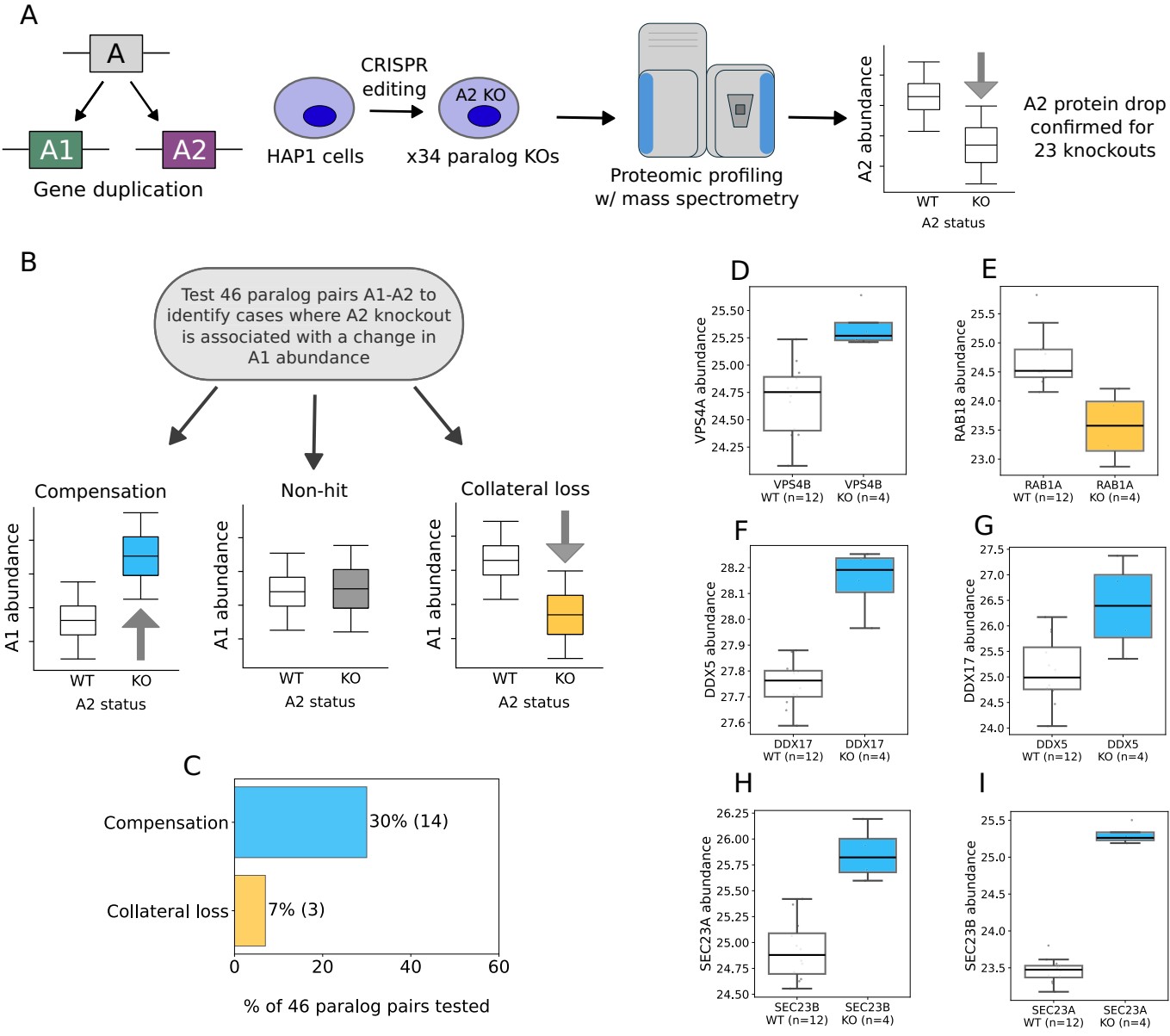

**Figure 1. Identifying paralog protein compensation and collateral loss in an isogenic model.**

(A) Experimental setup for proteomic profiling of HAP1 knockouts. (B) Workflow for identifying cases of paralog compensation and collateral loss. (C) Proportion of compensation and collateral loss pairs identified, with actual count in brackets. (D–I) Box plots for compensation and collateral loss hits showing increase (blue) or decrease (yellow) in the abundance of a protein following the knockout of its paralog. Sample sizes for each group are shown in parentheses. In all boxplots, the central line represents the median, box limits indicate the 25th and 75th percentiles (first and third quartiles), and whiskers extend to data points that are no more than 1.5 × interquartile range from either end of the box. Each grey dot represents a cell line. Four biological replicates were profiled for each knockout (Methods).

biological replicates. On average, we quantified 4544 proteins per clone profiled (Dataset EV1). We first verified that the mutation in each of the 34 profiled clones caused a reduction in the abundance of the associated protein. For 23 (~68%) of the models, we were able to verify a significant drop in the abundance of the protein product of the targeted gene (Dataset EV2). For the remainder we could either not reliably quantify the protein in the parental HAP1 model (8/34 models), so a reduction in the protein abundance could not be assessed, or we could quantify the protein but not verify an associated drop in abundance (3/34 models).

For the 23 genes whose loss-of-function mutation was proteomically verified, we then tested whether the mutation caused a change in the abundance of each of their detected paralogs (Methods). In total we tested 46 gene pairs, since many of the targeted genes had multiple identifiable paralogs (Fig. 1B). Using this approach, we identified 14 gene pairs with compensation relationships and 3 gene pairs with collateral loss relationships (Fig. 1C, Datasets EV3, EV7). Here, compensation refers to a pair A1-*A2* where a protein A1 has significantly (FDR < 5%; $p < 0.05$) increased abundance when its paralog *A2* is knocked out, and

collateral loss refers to a pair where A1 has significantly reduced abundance when *A2* is knocked out. For example, we identified a compensatory relationship between *VPS4B* and VPS4A (which encode ATPases involved in ESCRT-dependent membrane remodelling), a pair we and others have previously identified as synthetic lethal (McDonald et al, 2017; Lord et al, 2020; Neggers et al, 2020) (Fig. 1D). In contrast we found that loss of the Rab GTPase *RAB1A* was associated with collateral loss of its paralog Rab GTPase RAB18 (Fig. 1E). In 3 cases where we had knockouts of both genes in a pair, we observed reciprocal compensation effects. For example, in the case of the DEAD-box helicases *DDX5* and *DDX17*, which we have previously identified as synthetic lethal (Lord et al, 2020), *DDX17* loss resulted in increased protein abundance of DDX5 (Fig. 1F), while *DDX5* loss resulted in increased protein abundance of DDX17 (Fig. 1G). Similar reciprocal patterns were observed for SEC23A-SEC23B (Fig. 1H,I) and SMARCC1-SMARCC2 (Dataset EV3).

## Unbiased identification of paralog compensation and collateral loss in a panel of tumour proteomes

Our analysis of HAP1 paralog knockouts demonstrates that both proteomic compensation and collateral loss relationships can be detected by mass spectrometry profiling in isogenic models. However, there are some limitations to this approach—most notably, each gene of interest requires the creation of a new isogenic knockout. Due to this lack of scalability, we chose paralog pairs for which we anticipated seeing compensatory effects and our results do not represent an unbiased analysis. To perform a more systematic and unbiased analysis, we next analysed the consequences of paralog loss in a large compendium of tumour proteomes from the Clinical Proteomic Tumor Analysis Consortium (CPTAC). We analysed a set of 930 normalised tumour proteomes with matched transcriptomes and copy number profiles from this project (Methods). Since these samples come from nine different studies, each pertaining to a different cancer type, we incorporated cancer type as a covariate when assessing the association between gene loss and protein abundance. As recurrent homozygous gene loss is relatively uncommon, even in tumours, we focussed our analysis on single copy loss (Cheng et al, 2017; De Kegel and Ryan, 2023). For all paralogs that were subject to single copy loss in at least 20 samples, we assessed whether this loss was associated with a reduction in the abundance of the cognate protein. For 2502 out of 3439 (~73%) of tested genes, we found this to be the case (Fig. EV1A, Dataset EV4). We next asked, for each of these genes, how the copy number loss was associated with the abundance of each of their quantified paralogs (Fig. 2A). In total, we tested 5128 paralog associations (Fig. EV2, Datasets EV5, EV8). At an FDR of 5%, we identified 603 (12%) compensation and 532 (10%) collateral loss hits (Fig. 2B,C). For example, loss of the DNA topoisomerase *TOP2B* was associated with a significant increase of TOP2A protein (Figs. 2D and EV1B) (Champoux, 2001). In contrast, loss of the GDP-mannose pyrophosphorylase *GMPPB* was associated with a significant decrease in the abundance of its paralog GMPPA (Figs. 2G and EV1E). GMPPA and GMPPB form a heterodimer to enable GDP-Man synthesis and it is therefore possible that loss of *GMPPB* results in an overall reduction in the abundance of this complex leading to reduced stability of GMPPA (Franzka et al, 2021; Zheng et al, 2021).

In our analysis of the HAP1 data, we found that some pairs were observed as compensatory in a symmetric fashion—*A2* loss was associated with an increased abundance of A1, while loss of *A1* was associated with increased abundance of A2. With the CPTAC analysis, most paralog pairs were tested in an asymmetric fashion— i.e. we assessed the impact of *A2* loss on A1's protein abundance but not *A1*'s loss on A2's protein abundance. This was due to the filtering steps we applied, which excluded some pairs from being tested in both directions. However, we found that, for the subset of pairs that were tested in both directions, there was a significant enrichment for symmetric compensation pairs (OR = 2.7, $p = 2 \times 10^{-6}$; Appendix Fig. S1A) and for symmetric collateral loss pairs (OR = 1.7, $p = 0.02$; Appendix Fig. S1B). In other words, paralogs that were a hit in one direction were significantly more likely to be a hit in the other direction. For example, loss of *BRD4* was associated with increased abundance of BRD3 and vice-versa, suggesting symmetric compensation (Figs. 2E,F and EV1C,D). In contrast, loss of *DOK3* was associated with reduced protein abundance of DOK2 and vice-versa, suggesting symmetric collateral loss (Figs. 2H,I and EV1F,G). Interestingly, compensation pairs were significantly less likely to have at least one paralog in the family belong to a collateral loss pair (OR = 0.3, $p = 2.3 \times 10^{-29}$), suggesting that these effects are mutually exclusive within families.

Four compensation pairs identified with the unbiased CPTAC analysis were also compensation hits in the HAP1 analysis— SMARCC1-*SMARCC2* (i.e. SMARCC1 abundance increases when *SMARCC2* is lost), SEC23B-*SEC23A*, CHMP1B-*CHMP1A*, and DDX5-*DDX17* (Fig. EV3, Dataset EV5).

## Gene essentiality, sequence features, and protein–protein interaction features are associated with paralog compensation

We wished to understand what features might be associated with proteomic compensation and collateral loss in the CPTAC dataset. Paralog relationships are typically identified from sequence comparisons and we therefore initially focussed on sequence-derived features. The sequence identity between paralogs is often used as a proxy for functional similarity, with highly sequence-similar proteins presumed to be more functionally similar. We hypothesised that compensatory paralogs might be more functionally similar and therefore display higher sequence identity, as has previously been shown in yeast (DeLuna et al, 2010). We found that this is indeed the case—compensation hits had higher amino acid sequence identity than non-hits (two-tailed t-test, Cohen's d = 0.2, $p = 6.9 \times 10^{-4}$) (Fig. 3A).

Due to multiple rounds of duplication, often a single gene can have multiple paralogs. We anticipated that compensation might be more common among 'closest pairs'—i.e. pairs which are each other's most sequence-similar paralog within a larger gene family. We found that this is indeed the case for compensation pairs (Fisher's Exact Test, OR = 1.4; $p = 3.8 \times 10^{-4}$) while collateral loss pairs are less likely to be closest pairs (OR = 0.7, $p = 0.01$) (Fig. 3B). Compensation pairs also have significantly lower family sizes (Cohen's d = −0.4, $p = 3 \times 10^{-17}$) (Fig. 3C).

We hypothesised that proteomic compensation might provide a mechanism to stabilise the protein–protein interaction network against variation in protein abundance (either due to stochastic variation or external perturbations). If this is the case, one might

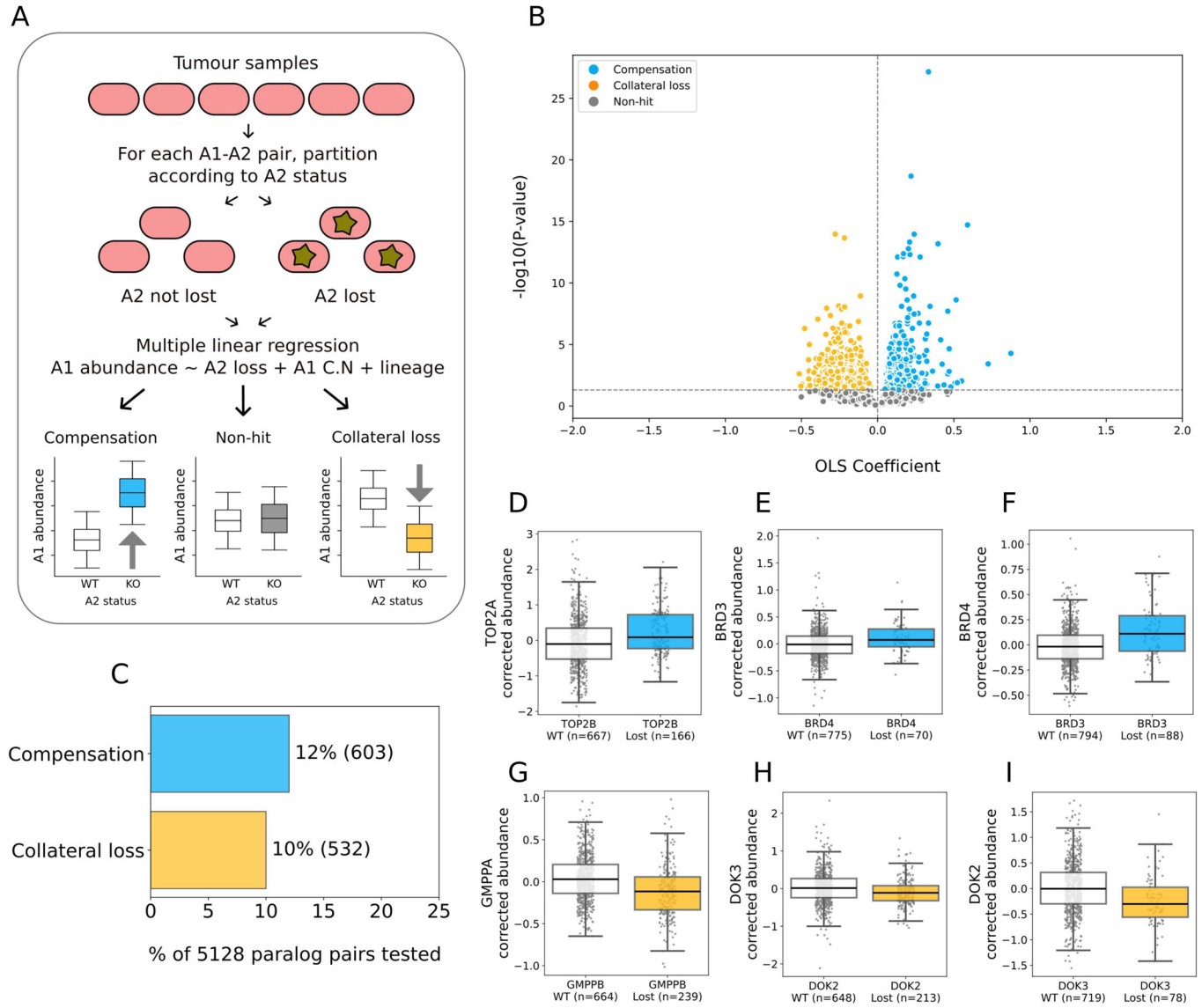

**Figure 2. Identifying paralog protein compensation and collateral loss using tumour profiles.**

(A) Workflow diagram for unbiased identification of collateral loss and compensation from tumour proteomic profiles. (B) Volcano plot showing −log10(FDR) vs ordinary least squares (OLS) coefficient for 5128 paralog tests run with the CPTAC dataset. Each dot represents a paralog pair. Each test involves an OLS linear regression fit to explain protein abundance using hemizygous paralog loss, with lineage and self-copy number as covariates. FDRs are derived by applying Benjamini–Hochberg multiple testing correction on the t-statistics of the paralog loss variable. Coefficients plotted are also for the paralog loss variable in a given test. (C) Proportion of compensation and collateral loss hits in the CPTAC analysis, with actual counts in brackets. (D–I) Box plots of compensation (TOP2A-TOP2B, BRD3-BRD4, BRD4-BRD3) and collateral loss (GMPPA-GMPPB, DOK3-DOK2, DOK2-DOK3) hits. In all boxplots, the central line represents the median, box limits indicate the 25th and 75th percentiles (first and third quartiles), and whiskers extend to 1.5 × interquartile range from either end of the box. Each grey dot represents a tumour sample. Position on the y-axis indicates protein abundance adjusted for copy number of the encoding gene and cancer type, which are used as covariates in the regression model. Sample sizes for each group are shown in parentheses.

anticipate that proteins with more interactions should be more likely to display compensatory relationships. We therefore calculated the degree centrality (total interactions normalised by network size) of all paralogs using protein–protein interactions from the BioGRID database (Oughtred et al, 2021). We found that the genes lost in compensation pairs (i.e. compensated-for genes) tend to have significantly higher protein–protein interaction degree centrality (Cohen's d = 0.3, $p = 6.2 \times 10^{-14}$) (Fig. 3D). In contrast, genes associated with collateral loss relationships had significantly

lower degree centrality (Cohen's d = −0.2, $p = 3.4 \times 10^{-4}$) (Fig. 3D). Similar observations were evident when using the STRING physical interaction network as the source of protein–protein interaction networks (Szklarczyk et al, 2023) (Cohen's d = 0.3, $p = 3.6 \times 10^{-11}$ for compensation and Cohen's d = −0.2, $p = 1.1 \times 10^{-3}$ for collateral loss) (Appendix Fig. S2A). The association with degree centrality could potentially be confounded by protein abundance—more abundant proteins might be more amenable to detection, resulting in such proteins having a larger number of known interactors.

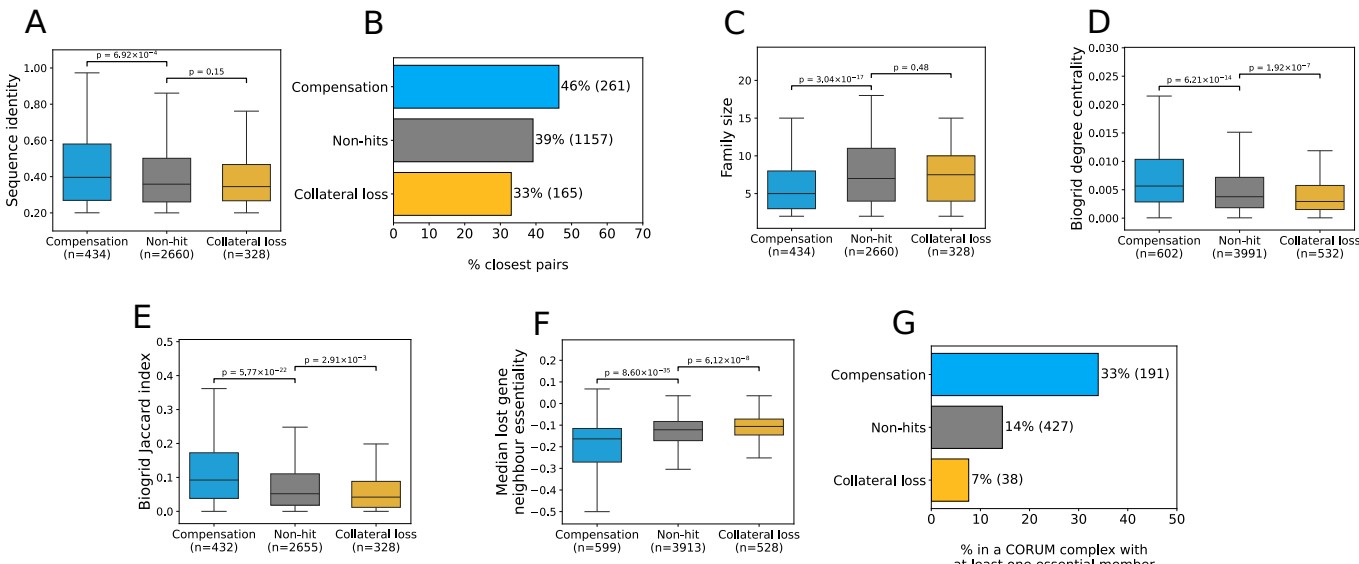

**Figure 3. Compensation pairs are more sequence similar, belong to smaller families, and are enriched among closest pairs.**

(A) Box plots showing the amino acid sequence identity of compensation pairs, collateral loss pairs, and non-hits. (B) Bar chart showing the percentage of paralog pairs in each group that are each other's 'closest paralog' among all members of a paralog family. (C) Box plots showing the family size (number of paralogs belonging to the same family) for pairs with compensation, collateral loss, and non-hits. (D) Box plots showing the degree centrality values of compensation pairs, collateral loss pairs, and non-hits, calculated using the BioGRID interaction repository (Oughtred et al, 2021). (E) Box plots showing the Jaccard indices of paralog pairs in all three groups, calculated using BioGRID. (F) Box plots showing the median essentiality of the neighbours of the lost gene, for paralog pairs in all three groups, calculated using BioGRID. (G) Bar chart showing the proportion of paralog pairs in each group where at least one paralog is a member of a CORUM protein complex with at least one broadly essential member (Methods). Counts and sample sizes for each group are shown in parentheses. All p-values are from two-t t-tests (assuming equal variances). In all boxplots, the central line represents the median, box limits indicate the 25th and 75th percentiles (first and third quartiles), whiskers extend to 1.5 × interquartile range from either end of the box, and outliers are not displayed.

However, this was not the case– when comparing mean abundances of lost paralogs in the GTEx dataset, deleted genes in compensation pairs had significantly lower mean abundance than non-hits (Cohen's d = −0.1, p = 0.03) (Appendix Fig. S2B). Conversely, both the deleted gene and the paralog with reduced abundance in collateral loss pairs had significantly higher mean abundance (Cohen's d = 0.1, p = 2.5 × 10$^{-3}$ for lost paralogs and Cohen's d = 0.3, p = 9 × 10$^{-9}$ for the other paralog) (Appendix Fig. S2C).

We reasoned that proteins can only compensate for each other's loss if they can interact with the same partners, and therefore tested whether compensatory gene pairs shared a higher proportion of their protein–protein interaction partners. The Jaccard index measures the proportion of shared interactors between two proteins when compared to the total number of interactors associated with either protein. Analysing the BioGRID database, we found that compensation hits had significantly higher Jaccard indices than non-hits (Cohen's d = 0.5, p = 5.8 × 10$^{-22}$), while collateral loss hits had significantly lower Jaccard indices (Cohen's d = −0.2, p = 3 × 10$^{-3}$) (Fig. 3E). As before, this trend is also significant for compensation pairs when using the STRING network (Cohen's d = 0.3, p = 2.6 × 10$^{-9}$ for compensation and Cohen's d = −0.005, p = 0.9 for collateral loss) (Appendix Fig. S2D).

Consistent with a central role in the protein–protein interaction network, we found that compensation gene pairs were significantly more likely to be annotated as belonging to protein complexes in the manually curated CORUM database (Tsitsiridis et al, 2023) (Appendix Fig. S2E). This holds both when considering either member of the pair being in a protein complex (OR = 1.8,

p = 9.4 × 10$^{-9}$) or both members being in the same protein complex (OR = 3.3, p = 2.4 × 10$^{-9}$). A significant enrichment for compensation hits was also observed using an alternative source of curated protein complexes from the EBI ComplexPortal (OR = 2.1, p = 1.6 × 10$^{-8}$; Appendix Fig. S2F) and a set of predicted protein complexes from the HuMap project (OR = 2, p = 3.1 × 10$^{-10}$; Appendix Fig. S2G) (Meldal et al, 2019; Drew et al, 2021).

Compensatory relationships between paralogs may exist as a means of maintaining essential cellular functions in the face of perturbations. If that is the case, then one would expect that compensation should be enriched among genes that are involved in cellular processes whose perturbation causes a cellular growth defect. To assess this, we calculated the average growth defect (inferred from CRISPR screens in 762 cell lines, reprocessed as outlined in previous work (De Kegel et al, 2021)) associated with the perturbation of each protein coding gene. We then asked if the protein–protein interaction partners of genes whose loss triggered compensation generally caused a greater growth defect, and found that this is indeed the case (Cohen's d = 0.5, p = 8.6 × 10$^{-35}$) (Fig. 3F). On the other hand, the interaction partners of genes whose loss triggered collateral loss of a paralog were significantly less essential than those of non-hits (Cohen's d = −0.2, p = 9 × 10$^{-5}$) (Fig. 3F). Consistent with this, we find that compensation is enriched among subunits of protein complexes with at least one broadly essential subunit (CERES score < −0.6 in at least 80% of cell lines) (OR = 3.4, p = 1.3 × 10$^{-25}$) (Fig. 3G) (Behan et al, 2019; Meyers et al, 2017). In contrast, collateral loss is depleted among such complexes (OR = 0.6, p = 6 × 10$^{-3}$) (Fig. 3G).

Genes that are broadly conserved across species tend to be involved in more essential processes. If compensation exists as a means of maintaining essential functions, then one might anticipate that compensatory gene pairs should also be more widely conserved across evolution. To test this, we assigned all paralogs a conservation score according to whether or not they had identifiable orthologs in 1472 species (Methods). Using this score, we found that genes involved in compensatory pairs were more widely conserved than non-hits (Cohen's d = 0.3, $p = 4.1 \times 10^{-8}$), while collateral loss gene pairs were less conserved than non-hits (Cohen's d = −0.2, $p = 2.7 \times 10^{-4}$) (Appendix Fig. S2H).

Since protein abundances and interactions are influenced by protein stability, we compared experimentally determined protein half-life across these groups (Zecha et al, 2019). We found no significant differences (Fig. EV4A,B; Dataset EV10). We then obtained protein length annotations from UniProt, comparing these across the three groups. Both the compensating paralog and the lost paralog in compensation pairs tend to be longer than their counterparts in non-hit pairs (Cohen's d = 0.2, $p = 2.4 \times 10^{-4}$ for lost paralog, Cohen's d = 0.2, $p = 1.7 \times 10^{-4}$ for the other paralog) (Fig. EV4C,D) (UniProt Consortium, 2025). Interestingly, this is also true of collateral loss pairs (Cohen's d = 1.5, $p = 8.4 \times 10^{-4}$ for lost paralog, Cohen's d = 1.5, $p = 1.5 \times 10^{-3}$ for the other paralog). However, two proteins of similar lengths might have a different proportion of their residues in an interaction interface. We assessed this directly using a dataset of 486,099 AlphaFold-predicted dimers, calculating, for each paralog with data available, the proportion of its residues ever present in an interaction interface according to a high-confidence model (Methods) (Jänes et al, 2024). We found that both the lost protein and its paralog in collateral loss pairs tend to have a lower proportion of their residues in interaction interfaces, in line with collateral loss paralogs generally having lower degree (Cohen's d = −0.18, $p = 7.7 \times 10^{-4}$ for the hemizygously lost paralog, Cohen's d = −0.1, $p = 0.04$ for the collaterally lost paralog) (Fig. EV4E,F). Work in yeast has suggested that compensation is more likely to be observed for co-regulated paralog pairs (DeLuna et al, 2010). As such, we used the Dorothea database of transcription factor-gene relationships to assess co-regulation by the same transcription factor (Garcia-Alonso et al, 2019). We found that collateral loss pairs are significantly more likely to be co-regulated by the same transcription factor (OR = 2, $p = 7.8 \times 10^{-3}$). However, only 6% of collateral loss paralogs were found to be co-regulated (Fig. EV4G) and there was no enrichment for compensation pairs. Finally, we assessed if physical interaction between paralogs played a role in mediating these effects by filtering BioGRID to direct interactions (Methods). We found that compensation pairs were significantly more likely to share a direct physical interaction with each other (OR = 1.6, $p = 1 \times 10^{-3}$) (Fig. EV4H).

Interestingly, symmetric compensation pairs, i.e. pairs exhibiting a compensatory relationship in both directions, were significantly more likely to be in CORUM complexes, both generally and when restricting to essential complexes, compared to asymmetric pairs (OR = 2.3, $p = 1.6 \times 10^{-2}$) (Appendix Fig. S3A,B). The lost paralogs in symmetric compensation pairs were also more essential (Cohen's d = 0.4, $p = 7.3 \times 10^{-4}$) and had significantly higher degree centrality values (Cohen's d = 0.5, $p = 4.6 \times 10^{-3}$) (Appendix Fig. S3C,D). Symmetric compensation pairs were also more likely to have a direct physical interaction than asymmetric

pairs (OR = 2.8, $p = 0.02$) (Appendix Fig. S3E). These symmetric cases might correspond to paralog pairs that have a stronger functional overlap.

Overall, these results suggest that compensation pairs are more central in the protein–protein interaction network, more conserved across species, and more likely to be involved in essential functions. In contrast, collateral loss genes are more peripheral on the protein–protein interaction network, less conserved, and less likely to work in essential complexes.

## Protein interaction features and family size are sufficient to explain paralog compensation without additional features

To understand the relative contributions of different features, we calculated Receiver Operating Characteristic (ROC) Area Under the Curve (AUC) values for all features (Fig. 4A). We found that neighbour essentiality had the highest predictive power (AUC 0.66), followed by BioGRID Jaccard index (AUC 0.62) and either member of the pair being in an essential protein complex (AUC 0.61) (Fig. 4A).

Many of the features we associated with proteomic compensation might be correlated with each other—for example, one might expect paralogs with greater sequence identity to share a larger subset of their interactors. We assessed the Pearson correlation between all features, excluding features calculated for orthogonal validation, e.g. BioGRID Jaccard index was retained and STRING Jaccard index was dropped. Most pairs of features were poorly correlated, with some exceptions such as sequence identity and family size (r = 0.4) (Appendix Fig. S4).

We wished to assess the unique contributions of these features to paralog compensation by incorporating them into a single statistical model. Since many of our features are weakly correlated, we decided to use a LASSO regression model. LASSO models are robust to moderately correlated features and penalize the inclusion of excess features in the model. Coefficients in LASSO models can also be reduced to zero, enabling feature selection rather than just weighting. This approach enables us to identify key features that contribute uniquely to paralog compensation (Methods). Features with high proportions of missing values (half-life, proportion of residues at interface) were excluded from the regression.

After fitting this model, we were left with 4 features—family size (inverted), neighbour essentiality, BioGRID Jaccard index, and BioGRID degree centrality—which when combined, provided a comparable AUC to that of a model fit using all features (Fig. 4B,C). We also fit a similar model to explain collateral loss and found that two variables, degree centrality and essentiality of the lost paralog, were sufficient to achieve an AUC similar to that of a model with all variables included (Appendix Fig. S5). Interestingly, collateral loss AUCs for these features were generally anti-correlated with compensation AUCs, with a Pearson correlation coefficient of −0.83. This is in line with compensation and collateral loss being mutually exclusive phenomena within paralog families.

## Compensation pairs are more likely to be synthetic lethal

If paralog protein compensation exists as a means of maintaining essential cellular functions in the face of perturbation, then one would anticipate that loss of one member of a compensating

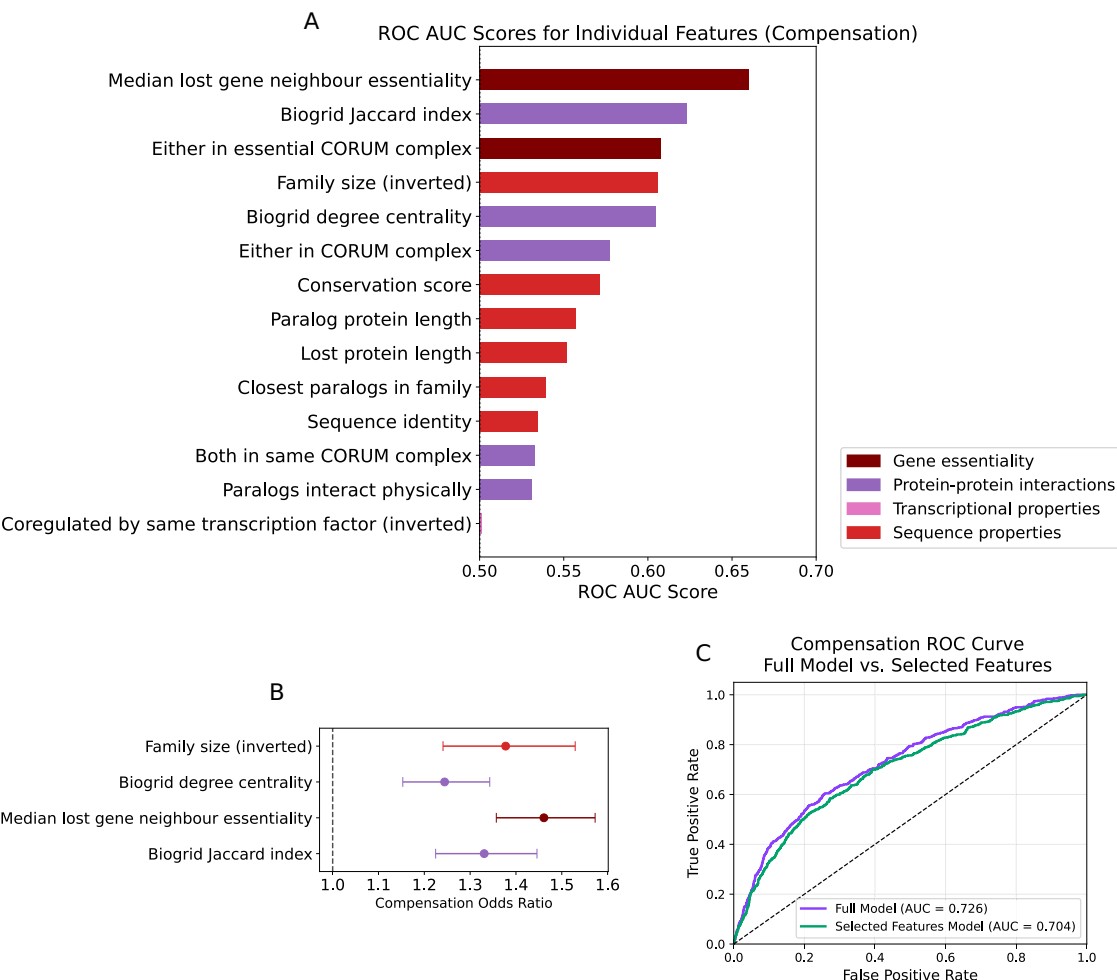

Figure 4. **Neighbour essentiality, Jaccard index, degree centrality, and family size are sufficient to predict paralog compensation.**

(A) Receiver Operating Characteristic (ROC) Area Under the Curve (AUC) values for individual predictors of proteomic compensation. Features with negative predictive value are inverted, i.e. multiplied by −1 ($n = 4170$). (B) Odds ratios and error bars for variables which were retained in a LASSO regression fit to predict proteomic compensation using the features in (A) ($n = 4170$, Methods). Error bars represent 95% confidence intervals derived from the standard errors of the logistic regression coefficient estimates. (C) ROC curve comparing the predictive power of the full model (logistic regression model containing all features) with the predictive power of the "selected features model" containing only the four features shown in (B) ($n = 4170$).

paralog pair would render the other paralog in the pair essential, i.e. the two genes should be synthetic lethal. This would be consistent with previous observations from budding yeast where 22 paralog pairs that exhibited compensatory relationships were found to be enriched in known synthetic lethal pairs (DeLuna et al, 2010). To test this hypothesis on our set of proteomic compensation pairs we compared them to two distinct sets of synthetic lethal pairs—one set derived from genome-wide CRISPR screens in a panel of cancer cell lines from DepMap (Meyers et al, 2017; Behan et al, 2019), where loss of function of one gene could be associated with increased sensitivity to loss of its paralog (De Kegel et al, 2021), and one set derived from multiplexed combinatorial CRISPR screens, where paralog pairs are knocked out in combination to identify cases where the observed fitness defect of perturbing both paralogs is significantly greater than the fitness defect expected from perturbing each paralog individually (Parrish et al, 2021; Thompson et al, 2021; Dede et al, 2020; Gonatopoulos-Pournatzis et al, 2020; Ito et al, 2021; De Kegel et al, 2021) (Methods). Using either

set of synthetic lethal pairs, compensation pairs were significantly enriched for synthetic lethality (DepMap: OR = 2.5; $p = 2 \times 10^{-4}$. Combinatorial screens: OR = 2.4; $p = 0.01$) (Fig. 5A,B). Synthetic lethal hits can vary across different cell lines and screens, but we found that proteomic compensation hits were enriched in among synthetic lethal pairs whether we required them to be observed in at least one screen (OR = 3, $p = 3.3 \times 10^{-9}$; Appendix Fig. S6) or multiple screens (Fig. 5B).

Synthetic lethal gene pairs that exhibit proteomic compensation include well studied pairs, such as *SMARCA2-SMARCA4* (Ehrenhöfer-Wölfer et al, 2019), as well as gene pairs that have been identified as synthetic lethal in multiple combinatorial screens, such as *CNOT7-CNOT8*, which was recently identified as the most reproducible synthetic lethality in a systematic analysis of paralog screens (Esmaeili Anvar et al, 2024). Among the less well characterised pairs are the paralogous heat-shock associated proteins HSPA4-HSPH1 (Tavaria et al, 1996; Dragovic et al, 2006). HSPA4-HSPH1 is a reciprocal compensation hit, i.e. *HSPH1*

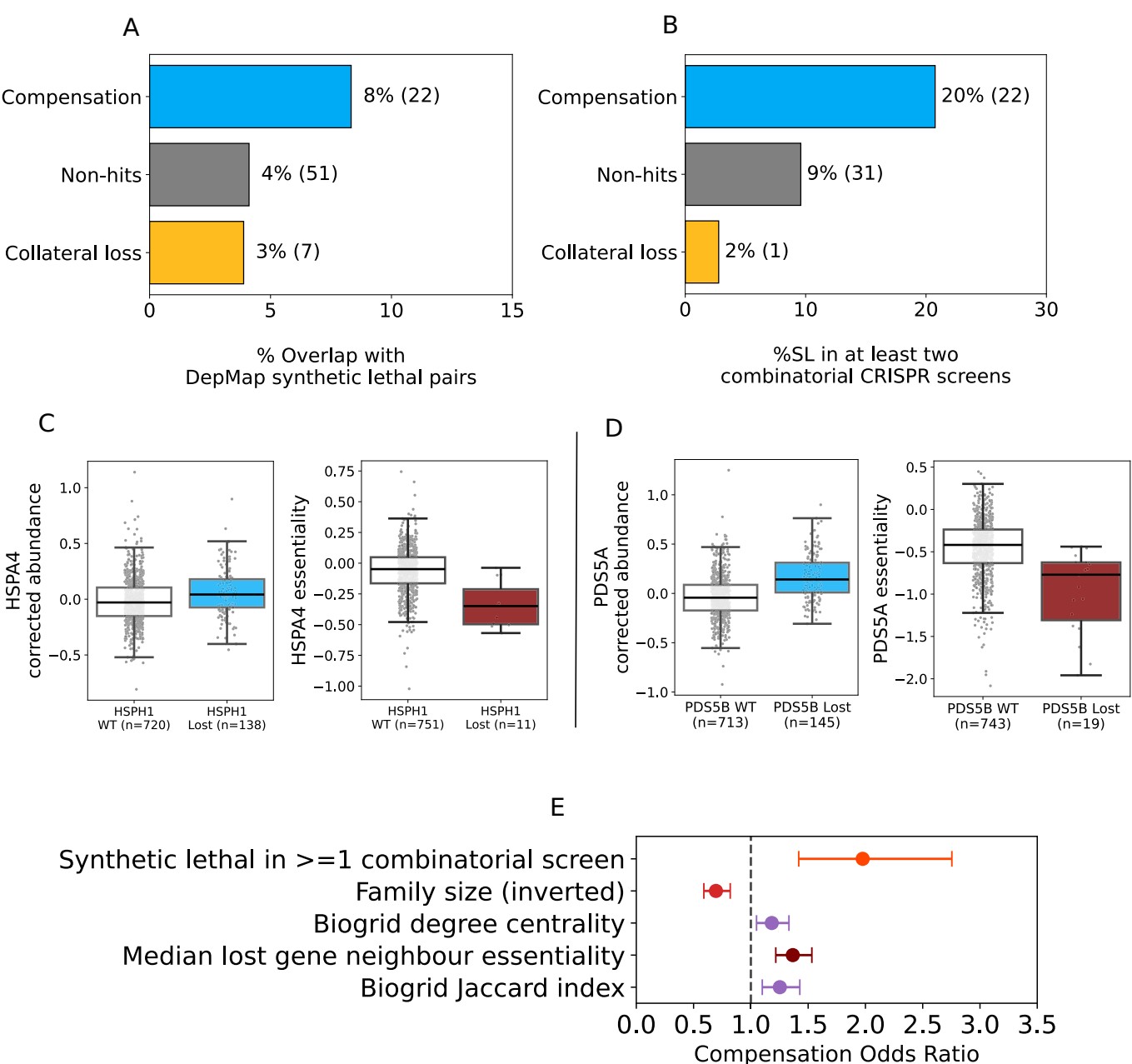

**Figure 5. Compensation pairs are more likely to be synthetic lethal.**

(A, B) Bar charts showing the percentage of compensation pairs, non-hits, and collateral loss pairs which are synthetic lethal based on analysis of (A) DepMap data (De Kegel et al, 2021) or (B) based on a consensus dataset assembled from five combinatorial CRISPR screens (Methods) (Parrish et al, 2021; Thompson et al, 2021; Dede et al, 2020; Gonatopoulos-Pournatzis et al, 2020; Ito et al, 2021). (C) Box plots showing that HSPA4 is more abundant and more essential (i.e. lower CERES score) in cells where *HSPH1* is lost. CPTAC data was used to plot abundance and DepMap (cell line) data was used to plot essentiality. Sample sizes for each group are shown in parentheses. (D) Similar box plots showing increase in *PDS5A* gene essentiality and PDS5A protein abundance upon *PDS5B* loss. In all boxplots, the central line represents the median, box limits indicate the 25th and 75th percentiles (first and third quartiles), and whiskers extend to 1.5 × interquartile range from either end of the box. Each grey dot represents a tumour sample in the abundance plots. Position on the y-axis in these plots indicates protein abundance adjusted for copy number of the encoding gene and cancer type, which are used as covariates in the regression model. Each grey dot represents a cell line in the essentiality plots, and position on the y-axis indicates the CERES dependency score of the gene in a given cell line. (E) Odds ratios with error bars for individual variables in a logistic regression model fit to predict proteomic compensation from synthetic lethality, family size, degree centrality, lost paralog neighbour essentiality, and paralog pair Jaccard index ($n = 1727$). Error bars represent 95% confidence intervals derived from the standard errors of the logistic regression coefficient estimates.

loss is associated with increased abundance of HSPA4 protein and *HSPA4* loss is associated with increased HSPH1 abundance. Our previous analysis of synthetic lethal relationships in a large cell line panel (DepMap) identified that cell lines that have lost *HSPH1* are more sensitive to the inhibition of *HSPA4* (Fig. 5C). An especially promising pair are the Precocious Dissociation of Sisters 5 paralogs *PDS5A* and *PDS5B*, which regulate the cohesin complex and play important roles in chromatid cohesion (Chan et al, 2013; Nishiyama et al, 2010). *PDS5B*, also known as *APRIN* and *AS3*, is a candidate tumour suppressor mutated or downregulated in a variety of cancer types (Xu et al, 2021; Brough et al, 2012; Kim et al, 2013; Zhang et al, 2008). Here we find that loss of *PDS5B* is associated with increased protein abundance of its paralog PDS5A (Fig. 5D). This pair was previously identified as a synthetic lethal pair in a high-throughput combinatorial CRISPR screen (Parrish et al, 2021), while our own previous analysis of single-gene knockout CRISPR screens suggested that loss of *PDS5B* was associated with increased dependency on *PDS5A* (Fig. 5D) (De Kegel et al, 2021). Our results here suggest that loss of *PDS5B* is associated with an increase in the protein abundance of PDS5A and a concordant increased dependency on its encoding gene.

The features we identified as predictive of proteomic compensation (shared protein–protein interactions, degree centrality, family size, neighbour essentiality) have also been found to be somewhat predictive of synthetic lethality (De Kegel et al, 2021). It is therefore possible that the enrichment for synthetic lethality simply reflects an enrichment for these features. However, integrating synthetic lethality with these four features into a single logistic regression demonstrated that displaying synthetic lethality was independently predictive of proteomic compensation. Indeed, synthetic lethality has the highest odds ratio estimate out of all variables in this model, suggesting a strong link between phenotypic and proteomic compensation (Fig. 5E).

## Compensation and collateral loss are driven by both transcriptional and post-transcriptional regulation

We wished to assess whether compensation and collateral loss hits arose through transcriptional regulation (i.e. loss of a protein results in altered transcription of its paralog) or through post-transcriptional mechanisms (e.g. loss of a protein results in altered translation or protein degradation of its paralog). As the CPTAC dataset contains matched genomes, proteomes, and transcriptomes, we were able to run the analysis pipeline outlined in Fig. 2A on transcriptomic profiles rather than proteomic profiles in order to identify cases of transcriptional compensation and collateral loss. Furthermore, we employed a linear regression approach (similar to that previously used by Gonçalves et al (Gonçalves et al, 2017) to create a protein residuals dataset where protein abundance was adjusted for transcriptional abundance, allowing us to directly assess effects mediated by post-transcriptional regulation (Methods; Appendix Fig. S7). We restricted our analysis to pairs quantified in both the proteomic and transcriptomic datasets. Testing the same 5128 paralog pairs with transcriptomic data, 633 compensation hits and 553 collateral loss hits were identified at the same FDR of 5% (Fig. 6A; Appendix Fig. S8A, Dataset EV6). The overlap between hits observed at the transcriptomic and proteomic level was significantly greater than expected by chance (OR = 13.4, $p = 4.2 \times 10^{-144}$ for compensation;

OR = 25.3, $p = 4.5 \times 10^{-195}$ for collateral loss) but still only partial—296 (~47%) compensation hits identified at the protein level were not significant at the mRNA level, and 326 (~51%) compensation hits identified at the mRNA level were not significant at the protein level (Fig. 6B). Testing the same set of pairs with the protein residual dataset, we identified 225 compensation and 169 collateral loss hits at an FDR of 5% (Fig. 6A; Appendix Fig. S8B, Dataset EV6). As might be expected, the majority of these residual hits were evident at the protein level (~92% of compensation hits and 76% of collateral loss) with a smaller overlap with the transcriptionally regulated pairs.

Hits identified with the transcriptomic and proteomic datasets but not the protein residual dataset likely correspond to cases of exclusively transcriptional regulation with no post-transcriptional component. For example, loss of *DAAM1* is associated with an increase in the transcript abundance and protein abundance of its paralog, DIAPH1 (Fig. 6C). DAAM1 and DIAPH1 are involved in actin polymerization and microtubule regulation (Lu et al, 2007; Bartolini and Gundersen, 2010; Luo et al, 2016). This relationship is not detectable with the protein residual dataset, suggesting that the protein-level relationship is entirely driven by transcript-level *DIAPH1* upregulation in response to *DAAM1* loss. Hits identified with the proteomic and protein residual datasets but not the transcriptomic dataset likely correspond to cases of exclusively post-transcriptional regulation. For example, loss of the transcriptional activator *SMARCA2* is associated with an increase in protein abundance of its paralog SMARCA4, but no corresponding relationship is visible at the transcript level (Muchardt et al, 1994) (Fig. 6D). SMARCA2 and SMARCA4 are mutually exclusive members of the chromatin remodelling SWI-SNF complex (Wilson et al, 2014). Hits identified with all three datasets likely correspond to cases of post-transcriptional regulation acting in the same direction as transcriptional regulation, resulting in an overall increase in protein abundance (a phenomenon previously observed in systematic studies of mRNA-protein correlation (Csárdi et al, 2015)). Compensation hits evident at both transcriptional and post-transcriptional levels include KIF2A-KIF2C, kinesin-13 proteins that regulate microtubule dynamics during mitosis (Manning et al, 2007). Loss of *KIF2A* is associated with increased mRNA abundance of KIF2C, suggesting transcriptional regulation, and an increase in the protein residuals of KIF2C, suggesting post-transcriptional regulation also (Fig. 6E). Some pairs were identifiable only when running the analysis with the post-transcriptional dataset, corresponding to cases where an effect at the post-transcriptional level is not identifiable with proteomic data alone due to transcript-level variation. However, the majority of compensation effects detected with the protein residual dataset are evident with proteomic data.

We also assessed the overlap between our compensation and collateral loss hits and the results of a Perturb-seq experiment where genes were inhibited with CRISPR interference (CRISPRi) and the transcriptional response of a paralogous gene was measured at the single-cell level (Methods) (Data ref: Replogle et al, 2022). Dividing our hits into three categories—transcriptional hits (detectable at the mRNA level), post-transcriptional hits (detectable at the protein residual level but not at the transcriptional level), and non-hits, we found that paralog pairs where the knockdown of one member causes transcriptional upregulation of the other member were significantly enriched in our set of

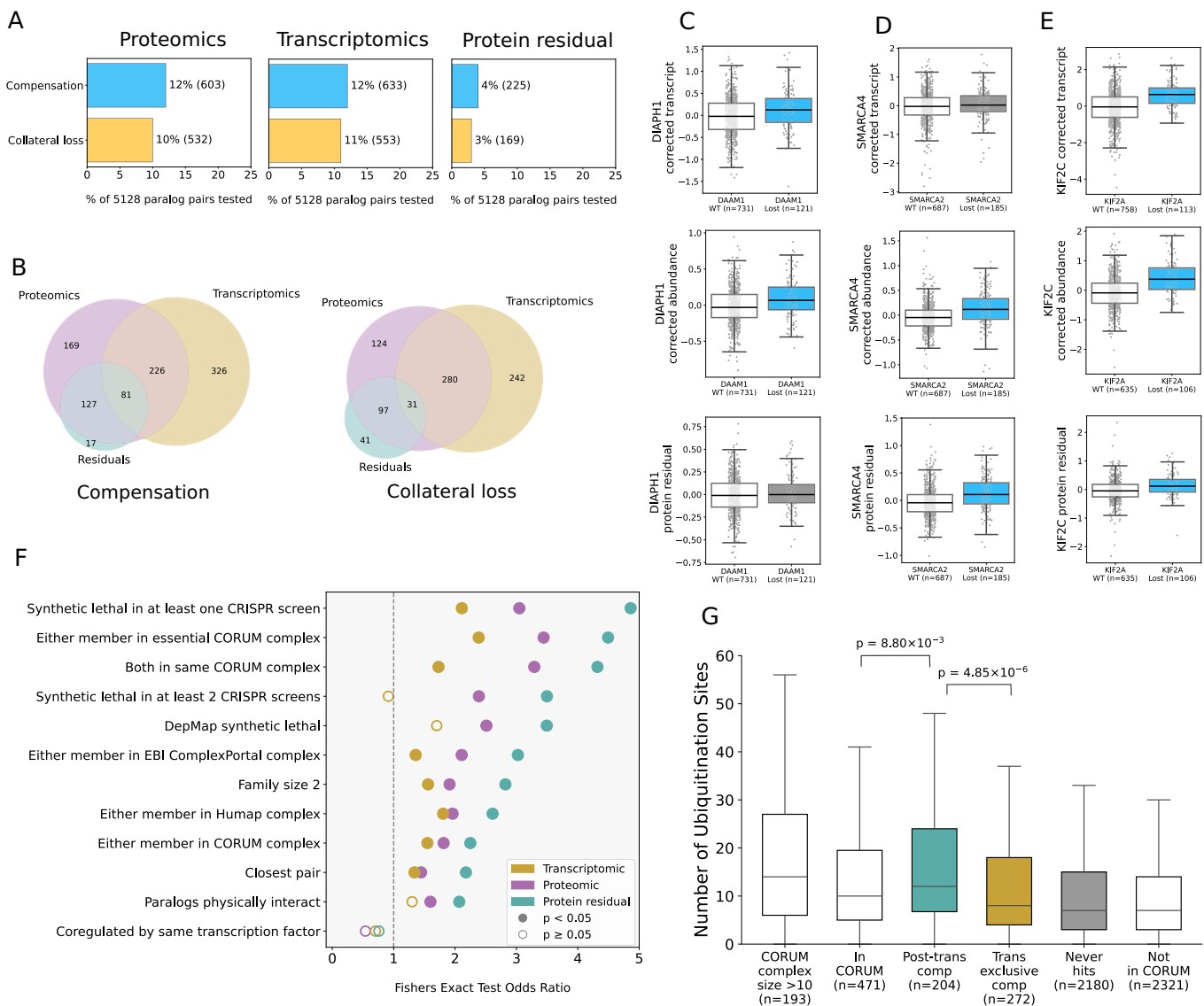

**Figure 6. Protein compensation is regulated transcriptionally and post-transcriptionally, with post-transcriptional pairs more enriched in synthetic lethality.**

(A) Bar charts showing the number of hits identified with the proteomic dataset, transcriptomic dataset, and protein residual datasets. (B) Venn diagrams showing the overlap between compensation and collateral loss hits identified with the transcriptomic, proteomic, and protein residual datasets. (C) Box plots showing DIAPH transcript, protein, and protein residual change associated with loss of its paralog *DAAM1*. Transcript and protein abundance shown has been corrected for lineage and self-copy number as these are covariates in the regression model. This pair is a transcriptionally-driven proteomic hit with no protein residual component. Sample sizes are shown in parentheses. (D) Similar box plots for SMARCA4-*SMARCA2*, a post-transcriptionally driven hit which is not a hit at the transcript level. (E) Similar box plots for KIF2C-*KIF2A*, a pair which is a compensation hit with transcriptomic data, proteomic data, and the protein residual dataset, suggesting transcriptional and post-transcriptional regulation (F) Odds ratios of significant and non-significant enrichments as calculated for proteomic, transcriptomic and protein residual compensation hits. (G) Box plot comparing number of ubiquitination sites between post-transcriptionally compensating paralogs (only observed in the residuals dataset), exclusively transcriptionally compensating paralogs (only observed transcriptionally), and non-compensating paralogs, with CORUM members, members of large CORUM complexes (no. of subunits >10) and non-CORUM members as controls (Methods). Sample sizes for each group are shown in parentheses. *p*-values shown are the results of two-sided Mann–Whitney tests comparing groups. In all boxplots, the central line represents the median, box limits indicate the 25th and 75th percentiles, and whiskers extend to 1.5 × interquartile range from either end of the box. Each grey dot in Fig. 6C–E represents a tumour sample. Sample sizes for each group are shown in parentheses.

transcriptional (OR = 7.3, $p = 2.4 \times 10^{-4}$) and proteomic (OR = 6.7, $p = 2.8 \times 10^{-4}$), but not post-transcriptional hits (OR = 2.5, $p = 0.4$) (Fig. EV5A). This provides orthogonal validation for our transcriptional compensation pairs and supports the idea that hits we observe at the protein residual level are not explainable by transcript-level variation. No significant enrichments were observed for collateral loss hits (Fig. EV5B).

## Post-transcriptional compensation pairs are more likely to be synthetic lethal and are more frequently ubiquitinated

We anticipated that compensatory relationships that are post-transcriptionally regulated might be more likely to involve members of protein complexes, as protein complex subunits have

previously been identified as being subject to more post-transcriptional regulation (Dephoure et al, 2014; Gonçalves et al, 2017; Sousa et al, 2019; Schukken and Sheltzer, 2022). We therefore tested post-transcriptionally regulated pairs (i.e. protein residual compensation hits) to see if they were enriched in protein complex subunits, alongside all previously tested relationships (i.e. synthetic lethality, closest pairs, smaller paralog families). As anticipated, we found that post-transcriptionally regulated compensation pairs were more enriched than transcriptionally regulated pairs for protein complex membership, regardless of the source of complex annotations (CORUM, ComplexPortal, HuMAP) (Fig. 6F). Surprisingly, we also found that post-transcriptionally regulated compensation pairs were more highly enriched in synthetic lethality than transcriptionally regulated pairs, suggesting that post-transcriptional regulation of paralog compensation is more predictive of synthetic lethality. We also observed a greater enrichment for physically interacting paralogs in the post-transcriptionally regulated set of compensation pairs (Fig. 6F). In contrast, transcriptionally-driven collateral loss hits were significantly less likely to be closest pairs, less likely to be each other's only paralogs, and less likely be in essential protein complexes (Appendix Fig. S9). Transcriptomic collateral loss hits showed the greatest enrichment for transcriptional co-regulation while post-transcriptional collateral loss hits showed no significant enrichment (Appendix Fig. S9).

Post-transcriptional regulation of compensatory genes could occur via a number of mechanisms, including increased translation of the associated transcript or reduced degradation of the resulting protein products. Previous work has suggested that, for protein complex subunits in particular, altered protein degradation may be a common mechanism by which cancer cells maintain protein complex balance in the face of copy number variation of individual subunits (Gonçalves et al, 2017; Sousa et al, 2019; Schukken and Sheltzer, 2022). Consistent with this, proteins with copy number variation that is attenuated at the protein level have greater numbers of ubiquitination sites (Schukken and Sheltzer, 2022; Senger et al, 2022). If paralog compensation effects are also regulated by altered protein degradation, we would expect to see greater numbers of ubiquitination sites in compensatory proteins. This should be especially evident for pairs that are regulated post-transcriptionally. To test this hypothesis, we compared the numbers of ubiquitination sites for the compensating paralog in pairs only observed transcriptionally (hits only in the transcriptional dataset), pairs that were regulated post-transcriptionally (hits in the protein residual dataset), and never-hits (i.e. A1s in A1-A2 pairs which were not hits in any dataset). As controls, we included proteins which are in large (>10 members) CORUM complexes (previously shown to be highly attenuated at the protein level (Gonçalves et al, 2017; Sousa et al, 2019; Schukken and Sheltzer, 2022), and proteins that never feature in a CORUM complex. All sets were restricted to paralogs tested for compensation/collateral loss in our analysis, to avoid any bias in the results. In line with our expectations, proteins which are never in a CORUM complex had the lowest number of ubiquitination sites (median 7), while members of large CORUM complexes had the highest number of sites (median 14) (Fig. 6G). Among our hits, post-transcriptionally compensating paralogs had the highest number of ubiquitination sites (median 12), significantly higher than compensating paralogs only observed transcriptionally (median 8) (Cliff's Delta = 0.2,

Mann–Whitney U test $p = 4.8 \times 10^{-6}$) (Fig. 6G). CORUM complex members overall had a median of 10 ubiquitination sites, significantly lower than our post-transcriptionally compensating pairs (Cliff's Delta = −0.1, $p = 8.8 \times 10^{-3}$). The higher number of ubiquitination sites on post-transcriptionally compensating proteins is consistent with a significant role for protein degradation in regulating paralog protein compensation. We observed a similar pattern for collateral loss—post-transcriptional hits had significantly greater numbers of ubiquitination sites than transcriptional hits (Cliff's Delta = 0.3, $p = 1.5 \times 10^{-6}$), while transcriptional hits had significantly lower numbers of ubiquitination sites than non-hits (Cliff's Delta = −0.2, $p = 1.3 \times 10^{-5}$) (Appendix Fig. S10).

# Discussion

Previous work has established that tumours and tumour cell lines are generally more tolerant of mutations and deletions of paralogs than singleton genes (Blomen et al, 2015; Zapata et al, 2018; De Kegel and Ryan, 2019, 2023). Other work has established that pairs of paralogs are frequently synthetic lethal, suggesting that they can directly compensate for each other's loss (Dede et al, 2020; Gonatopoulos-Pournatzis et al, 2020; Parrish et al, 2021; Thompson et al, 2021; Ito et al, 2021). How this compensation works at a molecular level has not been systematically explored in the context of cancer or indeed in any human cells. Here, we have used proteomic profiles of isogenic cell lines and tumour samples to understand how loss of one gene impacts the protein abundance of its paralog. We found evidence of relatively frequent proteomic compensation between paralogs as well as collateral loss, with the former being more common. Our subsequent analyses of the identified pairs suggest that compensatory relationships are more likely to be observed among pairs that are well connected on the protein–protein interaction network, involved in essential functions, and belong to smaller paralog families. In contrast, collateral loss is observed for pairs that are more peripheral on the protein–protein interaction network and less connected to essential processes. Compensatory relationships, unlike collateral loss relationships, are more likely to occur between pairs that are known to exhibit synthetic lethality. We note that the majority of paralog pairs do not exhibit compensatory relationships, suggesting that the loss of many genes can be tolerated without increasing the abundance of a paralog. This may be because the function of the lost gene is completely dispensable (i.e. there is no need to compensate for it), because the compensating paralog is already sufficiently abundant (i.e. compensation can happen without increased abundance), or because compensation happens via an alternative non-paralog mechanism.

Taken together, our results suggest a model whereby compensatory relationships stabilise essential protein–protein interaction subnetworks or complexes in the face of genetic perturbation in cancer. A LASSO regression model fit to predict proteomic compensation selects three interaction-based features (degree centrality, neighbour essentiality, and Jaccard index) and only one sequence-based feature (family size) from the full feature set, suggesting that the proteomic compensation effect is largely mediated by gene loss-induced changes to the protein–protein interaction network. Following loss of one gene, the protein abundance of a highly sequence similar paralog is increased, and

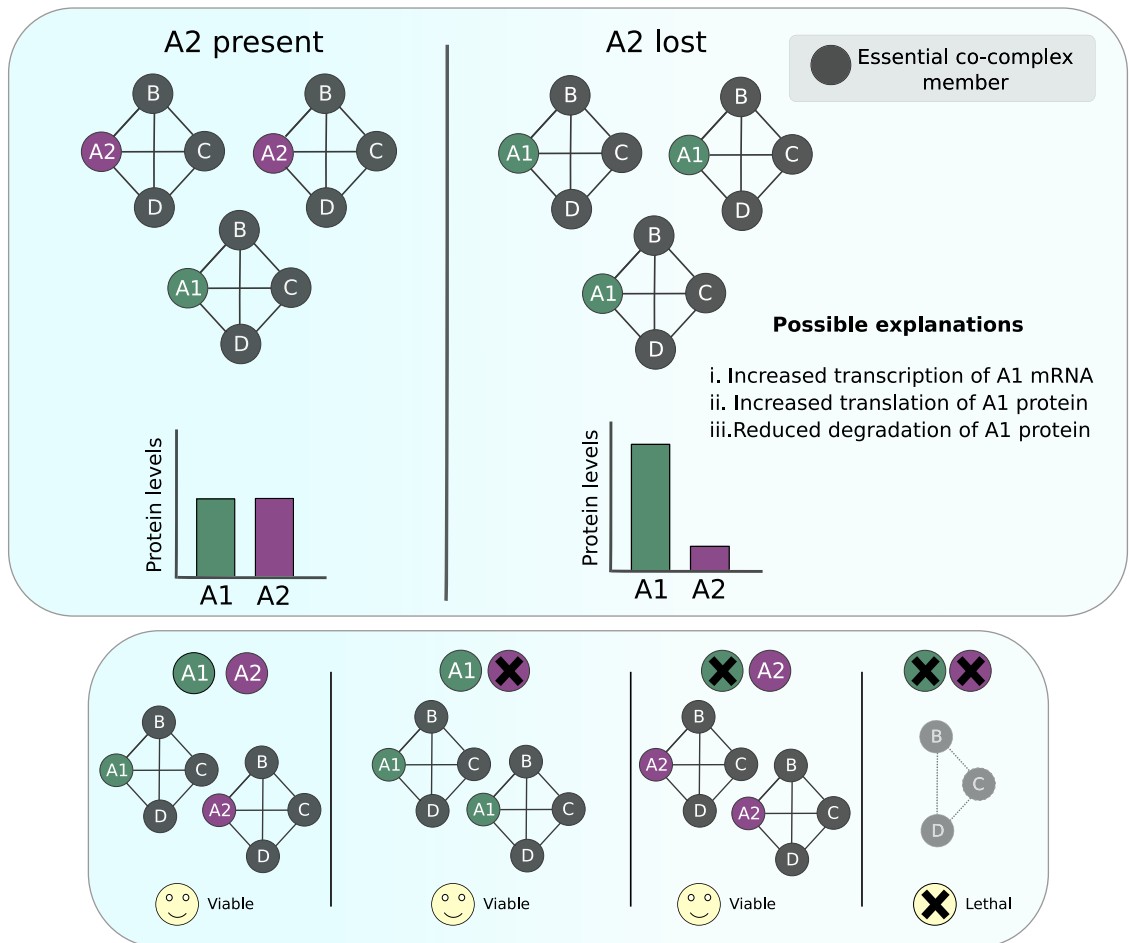

**Figure 7. Proposed model for protein–protein interaction network rewiring leading to proteomic compensation and synthetic lethality.**

Our results are consistent with a model where, for pairs with post-transcriptional compensation effects, increased incorporation of A1 into A2's essential protein interactions following A2 loss results in an increase of its abundance through reduced degradation of A1 subunits. A1 then becomes essential in backgrounds where A2 is lost, i.e. the pairs are synthetic lethal.

this protein takes the place of the lost gene in the protein–protein interaction network. Cancer cells then become dependent on this protein for survival (Fig. 7). Previous work has identified individual examples consistent with this model, for example loss of the cohesin subunit *STAG2* results in increased protein abundance of its paralog STAG1 and an increased sensitivity to genetic perturbation of *STAG1*; our results suggest that this may be a relatively common phenomenon (van der Lelij et al, 2017; Adane et al, 2021).

## Mechanisms of compensation and collateral loss

Compensatory relationships could occur through a number of mechanisms, including transcriptional regulation or post-transcriptional regulation. Our analysis suggests that both types of regulation contribute to compensation effects observed at the level of total protein abundance.

Considering only regulation of mRNA abundance, a diversity of mechanisms could be employed. Perhaps most obviously, paralogous transcription factors could directly regulate each other's transcription. However, alternative means of regulating mRNA

abundance also exist. For example, in mice, the ribosomal paralog Rpl22 regulates the protein abundance of its paralog *Rpl22l1* by directly binding and destabilising *Rpl22l1* mRNA (O'Leary et al, 2013). Also in mice, MBNL1, an RNA binding protein, regulates the splicing of its paralog *MBNL2* such that loss of *MBNL1* results in the altered inclusion of specific exons and a more stable protein (Nitschke et al, 2023). An alternative mechanism, termed genetic compensation or transcriptional adaptation, has recently been identified whereby nonsense mutation of one paralog may trigger an increase in the transcription of its paralog. It is not fully clear how this increased transcription occurs, but the mechanism appears to be dependent on the nonsense-mediated decay pathway and so is unlikely to be relevant for pairs triggered solely by copy loss (El-Brolosy and Stainier, 2017; El-Brolosy et al, 2019).

Post-transcriptional regulation is an important mechanism by which cancer cells adapt to genetic perturbation. In particular, multiple studies have found that copy number variation in cancer often results in concordant effects on mRNA abundance but that this is frequently attenuated at the protein level, i.e. a doubling of copy number may lead to a similar increase in mRNA abundance but a much smaller increase in protein abundance (Gonçalves et al,

2017; Sousa et al, 2019; Cheng et al, 2022). Genes subject to this attenuation are significantly more likely to encode subunits of protein complexes and also have more known ubiquitination sites, suggesting potential regulation via protein degradation (Gonçalves et al, 2017; Sousa et al, 2019; Schukken and Sheltzer, 2022). Other work has established that many protein complex subunits are produced in excess, with those proteins not incorporated into complexes subsequently degraded, providing a means of maintaining balance between subunits (McShane et al, 2016; Taggart and Li, 2018; Taggart et al, 2020; Hsu et al, 2022). Our finding that paralog compensation pairs, especially those that are post-transcriptionally regulated, are enriched in protein complex subunits and more highly ubiquitinated is in line with this model. If both members of a paralog pair compete for membership of the same protein complex, as is the case for many pairs, then any orphan subunits not incorporated into a complex will be degraded. When one paralog has reduced abundance, resulting from mutation or deletion, its counterpart will have increased incorporation into the complex, reduced degradation, and so increased abundance. This form of compensation could occur without any need for altered transcription. Further experiments will be required to disentangle the relative contributions of different regulatory mechanisms for compensatory pairs.

We do not have a simple model for collateral loss pairs. The simplest model, supported by systematic work in yeast, would be that pairs of paralogs often form heterodimers and hence stabilise each other (Diss et al, 2017; Dandage and Landry, 2019). When one paralog is lost, the other loses stability and is then degraded. The strongest effect we see, GMPPA/GMPPB, corresponds to a pair known to form a heterodimer, consistent with this model. However, we do not find direct physical interaction to have substantial predictive power for collateral loss. As such, this too is unlikely to explain all collateral loss pairs and so additional explanations are needed. Since there is considerable overlap between collateral loss hits identified at the transcript level and those identified at the protein level, it is also likely that many of these relationships involve some transcriptional co-regulation. We found that collateral loss pairs are more likely to be transcriptionally co-regulated than non-hits, although this only applies for 6% of pairs and so likely does not explain the majority of effects we observe.

## Why do active protein compensation relationships exist?

The compensatory relationships we observe here are only evident in the face of a genetic perturbation—loss or mutation of one paralog triggers an increase in the abundance of another. It is extremely unlikely that these compensatory relationships have evolved as a means of protecting cells against future, unseen genetic perturbations such as those that occur in cancer (Krakauer and Nowak, 1999; Lynch and Conery, 2000). One potential explanation, proposed for compensatory relationships observed in yeast, is that these regulatory relationships exist as a means of buffering cells against variation in gene expression so that the total abundance of a paralog pair is maintained despite fluctuations in the abundance of each individual paralog (Kafri et al, 2009). In this way the compensatory relationships we observe in cancer, that contribute to genetic robustness, may simply reflect existing regulatory relationships that have evolved to protect cells against stochastic and environmental variation.

## Limitations of experiments and data analysis

Although we have taken care to exclude confounding variables and ensure that our analysis is robust, there are some caveats to working with these datasets that inherently limit our analysis. Previous analysis has shown that many HAP1 'knockouts' display residual expression of the target transcripts, even in the absence of protein expression, potentially due to variable efficiency of nonsense-mediated decay (Smits et al, 2019). In our analysis we have only analysed HAP1 clones where we could validate a reduction in protein abundance of the target gene. We have not assayed mRNA abundance and so cannot rule out the possibility that residual expression of target transcripts may influence the observed compensation and collateral loss patterns.

Furthermore, a challenge with bottom-up proteomics is that protein abundance is quantified from the abundance of peptides which may match multiple proteins, which is especially common for paralogs. The CPTAC PanCan dataset was re-processed by the authors using FragPipe and the Philosopher pipeline, with razor peptides being assigned to a single protein based on the overall peptide evidence for that protein (Li et al, 2023). In our analysis of HAP1 proteomics we quantified all protein abundances using unique peptides only (those that map to only a single protein), which meant that we could not analyse the pair ASF1A-ASF1B because only shared peptides could be identified. Since peptide-level data is not readily available for the CPTAC dataset, it was not possible to exclude shared peptides. Although some collateral loss hits might reflect artefacts resulting from shared peptides, compensation hits, where the abundances of two paralogs are inversely correlated, are less likely to be affected. Furthermore, approximately half of the compensation and collateral loss hits are also evident at the transcriptomic level and these are unlikely to result from shared peptide artefacts.

Although we identify a number of statistical associations between gene loss and paralog abundance in the CPTAC dataset, further work is required to establish causality of these associations. Our assumption is that loss of one gene causes an increase in the abundance of another paralog. However, it is possible that, at least in some cases, causality flows the other direction and it is an increased abundance of a paralog that allows the copy loss to occur. Indeed in the case of cancer drug resistance, often resistant states exist in the cell population prior to drug treatment (Gerlinger et al, 2012; Prieto-Vila et al, 2019; Jacobo Jacobo et al, 2024; Emert et al, 2021). We cannot rule out the possibility that some of our 'compensatory' relationships reflect this alternative framing. However, the significant overlap between our compensation hits and those from Perturb-seq experiments would argue that our hits are enriched for real regulatory relationships, but some may reflect the case where higher-expression predates the genetic event.

Our analysis focussed on copy number loss events, which often affect multiple genes. As such, to avoid identifying spurious collateral loss pairs which are actually explained by the same deletion event, we excluded 1,400 paralog pairs which met all other filtering criteria, but were on the same chromosome. However, since HAP1 knockouts were generated using CRISPR-Cas9, this was not a concern with the HAP1 analysis. We tested two pairs, KATNAL1-VPS4B and DDX42-DDX5 which are on the same chromosome, and did not identify a significant effect for either pair.

## Future directions

Combinatorial CRISPR screens have revealed that many paralog pairs are synthetic lethal in a context-specific fashion, with some pairs synthetic lethal in one cell line but not others (Thompson et al, 2021; Ito et al, 2021; Parrish et al, 2021). Work in yeast has also established that at least some proteomic compensation relationships are only evident in specific contexts where such compensation is necessary for survival, e.g. for glucose metabolism in environments where glucose is the primary carbon source (DeLuna et al, 2010). In order to gain statistical power, we have here focussed on compensatory relationships that can be observed across samples from all cancer types combined. It is still unclear to what extent there may also be compensatory relationships evident within specific contexts or cancer types. This may be possible to address as the number of samples available for individual cancer types increases.

Our CPTAC analysis also was entirely focussed on detecting cases of paralog compensation caused by single copy loss in cancer. Another possible scenario is compensation by increased expression of the remaining allele—although we find that a majority of single copy loss events result in a significant drop in protein abundance, a systematic analysis of intra-locus compensation is an interesting possibility for future work.

We found that pairs that exhibit proteomic compensation are enriched among gene pairs with a known synthetic lethal relationship. Synthetic lethality also outperforms all other predictors of proteomic compensation in our logistic regression analysis. Previous work has identified a number of synthetic lethal relationships between pairs that exhibit proteomic compensation, including *STAG1/STAG2* and *SMARCA2/SMARCA4*. Our finding that there is a general relationship between the two is consistent with earlier reports in yeast, where 22 pairs that displayed compensatory relationships were found to be enriched in known synthetic lethal pairs (DeLuna et al, 2010). As the majority of paralog pairs have yet to be experimentally tested for synthetic lethality, the hundreds of protein compensation pairs we have identified represent candidates worth prioritising for future study. As these involve genes recurrently lost in cancer, they may be especially promising from a therapeutic point of view. Among the gene pairs with a compensatory relationship, and a known synthetic lethality, *PDS5A/PDS5B* appears especially promising as a therapeutic target—there is evidence of synthetic lethality both in a large scale analysis of diverse cell lines (De Kegel et al, 2021) and in a combinatorial screen (Parrish et al, 2021), and *PDS5B* is recurrently deleted in patient tumours.

# Methods

### Reagents and tools table

| Reagent/Resource | Reference or Source | Identifier or Catalog Number |
|---|---|---|
| **Experimental models** | | |
| HAP1_WT_KO | Horizon Discovery | C631 |
| HAP1_ARFGAP2_KO | Horizon Discovery | HZGHC84364 |
| HAP1_ARFGAP3_KO | Horizon Discovery | HZGHC26286 |

| Reagent/Resource | Reference or Source | Identifier or Catalog Number |
|---|---|---|
| HAP1_ARID1A_c003_KO | Horizon Discovery | HZGHC000618c003 |
| HAP1_ARID1A_c010_KO | Horizon Discovery | HZGHC000618c010 |
| HAP1_ARID1B_KO | Horizon Discovery | HZGHC000582c007 |
| HAP1_ASF1A_KO_clone4 | Horizon Discovery | HZGHC007215c004 |
| HAP1_ASF1A_KO_clone5 | Horizon Discovery | HZGHC007215c005 |
| HAP1_ASF1B_KO | Horizon Discovery | HZGHC55723 |
| HAP1_CDK8_KO | Horizon Discovery | HZGHC000325c001 |
| HAP1_CDK19_KO | Horizon Discovery | HZGHC000078c020 |
| HAP1_COPS7A_KO | Horizon Discovery | HZGHC000914c001 |
| HAP1_COPS7B_KO | Horizon Discovery | HZGHC000636c011 |
| HAP1_CHMP1A_KO | Horizon Discovery | HZGHC005340c003 |
| HAP1_CHMP1B_KO | Horizon Discovery | HZGHC007219c010 |
| HAP1_CREBBP_KO | Horizon Discovery | HZGHC001109c005 |
| HAP1_EP300_KO | Horizon Discovery | HZGHC001318c011 |
| HAP1_DDX5_KO | Horizon Discovery | HZGHC006136c012 |
| HAP1_DDX17_KO | Horizon Discovery | HZGHC007221c009 |
| HAP1_KAT2A_KO | Horizon Discovery | HZGHC000929c001 |
| HAP1_KAT2B_KO | Horizon Discovery | HZGHC000948c012 |
| HAP1_KMT2C_KO | Horizon Discovery | HZGHC003909c012 |
| HAP1_MAGT1_KO | Horizon Discovery | HZGHC84061 |
| HAP1_TUSC3_KO | Horizon Discovery | HZGHC004076c007 |
| HAP1_MBD2_KO | Horizon Discovery | HZGHC003628c010 |
| HAP1_MBD3_KO | Horizon Discovery | HZGHC001113c009 |
| HAP1_RAB1A_KO | Horizon Discovery | HZGHC007227c002 |
| HAP1_RAB1B_KO | Horizon Discovery | HZGHC001225c011 |
| HAP1_SEC23A_KO | Horizon Discovery | HZGHC001218c001 |
| HAP1_SEC23B_KO | Horizon Discovery | HZGHC020241c010 |
| HAP1_SEC31A_KO | Horizon Discovery | HZGHC22872 |
| HAP1_SEC31B_KO | Horizon Discovery | HZGHC25956 |
| HAP1_SMARCA2_KO | Horizon Discovery | HZGHC004055c012 |
| HAP1_SMARCA4_KO | Horizon Discovery | HZGHC002878c004 |
| HAP1_SMARCC1_KO | Horizon Discovery | HZGHC002476c011 |
| HAP1_SMARCC2_KO | Horizon Discovery | HZGHC002783c011 |
| HAP1_VPS4A_KO | Horizon Discovery | HZGHC004623c005 |
| HAP1_VPS4B_KO | Horizon Discovery | HZGHC006889c011 |
| **Recombinant DNA** | | |
| N/A | | |
| **Antibodies** | | |
| N/A | | |
| **Oligonucleotides and other sequence-based reagents** | | |
| N/A | | |
| **Chemicals, Enzymes and other reagents** | | |
| Guanidine-Hydrochloride | Merck Life Science Limited | 369080 |
| HEPES buffer | Thermofisher | 15630106 |

| Reagent/Resource | Reference or Source | Identifier or Catalog Number |
|---|---|---|
| Chloracetamide | Sigma-Aldrich | C0267-100G |
| Tris (2-carboxyethyl) phosphine hydrochloride | C4706 | Sigma-Aldrich |
| LysC | V1671 | Promega |
| Trypsin | V5111 | Promega |
| C18 stage tip | 2215 | Empore |
| Trifluoroacetic acid | 302031-100 ML | Sigma-Aldrich |
| IMDM | Corning | 10-016-CV |
| FBS | Thermo Fisher Scientific | 10270-106 |
| **Software** | | |
| MaxQuant 2.6.4.0 | https://maxquant.org/ | |
| Python 3.8.10 | https://www.python.org/ | |
| Pandas 2.0.3 | https://pandas.pydata.org/ | |
| Numpy 1.24.4 | https://numpy.org/ | |
| tqdm v4.66.4 | https://github.com/tqdm/tqdm | |
| Pandarallel v1.6.5 | https://github.com/nalepae/pandarallel | |
| Statsmodels v0.14.1 | https://www.statsmodels.org/ | |
| Scipy v1.10.1 | https://scipy.org/ | |
| NetworkX v3.1 | https://networkx.org/ | |
| Matplotlib v3.7.4 | https://matplotlib.org/ | |
| Seaborn v0.13.2 | https://seaborn.pydata.org/ | |
| Matplotlib-venn v0.13 | https://pypi.org/project/matplotlib-venn/ | |
| Inkscape v1.3 | https://inkscape.org/ | |
| Scikit-learn v1.3.2 | https://scikit-learn.org/ | |
| **Other** | | |
| Mass spectrometry | Thermo-Fisher | Q-Exactive Plus |

## Strategy for selection of paralog pairs for HAP1 knockouts

Seventeen paralog pairs were chosen to be knocked out in a HAP1 background to measure the effect of the loss of a paralog on the other member of a pair. From a candidate set of genes that are non-essential in the HAP1 background and detectable in the HAP1 transcriptome, we selected pairs that had a known or predicted synthetic lethal relationship (Table 1, Methods).

## Proteomic profiling of HAP1 knock-out (KO) cell lines

For protein abundance experiments, samples were collected in lysis buffer containing 6 M Guanidine-Hydrochloride (Gu-HCl), 200 mM

HEPES pH 8.5, 1 mg/ml Chloracetamide, 0 and 1.5 mg/ml TCEP Tris (2-carboxyethyl) phosphine hydrochloride. Cells were cultured in IMDM supplemented with 10% FBS. Cell pellets were probe sonicated and boiled at 95 °C for 5 min. Cell lysates were clarified by centrifugation ($20,000 \times g$, 5 min). HAP1 KO cell lines were lysed and digested in 3 batches. Batch 1 consisted of WT, ARFGAP2_KO, ARFGAP3_KO, ARID1A_c003_KO, ARID1A_c010_KO, ARID1B_KO, ASF1A_C4_KO, ASF1A_C5_KO, ASF1B_KO, CDK8_KO, CDK19_KO, CHMP1A_KO, and CHMP1B_KO cell lines. Batch 2 consisted of WT, COPS7A_KO, COPS7B_KO, DDX5_KO, DDX17_KO, CREBBP_KO, EP300_KO, KAT2A_KO, KAT2B_KO, KMT2C_KO, MAGT1_KO, and TUSC3_KO cell lines. Batch 3 consisted of WT, MBD2_KO, MBD3_KO, RAB1A_KO, RAB1B_KO, SEC23A_KO, SEC23B_KO, SEC31A_KO, SEC31B_KO, SMARCA2_KO, SMARCA4_KO, SMARCC1_KO, SMARCC2_KO, VPS4A_KO, and VPS4B_KO cell lines. 200 µg of proteins were digested by adding 0.4 µg LysC per sample for 4 h at 37 °C. Samples were diluted to 1 M Gu-HCl and then digested with Trypsin for 16 h at 37 °C. Trypsin activity was inhibited by acidification of samples to a concentration of 1% trifluoroacetic acid. Digests were clarified by centrifugation ($20,000 \times g$, 5 min), samples were desalted on a C18 Stage tip, and eluates were analysed by HPLC coupled to a Q-Exactive Plus mass spectrometer as described previously but with an extended gradient of 120 min et al, 2014). Peptides and proteins were identified and quantified with the MaxQuant software package (2.6.4.0), and label-free quantification was performed by MaxLFQ (Cox et al, 2014). The search included variable modifications for the oxidation of methionine, protein N-terminal acetylation, and carbamidomethylation as fixed modification. The false discovery rate, determined by searching a reverse database, was set at 0.01 for both peptides and proteins. Proteins identified were filtered to remove reverse sequences (which are used only for False Discovery Rate estimation) and potential contaminants. LFQ intensity values were log2-transformed to normalize the data distribution. A mean centering procedure was applied to adjust for systematic differences between samples. This involved calculating sample-specific adjustment factors based on the difference between each sample's mean and the global mean. These factors were then subtracted from all LFQ intensities. Proteins quantified in fewer than 20% of the samples were dropped. Missing values were imputed in each sample with the lowest protein measurement for that sample (Turriziani et al, 2014). Peptides and proteins were identified and quantified with the MaxQuant software package (2.6.4.0), and label-free quantification was performed by MaxLFQ (Cox et al, 2014). The search included variable modifications for the oxidation of methionine, protein N-terminal acetylation, and carbamidomethylation as fixed modification. The false discovery rate, determined by searching a reverse database, was set at 0.01 for both peptides and proteins. Proteins identified were filtered to remove reverse sequences (which are used only for False Discovery Rate estimation) and potential contaminants. LFQ intensity values were log2-transformed to normalize the data distribution. A mean centering procedure was applied to adjust for systematic differences between samples. This involved calculating sample-specific adjustment factors based on the difference between each sample's mean and the global mean. These factors were then subtracted from all LFQ intensities. Proteins quantified in fewer than 20% of the samples were dropped. Missing values were imputed in each sample with the lowest protein measurement for that sample.

**Table 1.** Rationale for choosing pairs to knockout in HAP1 background.

| Gene pair | Synthetic lethal | Synthetic lethal in | Protein complex members |
|---|---|---|---|
| CHMP1A-CHMP1B | Yes | Thompson et al, Parrish et al (Thompson et al, 2021; Parrish et al, 2021) | Yes |
| SMARCC1-SMARCC2 | No | NA | Yes |
| DDX17-DDX5 | Yes | Ito et al (Ito et al, 2021) | Yes |
| CDK19-CDK8 | No | NA | Yes |
| MBD2-MBD3 | Yes | De Kegel et al (De Kegel et al, 2021) | Yes |
| CREBBP-EP300 | Yes | De Kegel et al, Parrish et al, Gonatopoulos-Pournatzis et al, Ito et al (De Kegel et al, 2021; Parrish et al, 2021; Gonatopoulos-Pournatzis et al, 2020; Ito et al, 2021) | Yes |
| SEC31A-SEC31B | No | NA | No |
| SEC23A-SEC23B | Yes | Thompson et al, Parrish et al, Gonatopoulos-Pournatzis et al (Thompson et al, 2021; Gonatopoulos-Pournatzis et al, 2020; Parrish et al, 2021) | No |
| RAB1A-RAB1B | Yes | Gonatopoulos-Pournatzis et al (Gonatopoulos-Pournatzis et al, 2020) | No |
| VPS4A-VPS4B | Yes | De Kegel et al, Ito et al (De Kegel et al, 2021; Ito et al, 2021) | No |
| ARFGAP2-ARFGAP3 | No | NA | No |
| KAT2A-KAT2B | Yes | De Kegel et al, Thompson et al, Ito et al (De Kegel et al, 2021; Ito et al, 2021) | Yes |
| ARID1A-ARID1B | Yes | De Kegel et al, Thompson et al, Parrish et al, Gonatopoulos-Pournatzis et al, Ito et al (De Kegel et al, 2021; Thompson et al, 2021; Parrish et al, 2021; Gonatopoulos-Pournatzis et al, 2020; Ito et al, 2021) | Yes |
| SMARCA2-SMARCA4 | Yes | De Kegel et al, Thompson et al, Ito et al (De Kegel et al, 2021; Thompson et al, 2021; Ito et al, 2021) | Yes |
| MAGT1-TUSC3 | Yes | Gonatopoulos-Pournatzis et al (Gonatopoulos-Pournatzis et al, 2020) | Yes |
| COPS7A-COPS7B | Yes | Dede et al (Dede et al, 2020) | Yes |
| ASF1A-ASF1B | Yes | Parrish et al (Parrish et al, 2021) | Yes |

Synthetic lethality is defined as being synthetic lethal either in any background in at least one out of five combinatorial CRISPR screens, or being a gene pair where the loss of one gene results in an increase in essentiality of the other gene (based on DepMap data) (Dede et al, 2020; Parrish et al, 2021; Ito et al, 2021; Gonatopoulos-Pournatzis et al, 2020; Thompson et al, 2021; De Kegel et al, 2021). Protein complex membership is defined as either member of the pair being annotated as a member of a CORUM complex (Tsitsiridis et al, 2023).

## Differential expression of paralogous proteins upon knockout of the other paralog

A two-tailed equal variance t-test was performed for every HAP1 clone to compare the abundance of the knocked-out protein in the knockout cell lines vs. its abundance in the wild-type cell lines. Four biological replicates were profiled for every knockout while the wild-type had twelve biological replicates. Out of 34 candidate genes, this test could not be performed for 6 knockouts as their cognate proteins were not measured in any of the HAP1 wild-type samples. Tests were also not performed for ASF1A/ASF1B since MaxQuant failed to calculate unique abundance values for each protein (due to all measured peptides being shared between both paralogs). Following multiple testing correction using the Benjamini–Hochberg method, clones without significantly (FDR < 5% and uncorrected $p$-value < 0.05) lower protein abundance in the knockout were dropped from the analysis (3 out of 26 clones). For the 23 clones with validated knockouts, a similar t-test was performed to compare the abundance of each knocked out gene's paralogs. This was done for all paralogs that were measured in at least one sample. In addition to assessing paralog pairs with symmetric knockouts, i.e. A1-A2 pairs where we had knockouts for both A1 and A2, we also compared the levels of other paralogs in the family (see below for paralog selection criteria), e.g. In the case of the pair RAB1A-RAB1B, we compared RAB1A abundance in *RAB1B* knockout (KO) vs wild-type (WT) and RAB1B abundance in *RAB1A* KO vs WT. We also compared the abundances of RAB18 and

RAB35 in both these knockouts with the wild types. In this way, a total 46 t-tests were performed (Fig. 1). All t-tests were performed with 4 samples in the knockout group and 12 samples in the wild type group, i.e. 14 degrees of freedom.

## Detecting paralog compensation and collateral loss in primary tumours

The CPTAC PanCan datasets, containing collated and normalized proteomics and transcriptomics data for samples from 9 CPTAC studies, were used for this analysis (Data ref: Li et al, 2023) (Li et al, 2023). Copy number data was obtained from the PanCan CPTAC GISTIC dataset, which contained absolute gene-level copy number values. Ovarian cancer samples were dropped since this study had fewer than 90 samples. CPTAC PanCan data is accessible from the Proteomic Data Commons (https://proteomic.datacommons.cancer.gov/pdc/cptac-pancancer) (Proteomics: Proteome_BCM_GENCODE_v34_harmonized_v1.zip- all tumour datasets concatenated and Ensembl IDs mapped to gene symbols using HUGO Gene Nomenclature Committee, i.e. HGNC mapping, gene-level RNAseq files from tumour datasets and gene-level GISTIC copy number calls for tumor datasets) (Seal et al, 2023; Thangudu et al, 2020). In total, the datasets had 1023 samples, with each study representing a different tissue type. The cancer types represented in the final dataset are pancreatic ductal adenocarcinoma, breast cancer, clear cell renal cell

carcinoma, lung adenocarcinoma, head and neck squamous cell carcinoma, lung squamous cell carcinoma, glioblastoma, colon cancer, and uterine corpus endometrial carcinoma.

## Selection criteria for paralog pairs

A symmetric set of 83,404 paralog pairs was obtained from Ensembl release 93 (Zerbino et al, 2018). Pairs with amino acid sequence identity ≤0.2 and family size ≥20 were excluded. Paralog pairs were excluded when either paralogous protein was measured in fewer than half the samples present in the CPTAC dataset. To minimize the possible confounding effect of co-regulation or linkage, paralog pairs were excluded if both members of the pair were located on the same chromosome (Fig. EV2).

## Calling gene loss

Gene losses were annotated for every sample within each dataset. Absolute copy number values called using the GISTIC algorithm were used to annotate hemizygous deletions (i.e. copy number of −1 in a given sample instead of a copy number of 0) (Mermel et al, 2011). Gene losses were annotated for all genes that were members of suitable paralog pairs (as determined in the previous section). Genes were then filtered to a set that was hemizygusly lost in at least 20 samples. Genes were dropped when their protein abundance was not quantified in at least 20 of the samples where they were hemizygously lost. Gene losses were only considered to be valid if they were associated with a significant drop in the abundance of their cognate protein. To calculate this, a linear model was fit using ordinary least squares with protein abundance as the dependent variable, gene loss as the independent variable, and tissue-of-origin/study (as each CPTAC study corresponded to a different cancer type) for the sample as a covariate. The two-tailed $p$-values for the t-statistic of the gene loss variable in each model was calculated and nominally significant proteins with a $p$-value below 0.05 were retained. In this way, we identified cases where the loss of a gene affected its protein abundance. Paralog pairs where at least one member was lost in a minimum of 20 samples, with this loss being associated with a significant drop in protein abundance, were retained.

## Generation of protein residual datasets

A 'protein residual' dataset was generated to identify relationships between paralogs that arose solely post-transcriptionally. This was done by regressing out self-transcript abundance from protein abundance for each sample. In order to appropriately account for the effect of tissue-of-origin, which may impact mRNA and protein abundance separately, linear models were built for each gene to explain mRNA abundance and protein abundance (separately) with the study/tissue type variable. The residuals of these models, i.e. study corrected mRNA and study-corrected protein, were used to obtain protein residuals. This was done by fitting linear models to explain study-corrected protein abundance using study-corrected mRNA abundance and taking the residuals. (Appendix Fig. S7).

## Paralog compensation and collateral loss analysis

Gene losses were annotated in every sample and paralog pairs were filtered to a testable set as detailed above. For every paralog pair A1-A2, the dataset was partitioned into a set where A2 had been

hemizygously lost, and a set where two copies of A2 were present (excluding samples with amplifications or homozygous deletions). A linear model was then fit to explain protein abundance of the other paralog, A1, based on A2 loss (as a binary variable), cell line lineage/ tumour type, and A1 copy number. Lineage was included as a covariate to minimize the confounding effect of tissue-specific expression patterns. A1's own copy number was also included as a covariate to avoid misattributing variation caused by self-copy number changes to A2 loss. In total, 5128 pairs were tested when calling loss based on hemizygous copy number loss. Two-tailed $p$-values for the t-statistic of the A2 loss variable in each model were extracted, and following Benjamini–Hochberg correction for multiple testing, any pairs with False Discovery Rate (FDR) for A2 loss <5% and uncorrected $p$-value < 0.05 were considered to be hits- either compensation if A2 loss was associated with increased A1 abundance, or collateral loss if A2 loss was associated with decreased A1 abundance. The analysis was run separately for the proteomic, transcriptomic, and protein residual datasets detailed above. Lineage and self-copy number were not included as covariates for the protein residual regressions as these variables had already been regressed from the data.

## Assessing the enrichment of collateral loss/ compensation using binary variables

Several factors were assessed for enrichment in the sets of compensation and collateral loss hits (separately, as compared to non-hits). Only unique pairs were retained for this analysis. These included synthetic lethality, protein complex membership of either paralog, membership of both paralogs to the same protein complex, whether or not the pair is a 'closest pair', i.e. the members of the pair are each other's most sequence-similar paralogs and whether or not any other paralogs belong to the family. Synthetic lethal gene pairs from combinatorial screens were obtained from a consensus dataset previously generated by De Kegel et al, containing the results of four combinatorial CRISPR screens (Dede et al, 2020; Gonatopoulos-Pournatzis et al, 2020; Parrish et al, 2021; Thompson et al, 2021; De Kegel et al, 2021). This was augmented with hits from another combinatorial CRISPR screen (Ito et al, 2021). To ensure only synthetic lethal pairs, rather than negative genetic interaction pairs, were retained from Ito et al we filtered based on log fold change as performed for the Gonatopoulos-Pournatzis dataset in De Kegel et al An LFC threshold of −0.8 was used for the Ito et al screen. A separate set of synthetic lethal gene pairs generated by De Kegel et al by identifying increases in essentiality of a gene upon loss of its paralog was also used (De Kegel et al, 2021). Protein complex membership was annotated based on data from CORUM, EBI ComplexPortal, and HuMap (complexes with confidence between 1 and 3) (Meldal et al, 2019; Drew et al, 2021; Tsitsiridis et al, 2023). Sequence identity, and family size annotations were obtained from Ensembl 93 (Zerbino et al, 2018). Mean abundances were calculated by aggregating protein abundances across all tissue types in GTEx version 8 (normalized protein abundances transformed back to absolute abundances). The data used for the analyses described in this manuscript were obtained from the GTEx Portal on 19 September 2023. Protein length annotations were obtained programmatically from the UniProt API. Protein half-life annotations were obtained from a dataset of protein turnover rates experimentally determined using SILAC labelling and mass spectrometry (Zecha et al, 2019). Protein interface annotations were obtained from a collection of 486,099 AlphaFold-predicted dimer structures,

filtered to a set of 24,090 confident (pDockQ > 0.5) interactions (Jänes et al, 2024). For all proteins that featured in at least one high-confidence model, the number of residues that were ever annotated as being in the interface of a high-confidence model was calculated. Subsequently, UniProt protein length annotations were used to calculate the percentage of residues in the protein which were ever part of the interface (UniProt Consortium, 2025). Transcription factor-gene associations were obtained from the Dorothea dataset, which collates information from existing literature, chromatin immunoprecipitation experiments, transcription factor binding sites, and expression-inferred regulatory networks (Garcia-Alonso et al, 2019). Pairs were also annotated based on whether they were compensation or collateral loss hits in the other direction, if they were testable in this direction. For the latter tests, pairs were not filtered to unique pairs, i.e. A1-A2 was a distinct observation from A2-A1. For both sets of hits (i.e. compensation and collateral loss), a series of Fishers Exact Tests were performed to check whether a given factor was significantly (FDR < 5% following Benjamini–Hochberg multiple testing correction of the Fishers Exact Test $p$-values) enriched in the set of hits. All odds ratios reported are unconditional maximum likelihood estimates calculated using the Scipy stats function fishers_exact. Dataset EV7 contains all biological variable annotations for pairs tested in the HAP1 analysis, while Dataset EV8 contains annotations for all pairs tested in the CPTAC analysis. Dataset EV9 contains odds ratios, $p$-values, and the total number of overlapping pairs included for each test.

## Assessing the enrichment of collateral loss/compensation pairs using quantitative variables

t-tests (two-tailed, assuming equal variance) were used to assess whether compensation pairs and collateral loss pairs had significantly different sequence identities, family sizes, neighbour essentialities (i.e. mean fitness defect associated with deletion of the interaction partners of the lost gene), protein–protein interaction network degree centralities, and Jaccard Indices compared to non-hit pairs (FDR < 5% following Benjamini–Hochberg multiple testing correction). Sequence identity and family size were obtained from Ensembl 93. Essentiality data was obtained from DepMap 20Q4 (CERES scores) reprocessed as outlined in previous work (De Kegel et al, 2021). Jaccard indices were calculated separately from a network representation of the BioGRID database filtered to exclude genetic interaction data and from the STRING physical subnetwork (unweighted edges). The STRING and BioGRID databases were also used to compute degree centrality, shared interactors, and total interactors. Additionally, BioGRID was filtered to direct physical interactions (identified from yeast two-hybrid experiments, protein pull down experiments, biochemical assays, protein complementation assays, and far western blotting). Phylogenetic conservation scores correspond to the number of species possessing at least one known ortholog among 1472 eukaryote species in the orthology database OrthoInspector (Eukaryota2023). Ortholog relationships for human proteins were retrieved using the OrthoInspector REST API with their Uniprot accession identifiers (Nevers et al, 2019). Ensembl version 111 was used to calculate these scores (Harrison et al, 2024). All annotations are recorded in Datasets EV7 (for pairs tested in the HAP1 analysis) and EV8 (for pairs tested in the CPTAC analysis). Cohen's d values, t-statistics, $p$-values, FDR values, and degrees of freedom (comp_vs_non_df, cl_vs_non_df) for all t-tests run are recorded in Dataset EV10.

## Calculating the predictive power of individual features

The predictive power of individual features for protein compensation and collateral loss was evaluated separately using Receiver Operating Characteristic Area Under the Curve (ROC-AUC). AUC scores were calculated by comparing individual feature values against the binary compensation outcome (1 for compensation, 0 for no compensation). Features with AUC scores below 0.5 were identified as negative predictors and inverted to standardize interpretation across all features.

## Calculating correlation between features

Relationships between features were assessed using pairwise Pearson correlation coefficients. The analysis included both continuous features and binary variables (encoded as 0/1). Correlation values were calculated across all feature pairs within the protein–protein interactions, sequence properties, and gene essentiality categories to identify potential feature redundancies and underlying biological relationships.

## Logistic regression to predict paralog compensation and collateral loss

To predict protein compensation, a logistic regression model with feature selection was developed. After excluding paralog pairs with collateral loss, features were classified as binary (0/1) or continuous. Continuous features were standardized by z-scoring. Features with AUC scores below 0.5 were identified as negative predictors and inverted to standardize interpretation. For feature selection, LASSO regularization (logistic regression with L1 penalty, C = 0.01) was employed to identify predictors with non-zero coefficients. Features with high proportions of missing values (half-life, percentage residues at interface, and transcription factor co-regulation) were excluded from the regression. Since paralog lengths were highly correlated with each other, these were aggregated into one variable (mean paralog length). With these selected features, a final logistic regression model was fit using Statsmodels. Separately, a logistic regression model with all variables and no regularizaiton was fit. ROC AUC scores for both these models were calculated to estimate the predictive power that could be achieved with the selected set of features versus the full set. For each selected predictor, odds ratios with 95% confidence intervals were calculated to quantify their effects on compensation. The same approach was also used to fit a model to predict collateral loss from the same variables, and to fit a model with all predictors chosen by the LASSO regression and synthetic lethality (defined as a pair being synthetic lethal in at least one combinatorial CRISPR screen).

## Perturb-seq results

Perturb-seq expression data was analysed to identify regulatory relationships between paralog gene pairs, i.e. cases where the knockdown of one member of the pair triggers a significant change in the mRNA levels of the other member (Data ref: Replogle et al, 2022) (Replogle et al, 2022). FDR corrected Anderson-Darling $p$-values associating genetic perturbations with mRNA expression were obtained from previous work (Replogle et al, 2022). Only paralog perturbations significantly affecting their target genes (FDR < 0.1) (when considering the most effective construct) were considered.

Expression directionality was classified using normalized bulk RNA-seq data obtained from the same source, with paralog relationships categorized as upregulation (expression increase in response to paralog knockdown) or downregulation (expression decrease in response to paralog knockdown). Fisher's Exact Tests were then used (separate tests for upregulation and downregulation with the "False" group excluding the other category) to check overlap with the sets of compensation and collateral loss hits.

## Ubiquitination sites

Data on ubiquitination sites for each protein was obtained from PhosphoSitePlus v 6.7.4 (Hornbeck et al, 2015). Proteins not present in this dataset were annotated as having 0 ubiquitination sites. two-tailed Mann–Whitney U tests were performed to assess differences between groups (shown in Fig. 6G; Appendix Fig. S9).

## Computational tools used

All data analysis and visualization was performed using Python (v3.8.10), utilizing the following libraries: NumPy (v1.24.4), Pandas (v2.0.3), Pandarallel (v1.6.5), TQDM (v4.66.4), Statsmodels (v0.14.1), SciPy (v1.10.1), NetworkX (v3.1), Matplotlib (v3.7.4), Seaborn (v0.13.2), Scikit-learn (v.1.3.2), and matplotlib-venn (v0.13). Figures were created using Inkscape v1.3.

# Data availability

Data generated in this study, i.e. the HAP1 mass spectrometry proteomics data, have been deposited to the ProteomeXchange Consortium via the PRIDE partner repository with the dataset identifier PXD055643 and 10.6019/PXD055643 (Perez-Riverol et al, 2022). All code used to run this analysis and to generate all figures and supplementary tables is available at https://github.com/cancergenetics/proteomic_paralog_compensation.

The source data of this paper are collected in the following database record: biostudies:S-SCDT-10_1038-S44320-025-00122-4.

# Peer review information

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

## Acknowledgements

This work was supported by Research Ireland (formerly Science Foundation Ireland, under grant number 20/FFP-P/8641 and 18/CRT/6214) and the Irish Research Council (2017/2018 Laureate Award). CJR received funding from the UCD Foundation through the Newman fellowship programme. SRU was funded through the School of Computer Science, University College Dublin.

## Author contributions

**Anjan Venkatesh**: Conceptualization; Data curation; Software; Formal analysis; Investigation; Visualization; Writing—original draft; Writing—review and editing. **Niall Quinn**: Investigation. **Swathi Ramachandra Upadhya**: Formal analysis. **Barbara De Kegel**: Formal analysis; Writing—review and editing. **Alfonso Bolado-Carrancio**: Investigation. **Thomas Lefeivre**: Investigation; Writing—review and editing. **Olivier Dennler**: Formal analysis. **Kieran Wynne**: Investigation. **Alexander von Kriegsheim**: Investigation. **Colm J Ryan**:

Conceptualization; Supervision; Funding acquisition; Writing—original draft; Project administration.

Source data underlying figure panels in this paper may have individual authorship assigned. Where available, figure panel/source data authorship is listed in the following database record: biostudies:S-SCDT-10_1038-S44320-025-00122-4.

## Disclosure and competing interests statement

The authors declare no competing interests.

# Expanded View Figures

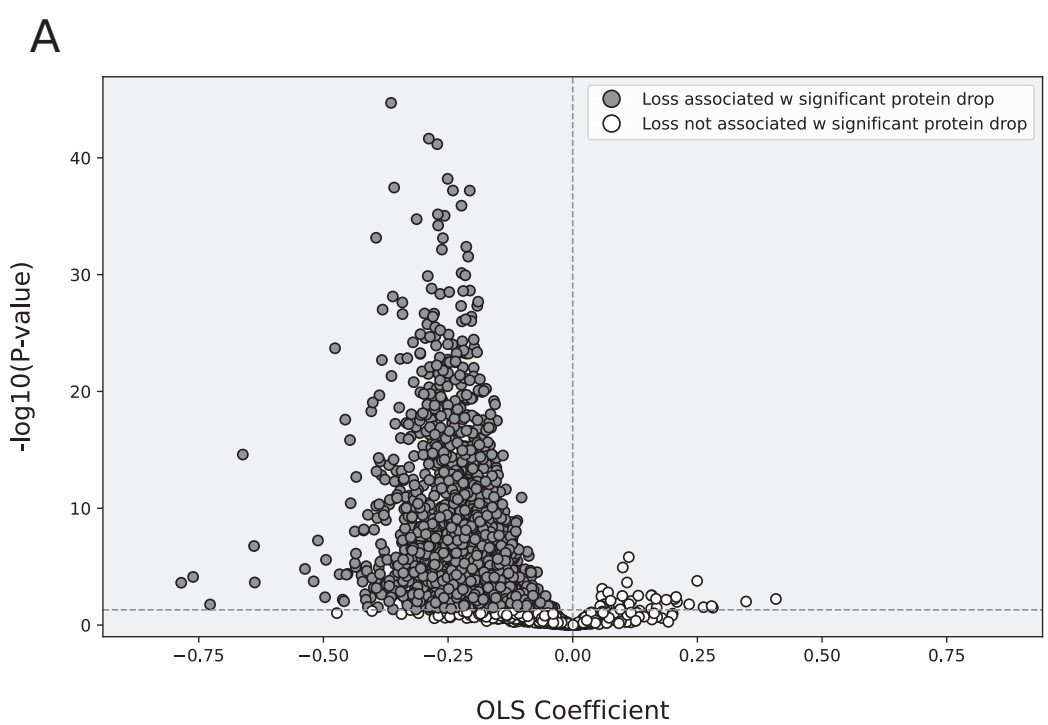

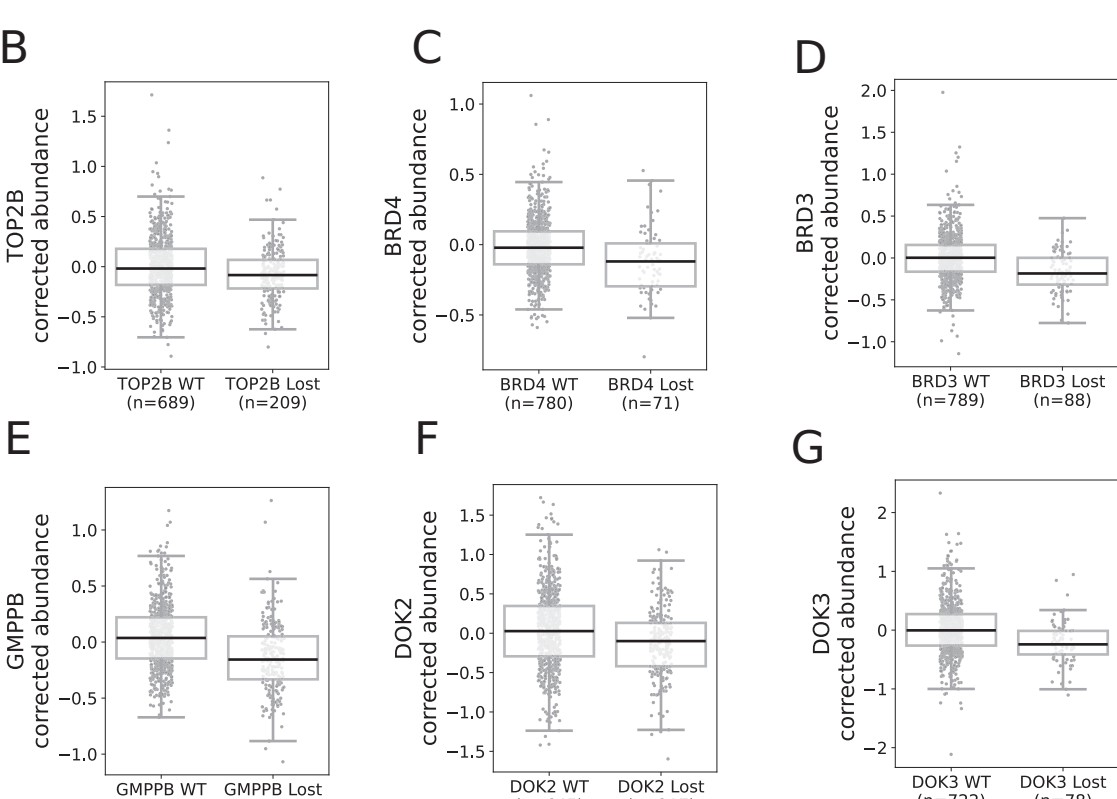

◀ **Figure EV1. Most hemizygous losses are associated with a drop in protein abundance.**

(A) Volcano plot showing decrease in protein abundance of most tested proteins following hemizygous loss (out of 3439 tests run using CPTAC data) (B–G) Box plots comparing abundance of the 'lost' paralog in samples where it is hemizygously lost versus samples retaining both copies of the gene, for all pairs shown in Fig. 2 (*TOP2B, BRD4, BRD3, GMPPB, DOK2, DOK3*). In all boxplots, the central line represents the median, box limits indicate the 25th and 75th percentiles (first and third quartiles), and whiskers extend to 1.5 × interquartile range from either end of the box. Each grey dot represents a tumour sample. Position on the y-axis indicates protein abundance adjusted for copy number of the encoding gene and cancer type, which are used as covariates in the regression model. Sample sizes for each group are shown in parentheses. Sample sizes for each group are shown in parentheses.

                                                      

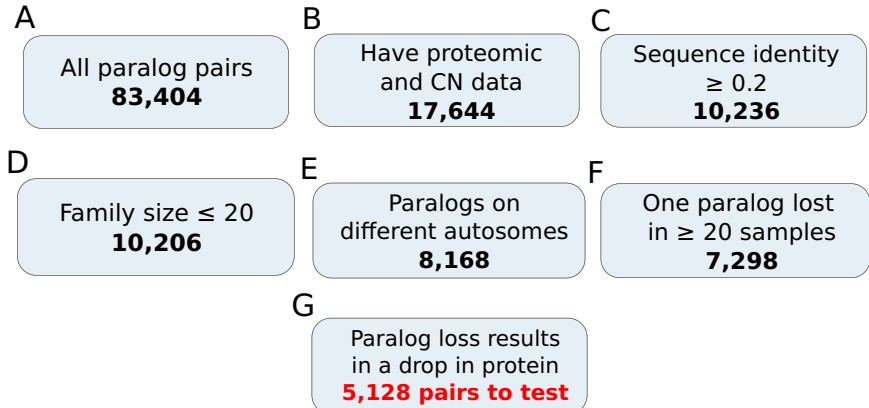

**Figure EV2. Workflow diagram showing the number of paralog pairs filtered out at each step.**

A–G. Number of paralog pairs remaining after applying each consecutive filtering step. 5128 paralog pairs are tested in the final analysis of CPTAC proteomic data.

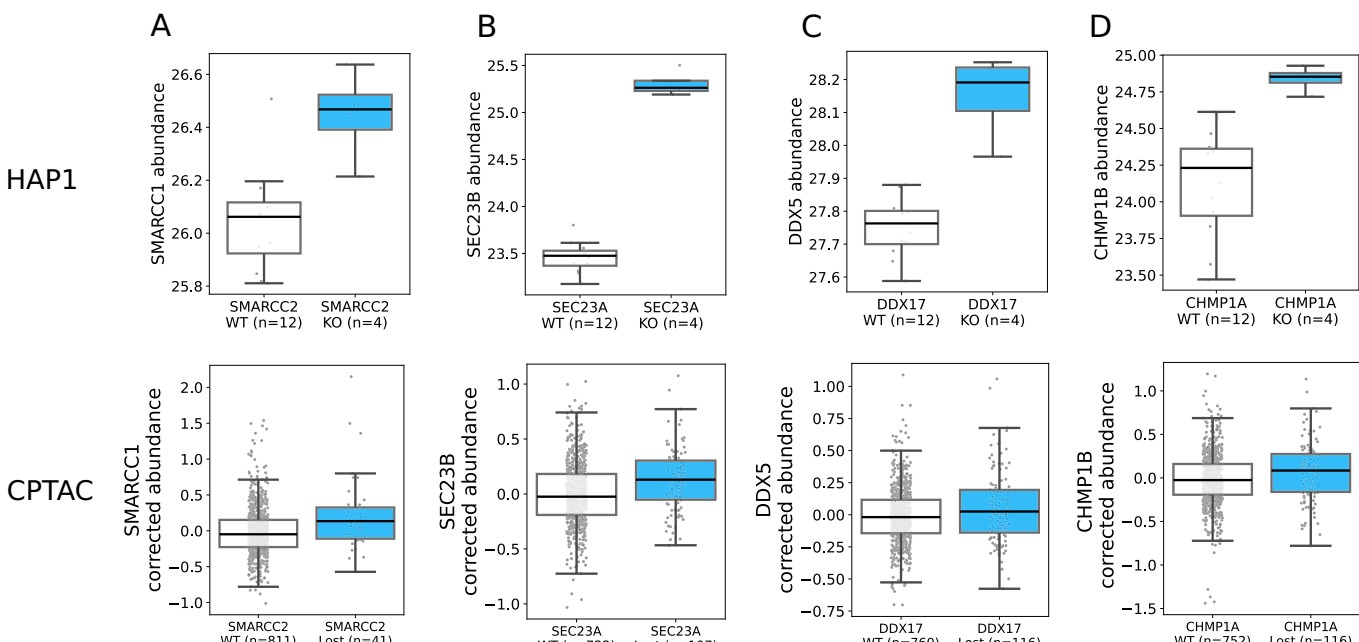

**Figure EV3.  Four paralog pairs are compensation hits both with the HAP1 cell line dataset and the CPTAC tumour sample dataset.**

(A) Box plots showing increase in SMARCC1 protein abundance associated with *SMARCC2* knockout in a HAP1 background and *SMARCC2* loss in tumour samples.
(B) Similar plots showing SEC23B abundance increase associated with *SEC23A* loss. (C) Similar plots showing DDX5 abundance increase associated with *DDX17* loss.
(D) Similar box plots showing CHMP1B abundance increase associated with *CHMP1A* loss. In all boxplots, the central line represents the median, box limits indicate the
25th and 75th percentiles (first and third quartiles), and whiskers extend to 1.5 × interquartile range from either end of the box. Each grey dot represents a cell line in the
HAP1 plots and a tumour sample in the CPTAC plots.

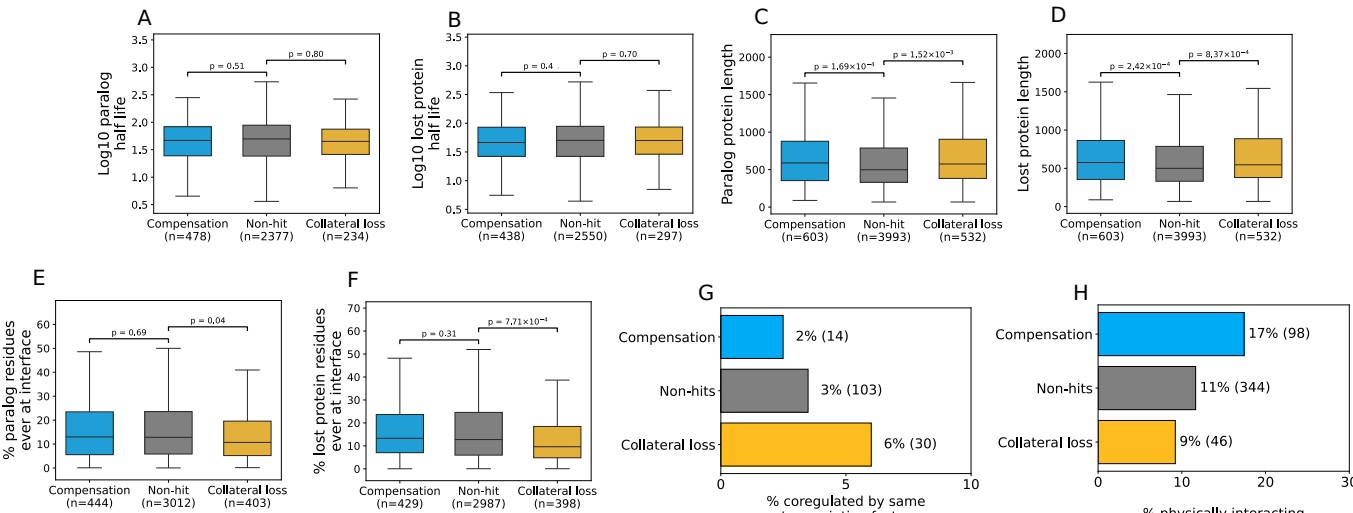

**Figure EV4. Paralogs in compensation and collateral loss pairs tend to be longer.**

Collateral loss paralogs tend to have fewer residues at protein–protein interaction interfaces and are more likely to be co-regulated by the same transcription factor. Compensation pairs are more likely to have a direct physical interaction. (**A**) Box plot showing the distributions of logged protein half-lives of paralogous proteins in the three groups. (**B**) Similar box plot showing the distributions of logged protein half-lives of the lost paralog in the three groups. (**C**) Box plot showing the distributions of protein lengths for paralogous (not lost) proteins in compensation pairs, non-hits, and collateral loss pairs. (**D**) Similar box plot showing the distributions of protein lengths for the lost paralog in the three groups. (**E**) Box plot showing the percentage of residues ever in a protein–protein interaction interface for paralogous proteins in compensation pairs, non-hits, and collateral loss pairs. (**F**) Similar box plot showing the percentage of residues ever in a protein–protein interaction interface for the lost paralogs in compensation pairs, non-hits, and collateral loss pairs. (**G**) Bar chart showing the percentage of compensation pairs, non-hits, and collateral loss pairs that are co-regulated by the same transcription factor. (**H**) Similar bar chart showing the percentage of pairs in the three groups that physically interact. In all boxplots, the central line represents the median, box limits indicate the 25th and 75th percentiles (first and third quartiles), whiskers extend to 1.5 × interquartile range from either end of the box, and outliers are not displayed. Sample sizes are shown in parentheses and all *p*-values shown correspond to two-sided equal variance t-tests.

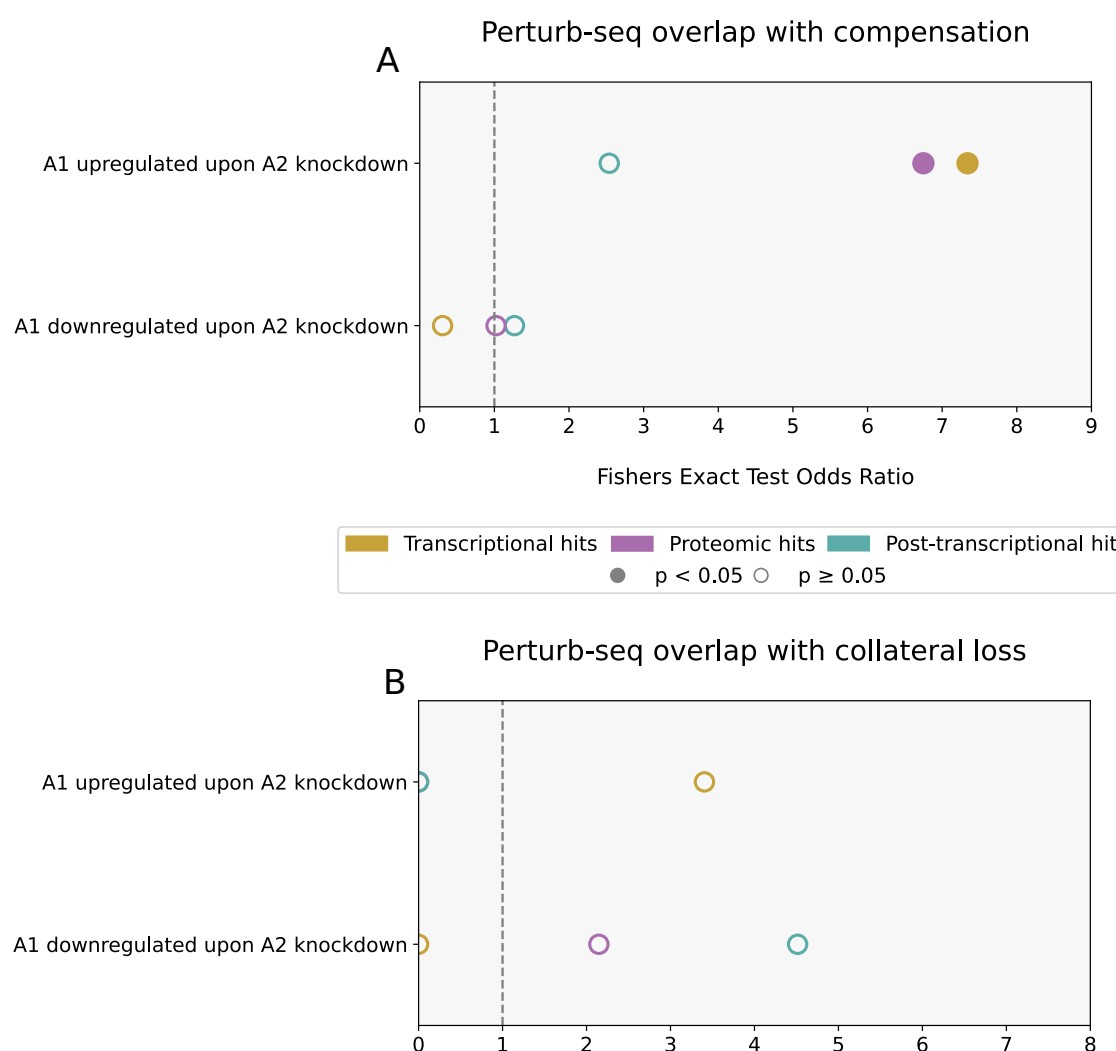

**Figure EV5. Only transcriptionally-driven compensation pairs are enriched for genes found to display upregulation upon paralog knockdown in a Perturb-seq experiment.**

(A) Dot plot showing odds ratios from Fisher's Exact tests assessing the enrichment of pairs found to be upregulated or downregulated upon paralog knockdown in a Perturb-seq experiment in three categories of pairs—transcriptional (compensation hits identified with the CPTAC transcriptomic dataset), post-transcriptional (compensation hits identified with the CPTAC protein residual dataset but not the transcriptomic dataset) and non-hits. (B) Similar dot plot showing enrichments for collateral loss hits in the three categories.

