## [Peer Review File · Molecular Systems Biology]

Proteomic compensation by paralogs preserves protein interaction networks after gene loss in cancer

Anjan Venkatesh, Niall Quinn, Swathi Ramachandra Upadhya, Barbara De Kegel, Alfonso Bolado-Carrancio, Thomas Lefevre, Olivier Dennler, Kieran Wynne, Alexander von Kriegsheim, and Colm Ryan

Corresponding author(s): Colm Ryan (colm.ryan@ucd.ie)

Review Timeline:

Submission Date:	15th Oct 24
Editorial Decision:	11th Nov 24
Revision Received:	4th Apr 25
Editorial Decision:	2nd May 25
Revision Received:	9th May 25
Accepted:	13th May 25

Editor: Jingyi Hou

Transaction Report:

11th Nov 2024

Manuscript Number: MSB-2024-12686

Title: Proteomic compensation by paralogs preserves protein interaction networks after gene loss in cancer

Author: Anjan Venkatesh

Niall Quinn

Swathi Ramachandra Upadhy

Barbara De Kegel

Alfonso Bolado-Carrancio

Thomas Lefeuvre

Olivier Dennler

Kieran Wynne

Alexander von Kriegsheim

Colm Ryan

Dear Dr Ryan,

Thank you again for submitting your work to Molecular Systems Biology. We have now received the reports from the three reviewers and as you will see below, the reviewers think that the study is potentially interesting. They raise however a series of concerns, which we would kindly ask you to convincingly address in a revision.

The reviewers' recommendations are relatively clear, so I won't reiterate them here. In particular, Reviewer #1 suggested that a deeper analysis of the mechanisms behind the reported cases of compensation and collateral loss would enhance the study's novelty and biological insights. We ask that you address this carefully, along with the other issues raised by the reviewers. As you may already know, our editorial policy allows in principle a single round of major revision, and it is therefore essential to provide responses to the reviewers' comments that are as complete as possible.

On a more editorial level, we would ask you to address the following issues:

- Please provide a .docx formatted version of the manuscript text (including legends for main figures, EV figures and tables). Please make sure that the changes are highlighted to be clearly visible.
- Please provide individual production quality figure files as .eps, .tif, .jpg (one file per figure).
- Please provide a .docx formatted letter INCLUDING the reviewers' reports and your detailed point-by-point responses to their comments. As part of the EMBO Press transparent editorial process, the point-by-point response is part of the Review Process File (RPF), which will be published alongside your paper.
- Please note that all corresponding authors are required to supply an ORCID ID for their name upon submission of a revised manuscript.
- We replaced Supplementary Information with Expanded View (EV) Figures and Tables that are collapsible/expandable online (see examples in <http://msb.embopress.org/content/11/6/812>). A maximum of 5 EV Figures can be typeset. EV Figures should be cited as 'Figure EV1, Figure EV2' etc... in the text and their respective legends should be included in the main text after the legends of regular figures.

Additional Tables/Datasets should be labeled and referred to as Table EV1, Dataset EV1, etc. Legends have to be provided in a separate tab in case of .xls files. Alternatively, the legend can be supplied as a separate text file (README) and zipped together with the Table/Dataset file.

For the figures and tables that you do NOT wish to display as Expanded View figures, they should be bundled together with their legends in a single PDF file called *Appendix*, which should start with a short Table of Content. Each legend should be below the corresponding Figure/Table in the Appendix. Appendix figures and tables should be referred to in the main text as: "Appendix Figure S1, Appendix Figure S2, Appendix Table S1" etc. See detailed instructions regarding expanded view here: <https://www.embopress.org/page/journal/17444292/authorguide#expandedview>.

- Before submitting your revision, primary datasets (and computer code, where appropriate) produced in this study need to be deposited in an appropriate public database (see <http://msb.embopress.org/authorguide-dataavailability> <https://www.embopress.org/page/journal/17444292/authorguide#dataavailability>). Please remember to provide a reviewer password if the datasets are not yet public. The accession numbers and database should be listed in a formal "Data Availability" section (placed after Materials & Method)

that follows the model below (see also <https://www.embopress.org/page/journal/17444292/authorguide#dataavailability>).

Please note that the Data Availability Section is restricted to new primary data that are part of this study.

Data availability

-At EMBO Press we ask authors to provide source data for the main figures. Our source data coordinator will contact you to discuss which figure panels we would need source data for and will also provide you with helpful tips on how to upload and organize the files.

- Our journal encourages inclusion of *data citations in the reference list* to directly cite datasets that were re-used and obtained from public databases. Data citations in the article text are distinct from normal bibliographical citations and should directly link to the database records from which the data can be accessed. In the main text, data citations are formatted as follows: "Data ref: Smith et al, 2001". In the Reference list, data citations must be labeled with "[DATASET]". A data reference must provide the database name, accession number/identifiers and a resolvable link to the landing page from which the data can be accessed at the end of the reference. Further instructions are available at .

- We updated our journal's competing interests policy in January 2022 and request authors to consider both actual and perceived competing interests. Please review the policy <https://www.embopress.org/competing-interests> and update your competing interests if necessary.

Please use the heading "Disclosure statement and competing interests".

- All Materials and Methods need to be described in the main text using our 'Structured Methods' format. According to this format, the Methods section includes a Reagents and Tools Table (listing key reagents, experimental models, software and relevant equipment and including their sources and relevant identifiers) followed by a Methods and Protocols section describing the methods, ideally using a step-by-step protocol format. The aim is to facilitate adoption of the methodologies across labs. Please download and fill our Reagents and Tools Table template (.docx), which you can find in our author guidelines: <https://www.embopress.org/page/journal/17444292/authorguide#structuredmethods>.

When submitting your revised manuscript, please do not include the Reagents and Tools Table in the Methods section of the manuscript but upload it as a separate file(.docx) choosing the file type "Reagent Table".

-Regarding data quantification:

Please ensure to specify the name of the statistical test used to generate error bars and P values, the number (n) of independent experiments (please specify technical or biological replicates) underlying each data point and the test used to calculate p-values in each figure legend. Discussion of statistical methodology can be reported in the materials and methods section, but figure legends should contain a basic description of n, P and the test applied.

Graphs must include a description of the bars and the error bars (s.d., s.e.m.).

- Please provide a "standfirst text" summarizing the study in one or two sentences (approximately 250 characters, including space), three to four "bullet points" highlighting the main findings and a "synopsis image" (550px width and 400-600 px height, PNG format) to highlight the paper on our homepage.

Here are a couple of examples:

<https://www.embopress.org/doi/10.15252/msb.20199356>

<https://www.embopress.org/doi/10.15252/msb.20209475>

<https://www.embopress.org/doi/10.15252/msb.209495>

When you resubmit your manuscript, please download our CHECKLIST (<https://www.embopress.org/pb-assets/embosite/EMBO%20Press%20Author%20Checklist-1642513524327.xlsx>) and include the completed form in your submission.

Please note that the Author Checklist will be published alongside the paper as part of the transparent process (<https://www.embopress.org/page/journal/17444292/authorguide#transparentprocess>).

If you feel you can satisfactorily deal with these points and those listed by the referees, you may wish to submit a revised version of your manuscript. Please attach a covering letter giving details of the way in which you have handled each of the points raised by the referees. A revised manuscript will be once again subject to review and you probably understand that we can give you no guarantee at this stage that the eventual outcome will be favorable.

I look forward to receiving your revised manuscript soon.

Kind regards,
Jingyi

Jingyi Hou, PhD
Scientific Editor
Molecular Systems Biology

We realize that it is difficult to revise to a specific deadline. In the interest of protecting the conceptual advance provided by the work, we recommend a revision within 3 months (9th Feb 2025). Please discuss the revision progress ahead of this time with the editor if you require more time to complete the revisions.

IMPORTANT: When you send your revision, we will require the following items:

1. the manuscript text in LaTeX, RTF or MS Word format
2. a letter with a detailed description of the changes made in response to the referees. Please specify clearly the exact places in the text (pages and paragraphs) where each change has been made in response to each specific comment given
3. three to four 'bullet points' highlighting the main findings of your study
4. a short 'blurb' text summarizing in two sentences the study (max. 250 characters)
5. a 'thumbnail image' (550px width and max 400px height, Illustrator, PowerPoint or jpeg format), which can be used as 'visual title' for the synopsis section of your paper.
6. Please include an author contributions statement after the Acknowledgements section (see <https://www.embopress.org/page/journal/17444292/authorguide>)
7. Please complete the CHECKLIST available at (<https://bit.ly/EMBOPressAuthorChecklist>). Please note that the Author Checklist will be published alongside the paper as part of the transparent process (<https://www.embopress.org/page/journal/17444292/authorguide#transparentprocess>).
8. When assembling figures, please refer to our figure preparation guideline in order to ensure proper formatting and readability in print as well as on screen:
<https://bit.ly/EMBOPressFigurePreparationGuideline>
See also figure legend guidelines: <https://www.embopress.org/page/journal/17444292/authorguide#figureformat>
9. Please note that corresponding authors are required to supply an ORCID ID for their name upon submission of a revised manuscript (EMBO Press signed a joint statement to encourage ORCID adoption). (<https://www.embopress.org/page/journal/17444292/authorguide#editorialprocess>)
Currently, our records indicate that the ORCID for your account is 0000-0003-2750-9854.

Link Not Available

11. Include a Reagents and Tools Table as part of the Methods section, which can be downloaded from our author guidelines (<https://www.embopress.org/page/journal/17444292/authorguide#structuredmethods>)

*** PLEASE NOTE *** As part of the EMBO Press transparent editorial process initiative (see our Editorial at <https://dx.doi.org/10.1038/msb.2010.72>), Molecular Systems Biology publishes online a Review Process File with each accepted manuscripts. This file will be published in conjunction with your paper and will include the anonymous referee reports, your point-by-point response and all pertinent correspondence relating to the manuscript. If you do NOT want this File to be published, please inform the editorial office at msb@embo.org within 14 days upon receipt of the present letter.

Reviewer #1:

This is an exciting paper by Venkatesh et al. The authors examine what happens at the molecular level following the inactivation of a gene with paralogous copies in the genome. One phenomenon observed in model systems such as yeast is that the remaining paralogous proteins sometimes increase in abundance (compensation) or decrease in abundance (collateral loss). They find cases for both types of impacts. The paper has two parts. One experiment was where the authors generated proteomics data on cell lines that have genes inactivated, and another was where they used publicly available transcriptome and proteome data. Once they have identified cases of compensation and collateral losses, they try to associate the properties of the genes and proteins to these cellular responses. The well-written introduction gives an excellent overview of the field and significant observations that led to this work. The work appears to be well performed, and many compensation and collateral loss cases appear robust and well supported (although none are validated using alternative approaches, which could have been interesting to see). This could be an important contribution to the field.

What I find is lacking from the manuscript is something that would bring us further than the description of cases of compensation and collateral loss, something that would bring more novelty and insight into the question, for instance, a better analysis of the mechanisms of each phenomenon (transcriptional regulation by the same TFs, protein interaction interfaces and stability). The connection between the two parts of the paper is not detailed either, and it makes us wonder why the first experimental part was not used to validate the second part. In general, the statistical analyses could also be improved and put into a single statistical model (GLM, ML, or others) to isolate the most critical factors associated with the two potential responses rather than looking at each feature individually and reporting several dozens of pairwise or tri-partite comparisons using boxplots. I provide detailed comments below to help the authors revise their manuscript.

Comments:

The title suggests a connection with cancer, but apart from the use of cancer cell lines, there is not a lot of connection with cancer

Sample sizes should be written everywhere on the plots and in statistical tests reported in the text for all tests. Effect sizes should also be mentioned for test results.

Figure 7 summarizes what we knew before this work was done rather than describing its novelty. It could be used as a panel of Figure 1 to introduce the work.

There are many confounding factors in this type of study that could be taken into account. For instance, more abundant proteins are more accessible to detect by mass spec and provide more statistical power to detect changes in abundance. They also tend to have more protein interaction partners, are more conserved, and have more post-translational modifications reported. This may create false associations if abundance is not considered as a covariate. Many effect sizes are small (even if highly significant) so many features examined could disappear in a multivariate analysis, which would facilitate interpretation and maybe help us better understand the mechanisms.

It is a bit troublesome that the nature and extent of gene inactivation in the HAP1 lines do not appear to be well known or established. If the "inactivated" gene produces a transcript, how does this impact compensation or collateral loss if it occurs at the transcriptional level?

The proteomics analysis did not specify well how the conserved peptides between paralogs were handled and how this may impact the power to detect differences between paralogs with high sequence identity.

For the analysis of the tumor proteomes, the authors interpret the results as if the higher abundance of a paralog resulted from the inactivation of the other. Could it be that since the other was more expressed for unknown reasons, the first could lose a copy without much consequence? The "compensation" would then predate gene loss and could not cause it? It may be possible to discard this possibility doing some analysis.

One additional piece of analysis that could have been done would be to look for intra-locus compensation. The authors mention that gene loss is primarily heterozygous in those tumor cells. If there is compensation among paralogs, it could come from the exact mechanisms that cause intra-locus compensation, such as overexpression of an allele of a gene when the other allele is inactive. This could be done by comparing transcript or protein levels among cell lines. Either this would happen concomitantly with paralogous compensation or would prevent compensation from happening.

Collateral loss does not receive much attention regarding mechanisms and is more surprising. Given that some paralogs showing this response appear to be forming dimers, maybe there is something special about dimer interfaces that stabilise these proteins.

It took me a long time to realize that the HAP1 cell lines were purchased and not constructed. Maybe mention it earlier in the paper. I was looking for the methods of their construction. The term biological replicates is used, but it is unclear at what step

was replication done.

It is unclear to me why paralogous genes on the identical chromosomes were excluded from the analysis. This would have been a nice dataset to dissect the mechanisms of compensation based on chromosomal proximity, etc.

Line 80: what are the differences between phenotypic and molecular compensation? Protein abundance is also a phenotype. Many be more precise?

Line 111: cite the papers from yeast here as well

Line 138: what does unbiased mean here?

Line 140: Is non-heterozygous the same as homozygous?

Line 248: the end of the paper makes it clear that compensation or duplication has not evolved to provide robustness to mutations. But, some of the writing is done in a way that may suggest it is the case. For instance, here, "be a means" seems to indicate that there is an intent.

Line 451: Maybe the authors could specify better what they mean by active compensation. Is this a nomenclature accepted upon or published before? Same for collateral loss.

Line 466: twice the same gene name?

Line 501: Suggest a similar model or support the same model.

Line 513: I would need references here. Same at line 543.

Line 565: problem with parentheses. Also, why 0 and 50mM TCEP?

Line 575: space before 200

Line 586: why remove reverse sequences?

Line 631: I assume sequence identity is for AA sequences?

Line 632: what measures?

Line 650: p-values is not always written the same way

Line 729: Biogrid is not always written the same way

Line 730: The STRING database also includes predicted PPIs based on gene features that could create false association in the analyses

Figure 1: Expected results could be shown using schematics rather than "fake" boxplots in panel b.

Figure 1C: collateral loss appears almost as frequent as compensation. It would be good to know if they co-occur in the same paralog family or if these are exclusive.

Figure 6: some legend elements are inverted, g, and f.

Reviewer #2:

Thank you for the opportunity to review "Proteomic compensation by paralogs preserves protein interaction networks after gene

loss in cancer" by Venkatesh et al. This manuscript discusses how proteins respond to their paralog deletions. Authors use HAP1 loss of functional mutant cell lines which they profile with proteomics as well as analyze CPTAC proteomic data based on samples with copy number variation in paralogs. They show that proteins that compensate for the loss of their paralog are more conserved, central in the PPI network and belong to essential protein complexes compared to those that do not compensate or depend on their paralog.

This study investigates how proteins respond to their paralog perturbation and implications for tumor biology. This addresses an important question in the field of paralog evolution and protein interactions and provides insight about the mechanisms of paralog buffering. This is an interesting study although I have a few comments.

The authors mention that analysis presented in Figure 2 used cancer type as a covariate. However, context specific paralog relations were not reported in the manuscript resulting from this analysis, but this is important since not all genes are expressed in all tissues and represent real biological paralog relationships. These data should be presented in the results and discussed. In other words if both paralogs are detectable in a given tumor type but not all tumor types, there is no biological basis for remove these data from the analysis.

Authors use the DepMap to identify cases of essentiality of paralogs. These are useful data but should also be complemented with the existing multiplexed screens such as Dede et al Genome Biology 2020, Gonatopoulos-Pournatzis et al Nature Biotech 2020 and Esmaeili Anvar et al Nat Com 2024 which specifically probed paralog pairs.

The term collateral loss is confusing and a meaning of an unrelated consequence comes to mind, which is not the case here involving the paralogous protein rather than any random protein. The community previously proposed the term to describe the loss of protein abundance of one protein in response to the deletion of its paralog as "dependency" (Diss et al Science 2017, Dandage et al bioRxiv 2023). Perhaps it would be useful use this term instead of introducing yet another term for the same phenomenon.

Minor comments

To improve readability e should be displayed as $\times 10$ to the power of exponent in the main text and all the figures.

The y-axis on figure 5c-d and 6c-e should be clearly labeled to contain the word "level." Otherwise, it is not clear what is actually shown.

Reviewer #3:

In this manuscript, Venkatesh and colleagues investigated how the copy number loss of paralog genes is tolerated in tumours and human cell lines with a focus on proteomic response. They find that the loss of one paralog can lead to either an increase, active compensation, or a decrease, collateral loss, in the protein abundance of its paralog, with active compensation being more prevalent. Further, the authors show that compensation paralog pairs, unlike collateral loss relationships, are more sequence similar, more central in protein-protein interaction network, more likely to be synthetic lethal, and involved in essential functions.

By using an integrative approach that combines genomic, transcriptomic and proteomic data, the authors shed light on underlying regulatory mechanisms of this compensation, and provide evidence that both transcriptional and post-transcriptional mechanisms are under control. By taking advantage of large cancer datasets, the authors characterise proteomic response to paralog loss in tumours and address the gap left by previous studies which have mainly focused on non-human models such as yeast. This work advances our understanding of molecular consequences of paralog loss, with potential clinical relevance for synthetic lethality approaches in cancer therapy.

Overall, this manuscript is really interesting and makes a great contribution to the field. I have a few specific questions and suggestions that I believe could help improve and strengthen the work.

Major points

1. I found the analysis in lines 208-219 somewhat difficult to follow, particularly the lines 231-215. From my understanding, the authors tested the overlap between paralog pairs showing reciprocal relationships separately for compensation pairs and collateral pairs, finding that reciprocal relationships are enriched among compensation pairs. If so, the results could be presented more clearly (e.g. indicating the number of pairs have reciprocal relationships out of 4,568 paralogs tested - line 198). I assume the total number should be the sum of all groups in the supplementary figure 3a and 3b? Also, line 214, "overlap among reciprocal compensation pairs..", could be simplified to "... among compensation pairs". Clarification of this section would improve readability.

1.a. Given that active compensation pairs can have collateral loss relationships when tested in the other direction, it would also be valuable to investigate how the characteristics (sequence similarity, degree centrality etc) differ when compensation pairs are grouped as those with symmetrical and asymmetrical relationships.

2. Based on lines 80-83, "deletion of one gene was associated with reduced protein abundance of its paralog ... due to

stoichiometric requirements of protein complexes containing both paralogs.", as well as lines 90-92, "Further analysis suggested that ... dependency are enriched among paralogs that form heteromers ...", and lines 201-206, where the authors provide evidence that they observed similar pattern like dependency - collateral loss- between a paralog pair forms heterodimer. Considering the percentage of post-transcriptionally regulated paralog pairs does not differ much between compensation and collateral loss pairs (Fig 6a) and the hypothesis in lines 402-405 that "...post-transcriptionally regulated pairs might be more likely to involve members of protein complexes ..", it would be interesting to test if this is the case also for collateral pairs (considering the loss of a paralog would leave its paralog orphan and target for degradation)? Alternatively, can the authors justify/clarify why this analysis (number of ubiquitination sites) was performed only for compensation pairs, but not for collateral loss pairs?

Minor points

1. Line 165-166, "For example, we identified between VPS4B and VPS4B" should be VPS4A and VPS4B.
2. It would be helpful to specify the cancer types used from the CPTAC dataset in the Method section. This addition would improve the transparency and reproducibility.
3. (Important but not-critical) It would be a good addition to include additional discussion on non-hit paralog pairs, particularly since, for CPTAC tumour samples, the majority of paralog pairs (based on line 199: 12% and 10% of paralog pairs are compensation and collateral loss pairs, respectively) fall under non-hit category. Some discussion on possible explanations for this observation would be helpful.

Proteomic compensation by paralogs preserves protein interaction networks after gene loss in cancer: Response to reviewers

We thank the reviewers for their helpful comments and positive feedback. We address these here and in the attached manuscript.

We note that during the revision we discovered that a sample ID matching error in our code resulted in the exclusion of 122 breast cancer samples from the analysis. This has been corrected, allowing us to test 470 additional paralog pairs, for a total of 5,128 paralog tests. As a result, most of the exact numbers quoted in the original manuscript have been modified. None of our overall conclusions are affected and some are bolstered by these new findings. We now observe a significant depletion of synthetic lethal paralogs (defined as being synthetic lethal in at least one context in a combinatorial CRISPR screen) and closest pairs in our set of collateral loss hits. Further, one of our HAP1 compensation hits, CHMP1B-CHMP1A, is now also identified as a compensation hit in the CPTAC analysis. We apologize for the errors and we have corrected all references to the old figures throughout the manuscript. All changes and additions have been highlighted in the revised manuscript.

Reviewer 1

- This is an exciting paper by Venkatesh et al. The authors examine what happens at the molecular level following the inactivation of a gene with paralogous copies in the genome. One phenomenon observed in model systems such as yeast is that the remaining paralogous proteins sometimes increase in abundance (compensation) or decrease in abundance (collateral loss). They find cases for both types of impacts. The paper has two parts. One experiment was where the authors generated proteomics data on cell lines that have genes inactivated, and another was where they used publicly available transcriptome and proteome data. Once they have identified cases of compensation and collateral losses, they try to associate the properties of the genes and proteins to these cellular responses. The well-written introduction gives an excellent overview of the field and significant observations that led to this work. The work appears to be well performed, and many compensation and collateral loss cases appear robust and well supported (although none are

validated using alternative approaches, which could have been interesting to see). This could be an important contribution to the field.

We thank the reviewer for the careful reading of our manuscript and for the positive feedback.

- What I find is lacking from the manuscript is something that would bring us further than the description of cases of compensation and collateral loss, something that would bring more novelty and insight into the question, for instance, a better analysis of the mechanisms of each phenomenon (transcriptional regulation by the same TFs, protein interaction interfaces and stability).

We note that this is the first large-scale analysis of collateral loss and compensation in humans and so this aspect in itself is novel. Even in budding yeast, factors that are predictive of compensation and collateral loss are relatively poorly characterised. DeLuna et al. (2010) found that only about 10% of paralogous proteins in yeast show significant up-regulation in response to deletion of their duplicate genes, with compensation occurring almost exclusively in duplicate pairs whose overlapping function is required for growth (DeLuna et al, 2010). They found that compensation pairs were enriched in gene pairs that have similar amino acid sequences and are transcriptionally co-regulated, but did not analyse any of the network features we have identified here (degree, shared interactions). In related work, Diss et al. (2017) demonstrated that paralog dependency, where the deletion of one gene results in its paralog losing protein-protein interactions, is as widespread as compensatory relationships, where deletion of one gene causes its paralog to gain protein-protein interactions, with physical interaction between paralogs strongly predicting dependency (Diss et al, 2017).

We note that we should have explicitly referenced the earlier finding regarding sequence identity in the text and have now done so in the text:

We hypothesised that compensatory paralogs might be more functionally similar and therefore display higher sequence identity, as has previously been shown in yeast (DeLuna et al, 2010). We found that this is indeed the case – compensation hits had higher amino

acid sequence identity than non-hits (two-tailed t-test, Cohen's $d = 0.2$, $p = 6.9 \times 10^{-4}$) (Fig. 3a) (lines 261 to 265).

We have assessed the overlap of our hits with transcriptional co-regulation (using the Dorothea database of transcription factor-gene relationships), protein interaction interfaces (using a database of AlphaFold-predicted dimers), and stability (using a database of experimentally determined protein half-lives). We found that collateral loss pairs are significantly more likely to be regulated by the same transcription factor and tend to have smaller interaction interfaces. We found no significant relationship with protein stability. We have also assessed protein length, finding both compensation and collateral loss paralogs to have longer sequences than non-hits. These results are shown in Figure EV4 and discussed in the text as follows-

Since protein abundances and interactions are influenced by protein stability, we compared experimentally determined protein half-life across these groups (Zecha et al, 2019). We found no significant differences (Figs. EV4a, EV4b; Appendix Table 10). We then obtained protein length annotations from UniProt, comparing these across the three groups. Both the compensating paralog and the lost paralog in compensation pairs tend to be longer than their counterparts in non-hit pairs (Cohen's $d = 0.2$, $p = 2.4 \times 10^{-4}$ for lost paralog, Cohen's $d = 0.2$, $p = 1.7 \times 10^{-4}$ for the other paralog) (Figs. EV4c, EV4d) (UniProt Consortium, 2025). Interestingly, this is also true of collateral loss pairs (Cohen's $d = 1.5$, $p = 8.4 \times 10^{-4}$ for lost paralog, Cohen's $d = 1.5$, $p = 1.5 \times 10^{-3}$ for the other paralog). However, two proteins of similar lengths might have a different proportion of their residues in an interaction interface. We assessed this directly using a dataset of 486,099 AlphaFold-predicted dimers, calculating, for each paralog with data available, the proportion of its residues ever present in an interaction interface according to a high-confidence model (Methods) (Jänes et al, 2024). We found that both the lost protein and its paralog in collateral loss pairs tend to have a lower proportion of their residues in interaction interfaces, in line with collateral loss paralogs generally having lower degree (Cohen's $d = -0.18$, $p = 7.7 \times 10^{-4}$ for the hemizygotously lost paralog, Cohen's $d = -0.1$, $p = 0.04$ for the collaterally lost paralog) (Figs.

EV4e, EV4f). Work in yeast has suggested that compensation is more likely to be observed for co-regulated paralog pairs (DeLuna et al, 2010). As such, we used the Dorothea database of transcription factor-gene relationships to assess co-regulation by the same transcription factor (Garcia-Alonso et al, 2019). We found that collateral loss pairs are significantly more likely to be co-regulated by the same transcription factor ($OR = 2$, $p = 7.8 \times 10^{-3}$). However, only 6% of collateral loss paralogs were found to be co-regulated (Fig. EV4g) and there was no enrichment for compensation pairs. Finally, we assessed if physical interaction between paralogs played a role in mediating these effects by filtering BioGRID to direct interactions (Methods). We found that compensation pairs were significantly more likely to share a direct physical interaction with each other ($OR = 1.6$, $p = 1 \times 10^{-3}$) (Fig. EV4h). (lines 339 to 366).

Figure EV4. Paralogs in compensation and collateral loss pairs tend to be longer. Collateral loss paralogs tend to have fewer residues at protein-protein interaction interfaces and are more likely to be co-regulated by the same transcription factor. Compensation pairs are more likely to have a direct physical interaction (a) Box plot showing the distributions of logged protein half-lives of paralogous proteins in the three groups. (b) Similar box plot showing the distributions of logged protein half-lives of the lost paralog in the three groups. (c) Box plot showing the distributions of protein lengths for paralogous (not lost) proteins in compensation pairs, non-hits, and collateral loss pairs. (d) Similar box plot showing the distributions of protein lengths of the lost paralog in the three groups. (e) Bar chart showing the percentage of paralogous proteins that share at least one residue at a protein-protein interaction interface in compensation pairs, non-hits, and collateral loss pairs. (f) Bar chart showing the percentage of lost paralogous proteins that share at least one residue at a protein-protein interaction interface in compensation pairs, non-hits, and collateral loss pairs. (g) Bar chart showing the percentage of paralogous proteins that are coregulated by the same transcription factor in compensation pairs, non-hits, and collateral loss pairs. (h) Bar chart showing the percentage of paralogous proteins that are physically interacting in compensation pairs, non-hits, and collateral loss pairs.

Similar box plot showing the distributions of protein lengths for the lost paralog in the three groups. (e) Box plot showing the percentage of residues ever in a protein-protein interaction interface for paralogous proteins in compensation pairs, non-hits, and collateral loss pairs. (f) Similar box plot showing the percentage of residues ever in a protein-protein interaction interface for the lost paralogs in compensation pairs, non-hits, and collateral loss pairs. (g) Bar chart showing the percentage of compensation pairs, non-hits, and collateral loss pairs that are co-regulated by the same transcription factor. (h) Similar bar chart showing the percentage of pairs in the three groups that physically interact. Sample sizes are shown in parentheses and all p-values shown correspond to two-sided equal variance t-tests.

- The connection between the two parts of the paper is not detailed either, and it makes us wonder why the first experimental part was not used to validate the second part.

We agree it would be nice if the order of the paper was reversed and the experimental work was used to validate the large-scale computational analysis. However, the work is presented as it was performed – the HAP1 analysis gave us the confidence that compensation could be detected simply using whole proteome profiling (in contrast to mating deletion strains to GFP tagged strains as was performed in earlier yeast studies). Our early efforts to systematically identify compensation from CPTAC data were plagued with systematic issues, which precluded them from being used to nominate candidates for study in the HAP1 model. It was only the availability of the harmonised reprocessed data that made the current analysis possible. We note that multiple HAP1 hits from our experimental work validate in the CPTAC analysis. To further validate our compensation hits, we now perform a comparison with hits identified in genome-scale Perturb-seq experiments (Replogle et al, 2022). With this, we sought to identify genes whose inhibition with CRISPR interference (CRISPRi) led to a significant increase in the mRNA abundance of their paralog. We find a significant overlap between the compensation hits identified in our analysis and the hits identified using PerturbSeq and we show that this is more pronounced for transcriptional hits than hits that are post-transcriptional only (Fig EV5).

Figure EV5. Only transcriptionally-driven compensation pairs are enriched for genes found to display upregulation upon paralog knockdown in a Perturb-seq experiment. (a) Dot plot showing odds ratios from Fisher's Exact tests assessing the enrichment of pairs found to be upregulated or downregulated upon paralog knockdown in a Perturb-seq experiment in three categories of pairs - transcriptional (compensation hits identified with the CPTAC transcriptomic dataset), post-transcriptional (compensation hits identified with the CPTAC protein residual dataset but not the transcriptomic dataset) and non-hits. (b) Similar dot plot showing enrichments for collateral loss hits in the three categories.

- In general, the statistical analyses could also be improved and put into a single statistical model (GLM, ML, or others) to isolate the most critical factors associated with the two potential responses rather than looking at each feature individually and reporting several dozens of pairwise or tri-partite comparisons using boxplots.

We thank the author for this suggestion, we believe that it has clarified our conclusions. We have now performed a new analysis where we compare all predictive features using a machine learning approach (comparing the area under the ROC curve achieved with each individual feature). We find that by far the most predictive features are those derived from protein-protein interaction networks – namely the essentiality of interaction neighbours and proportion of shared interactions (Fig. 4a).

We use a LASSO logistic regression approach to identify four features (essentiality of the lost paralog's neighbours in the BioGRID network, degree centrality of the lost paralog, family size, and Jaccard index of the pair) that achieve approximately the same predictive power as all features combined. Obviously, many of the features are highly correlated (e.g. sequence identity is correlated with shared interactors) and so we cannot state that sequence identity is not predictive of compensation, but rather we can see that it does not add anything beyond what is captured by shared interactors alone. This new analysis is presented in Figure 4 and discussed in the text as follows:

Results:

To understand the relative contributions of different features, we calculated Receiver Operating Characteristic (ROC) Area Under the Curve (AUC) values for all features (Fig. 4a). We found that neighbour essentiality had the highest predictive power (AUC 0.66), followed by BioGRID Jaccard index (AUC 0.62) and either member of the pair being in an essential protein complex (AUC 0.61) (Fig. 4a).

Many of the features we associated with proteomic compensation might be correlated with each other— for example, one might expect paralogs with greater sequence identity to share a larger subset of their interactors. We assessed the Pearson correlation between all

features, excluding features calculated for orthogonal validation, e.g. BioGRID Jaccard index was retained and STRING Jaccard index was dropped. Most pairs of features were poorly correlated, with some exceptions such as sequence identity and family size ($r = 0.4$) (Appendix Fig. S4).

We wished to assess the unique contributions of these features to paralog compensation by incorporating them into a single statistical model. Since many of our features are weakly correlated, we decided to use a LASSO regression model. LASSO models are robust to moderately correlated features and penalize the inclusion of excess features in the model. Coefficients in LASSO models can also be reduced to zero, enabling feature selection rather than just weighting. This approach enables us to identify key features that contribute uniquely to paralog compensation (Methods). Features with high proportions of missing values (half-life, proportion of residues at interface) were excluded from the regression.

After fitting this model, we were left with 4 features— family size (inverted), neighbour essentiality, BioGRID Jaccard index, and BioGRID degree centrality— which when combined, provided a comparable AUC to that of a model fit using all features (Figs. 4b, 4c). We also fit a similar model to explain collateral loss and found that two variables, degree centrality and essentiality of the lost paralog, were sufficient to achieve an AUC similar to that of a model with all variables included (Appendix Fig. S5). Interestingly, collateral loss AUCs for these features were generally anti-correlated with compensation AUCs, with a Pearson correlation coefficient of -0.83 . This is in line with compensation and collateral loss being mutually exclusive phenomena within paralog families.

(lines 384 to 415).

Figure 4. Neighbour essentiality, Jaccard index, degree centrality, and family size are sufficient to predict paralog compensation. (a). Receiver Operating Characteristic (ROC) Area Under the Curve (AUC) values for individual predictors of proteomic compensation. Features with negative predictive value are inverted, i.e. multiplied by -1. **(b).** Odds ratios and error bars for variables which were retained in a LASSO regression fit to predict proteomic compensation using the features in (a) (Methods). **(c)** ROC curve comparing the predictive power of the full model (logistic regression model containing all features) with the predictive power of the “selected features model” containing only the four features shown in (b).

Appendix Figure S4. Most variables predictive of proteomic compensation are poorly correlated. Heatmap showing all-by-all Pearson correlations for variables assessed for the task of predicting proteomic compensation. Pearson correlation coefficients are displayed for each pairwise relationship.

Appendix Figure S5. Collateral loss is predicted by degree centrality and neighbour essentiality. (a) Barchart showing ROC AUC values for 12 variables for predicting collateral loss. **(b)** Dot plot showing odds ratios with error bars for individual variables in a LASSO regression model fit to predict collateral loss from the variables in (a).

Discussion:

Taken together, our results suggest a model whereby compensatory relationships stabilise essential protein-protein interaction subnetworks or complexes in the face of genetic perturbation in cancer. A LASSO regression model fit to predict proteomic compensation selects three interaction-based features (degree centrality, neighbour essentiality, and Jaccard index) and only one sequence-based feature (family size) from the full feature set, suggesting that the proteomic compensation effect is largely mediated by gene loss-induced changes to the protein-protein interaction network. Following loss of one gene, the protein abundance of a highly sequence similar paralog is increased, and this protein takes the

place of the lost gene in the protein-protein interaction network. Cancer cells then become dependent on this protein for survival (Fig. 7). Previous work has identified individual examples consistent with this model, for example loss of the cohesin subunit STAG2 results in increased protein abundance of its paralog STAG1 and an increased sensitivity to genetic perturbation of STAG1; our results suggest that this may be a relatively common phenomenon (van der Lelij et al, 2017; Adane et al, 2021). (lines 626 to 639).

Figure 7. Proposed model for protein-protein interaction network rewiring leading to proteomic compensation and synthetic lethality. Our results are consistent with a model where, for pairs with post-transcriptional compensation effects, increased incorporation of A1 into A2’s essential protein interactions following A2 loss results in an increase of its abundance through reduced degradation of A1 subunits. A1 then becomes essential in

backgrounds where A2 is lost, i.e. the pairs are synthetic lethal.

- The title suggests a connection with cancer, but apart from the use of cancer cell lines, there is not a lot of connection with cancer

We are only studying genetic alterations in cancer cell lines and tumour profiles and felt it was more reasonable for the title to reflect this limitation. We think it is likely that compensatory relationships will occur outside of this context, but we cannot study it systematically as of yet due to the lack of proteomic profiles of non-tumour samples with gene deletions.

- Sample sizes should be written everywhere on the plots and in statistical tests reported in the text for all tests. Effect sizes should also be mentioned for test results.

We have annotated all plots with sample size information. We also report effect size for all tests- odds ratios for Fisher's Exact Tests (which we have retained from the first version of the manuscript), Cohen's d for t-test comparisons, and Cliff's delta for Mann-Whitney U-test comparisons. This information is also included in Appendix Table 10.

- Figure 7 summarizes what we knew before this work was done rather than describing its novelty. It could be used as a panel of Figure 1 to introduce the work.

We have modified Figure 7 to make it clear that it illustrates cases of paralog compensation in essential protein complexes, i.e. cases where the inability of the other paralog in the pair to compensate for the lost paralog would result in cell death. This illustrates the link between proteomic compensation and synthetic lethality, i.e. functional compensation, that our results suggest.

- There are many confounding factors in this type of study that could be taken into account. For instance, more abundant proteins are more accessible to detect by mass spec and provide more statistical power to detect changes in abundance. They also tend to have more protein interaction partners, are more conserved, and have

more post-translational modifications reported. This may create false associations if abundance is not considered as a covariate. Many effect sizes are small (even if highly significant) so many features examined could disappear in a multivariate analysis, which would facilitate interpretation and maybe help us better understand the mechanisms.

We have analyzed protein abundance of the lost paralog and the other member of the pair and interestingly, we note that collateral loss paralogs have significantly higher mean abundance in the GTEx resource, while the lost paralogs of compensation pairs have significantly lower abundances, albeit this is a small effect size. We have added multivariate models to predict compensation and collateral loss, as discussed earlier. We have included a discussion of protein abundance in the text as follows-

Degree centrality could potentially be confounded by protein abundance— more abundant proteins might be more amenable to detection, resulting in such proteins having a larger number of known interactors. However, this was not the case- when comparing mean abundances of lost paralogs in the GTEx dataset, deleted genes in compensation pairs had significantly lower mean abundance than non-hits (Cohen's $d = -0.1$, $p = 0.03$) (Appendix Fig. S2b). Conversely, both the deleted gene and the paralog with reduced abundance in collateral loss pairs had significantly higher mean abundance (Cohen's $d = 0.1$, $p = 2.5 \times 10^{-3}$ for lost paralogs and Cohen's $d = 0.3$, $p = 9 \times 10^{-9}$ for the other paralog) (Appendix Fig. S2c).

(lines 287 to 295).

Appendix Figure S2. Compensation pairs are more central in the protein-protein interaction network when analyzing the STRING physical subnetwork, EBI ComplexPortal, and the HuMap database. (a) Box plot showing distributions of STRING degree centrality for compensation pairs, non-hits, and collateral loss pairs (Methods). **(b)** Box plot showing the mean abundance of the paralogue (not-lost) protein in the GTEx dataset of proteomic profiles from healthy tissues for all three groups. **(c)** Similar box plot showing the mean abundance of the lost protein in the GTEx dataset of proteomic profiles from healthy tissues for all three groups. **(d)** Similar box plot comparing the distributions of Jaccard indices calculated with the STRING network. **(e)** Bar chart showing the proportion of paralog pairs in each group where at least one paralog is a CORUM protein complex member (Methods). **(f)** Bar chart comparing the percentage of pairs in each set where either member of the pair is a member of a protein complex in EBI Complex Portal (Methods). **(g)** Similar bar chart plotted to show the percentages of pairs where either member is in a predicted HuMap complex (Methods). **(h)** Boxplots showing the conservation scores of paralog pairs in all three groups, using the presence of a known ortholog in other species as a proxy for conservation across evolutionary time (Methods). All p-values are from two-tailed t-tests (assuming equal variances). Counts and sample sizes for each group are shown in parentheses.

- It is a bit troublesome that the nature and extent of gene inactivation in the HAP1 lines do not appear to be well known or established. If the "inactivated" gene produces a transcript, how does this impact compensation or collateral loss if it occurs at the transcriptional level?

We have added a discussion of this as follows:

Previous analysis has shown that many HAP1 'knockouts' display residual expression of the target transcripts, even in the absence of protein expression, potentially due to variable efficiency of nonsense-mediated decay (Smits et al, 2019). In our analysis we have only analysed HAP1 clones where we could validate a reduction in protein abundance of the target gene. We have not assayed mRNA abundance and so cannot rule out the possibility that residual expression of target transcripts may influence the observed compensation and collateral loss patterns. (lines 716 to 724).

- The proteomics analysis did not specify well how the conserved peptides between paralogs were handled and how this may impact the power to detect differences between paralogs with high sequence identity.

We have added a discussion of this as follows:

Furthermore, a challenge with bottom-up proteomics is that protein abundance is quantified from the abundance of peptides which may match multiple proteins, which is especially common for paralogs. The CPTAC PanCan dataset was re-processed by the authors using FragPipe and the Philosopher pipeline, with razor peptides being assigned to a single protein based on the overall peptide evidence for that protein (Li et al, 2023). In our analysis of HAP1 proteomics we quantified all protein abundances using unique peptides only (those that map to only a single protein), which meant that we could not analyse the pair ASF1A-ASF1B because only shared peptides could be identified. Since peptide-level data is not

readily available for the CPTAC dataset, it was not possible to exclude shared peptides. Although some collateral loss hits might reflect artefacts resulting from shared peptides, compensation hits, where the abundances of two paralogs are inversely correlated, are less likely to be affected. Furthermore, approximately half of the compensation and collateral loss hits are also evident at the transcriptomic level and these are unlikely to result from shared peptide artefacts. (lines 726 to 739).

- For the analysis of the tumor proteomes, the authors interpret the results as if the higher abundance of a paralog resulted from the inactivation of the other. Could it be that since the other was more expressed for unknown reasons, the first could lose a copy without much consequence? The "compensation" would then predate gene loss and could not cause it? It may be possible to discard this possibility doing some analysis.

This is a very interesting idea and not one we had considered. In our model, we adjust for the copy number status of the compensating gene and so the higher expression would presumably be a result of an overall epigenetic state of the cell rather than a genetic change to the compensating gene. We do not think we could discard this possibility without doing careful single-cell analysis before and after genetic perturbation similar to what has been performed to dissect cell populations that confer resistance to drugs (Prieto-Vila et al, 2019; Emert et al, 2021; Jacobo Jacobo et al, 2024). The single-cell proteomics data, in the presence of genetic perturbations, that would be required to address does not yet exist. We have added a discussion of this possibility as follows:

Although we identify a number of associations between gene loss and changes in paralog abundance, it might not be possible to demonstrate a causal link for all these cases. Our assumption is that loss of one gene causes an increase in the abundance of another paralog. However, it is possible that, at least in some cases, causality flows the other direction and it is an increased abundance of a paralog that allows the copy loss to occur. Indeed in the case of cancer drug resistance, often resistant states exist in the cell population prior to drug treatment (Gerlinger et al, 2012; Prieto-Vila et al, 2019; Jacobo

Jacobo et al, 2024; Emert et al, 2021). We cannot rule out the possibility that some of our ‘compensatory’ relationships reflect this alternative framing. However, the significant overlap between our compensation hits and those from Perturb-seq experiments would argue that our hits are enriched for real regulatory relationships, but some may reflect the case where higher-expression predates the genetic event. (lines 741 to 752).

- One additional piece of analysis that could have been done would be to look for intra-locus compensation. The authors mention that gene loss is primarily heterozygous in those tumor cells. If there is compensation among paralogs, it could come from the exact mechanisms that cause intra-locus compensation, such as overexpression of an allele of a gene when the other allele is inactive. This could be done by comparing transcript or protein levels among cell lines. Either this would happen concomitantly with paralogous compensation or would prevent compensation from happening.

We have not set up our analysis pipeline to analyse intra-locus compensation. We specifically focus on events where the genetic perturbation (copy loss) of one gene is associated with a reduction in the abundance of the encoded protein. This is necessary because of the noisiness of the samples and profiles (imperfect copy number calls, impure samples). We have added a discussion of this possibility as follows:

Our analysis also was entirely focussed on detecting cases of paralog compensation caused by single copy loss in cancer. Another possible scenario is compensation by increased expression of the remaining allele— although we find that a majority of single copy loss events result in a significant drop in protein abundance, a systematic analysis of intra-locus compensation is an interesting possibility for future work. (lines 776 to 780).

- Collateral loss does not receive much attention regarding mechanisms and is more surprising. Given that some paralogs showing this response appear to be forming dimers, maybe there is something special about dimer interfaces that stabilise these proteins.

We have examined interface size and direct physical interaction (defined as interactions in BioGRID identified using yeast two-hybrid screens, pulldown experiments, biochemical assays, protein complementation assays, or far western blotting) as predictors of both compensation and collateral loss (Fig. EV6). Paralogs in collateral loss pairs tend to have a lower proportion of their residues at interaction interfaces, which is in line with their lower degree in interaction networks.

Compensation pairs are significantly more likely to heterodimerize, with no significant relationship observed for collateral loss pairs. We have also examined transcriptional co-regulation and found that collateral loss pairs are significantly more likely to be co-regulated by the same transcription factor than non-hits, although this is only true of 6% of pairs and so likely does not explain the majority of cases. We have discussed these results in the text as follows-

Since protein abundances and interactions are influenced by protein stability, we compared experimentally determined protein half-life across these groups (Zecha et al, 2019). We found no significant differences (Figs. EV4a, EV4b; Appendix Table 10). We then obtained protein length annotations from UniProt, comparing these across the three groups. Both the compensating paralog and the lost paralog in compensation pairs tend to be longer than their counterparts in non-hit pairs (Cohen's $d = 0.2$, $p = 2.4 \times 10^{-4}$ for lost paralog, Cohen's $d = 0.2$, $p = 1.7 \times 10^{-4}$ for the other paralog) (Figs. EV4c, EV4d) (UniProt Consortium, 2025). Interestingly, this is also true of collateral loss pairs (Cohen's $d = 1.5$, $p = 8.4 \times 10^{-4}$ for lost paralog, Cohen's $d = 1.5$, $p = 1.5 \times 10^{-3}$ for the other paralog). However, two proteins of similar lengths might have a different proportion of their residues in an interaction interface. We assessed this directly using a dataset of 486,099 AlphaFold-predicted dimers, calculating, for each paralog with data available, the proportion of its residues ever present in an interaction interface according to a high-confidence model (Methods) (Jänes et al, 2024). We found that both the lost protein and its paralog in collateral loss pairs tend to have a lower proportion of their residues in interaction interfaces, in line with collateral loss paralogs generally having lower degree (Cohen's $d = -0.18$, $p = 7.7 \times 10^{-4}$ for the hemizygotously lost paralog, Cohen's $d = -0.1$, $p = 0.04$ for the collaterally lost paralog) (Figs.

EV4e, EV4f). Work in yeast has suggested that compensation is more likely to be observed for co-regulated paralog pairs (DeLuna et al, 2010). As such, we used the Dorothea database of transcription factor-gene relationships to assess co-regulation by the same transcription factor (Garcia-Alonso et al, 2019). We found that collateral loss pairs are significantly more likely to be co-regulated by the same transcription factor (OR = 2, $p = 7.8 \times 10^{-3}$). However, only 6% of collateral loss paralogs were found to be co-regulated (Fig. EV4g) and there was no enrichment for compensation pairs. Finally, we assessed if physical interaction between paralogs played a role in mediating these effects by filtering BioGRID to direct interactions (Methods). We found that compensation pairs were significantly more likely to share a direct physical interaction with each other (OR = 1.6, $p = 1 \times 10^{-3}$) (Fig. EV4h). **(lines 339 to 366).**

We have also amended the relevant section of the discussion-

We do not have a simple model for collateral loss pairs. The simplest model, supported by systematic work in yeast, would be that pairs of paralogs often form heterodimers and hence stabilise each other (Diss et al, 2017; Dandage & Landry, 2019). When one paralog is lost, the other loses stability and is then degraded. The strongest effect we see, GMPPA/GMPPB, corresponds to a pair known to form a heterodimer, consistent with this model. Since there is considerable overlap between collateral loss hits identified at the transcript level and those identified at the protein level, it is also likely that many of these relationships involve some transcriptional co-regulation. We found that collateral loss pairs are more likely to be transcriptionally co-regulated than non-hits, although this only applies for 6% of pairs and so likely does not explain the majority of effects we observe. However, we do not find direct physical interaction to have substantial predictive power for collateral loss. As such, this too is unlikely to explain all collateral loss pairs and so additional explanations are needed. **(lines 685 to 697).**

- It took me a long time to realize that the HAP1 cell lines were purchased and not constructed. Maybe mention it earlier in the paper. I was looking for the methods of their construction. The term biological replicates is used, but it is unclear at what step was replication done.

We apologize for the confusion and have clarified this in the text as follows. We note that approximately half of the cell lines used were created specifically for this study, i.e. not previously available in the Horizon catalogue.

We obtained knockout HAP1 cell lines from Horizon discovery and performed whole-cell lysate proteomic profiling of 34 HAP1-derived clones, each of which had a frameshift insertion or deletion of a single paralog, and the parental HAP1 strain for comparison (Fig. 1a, Methods). Each clone was grown in four separate cultures, constituting four biological replicates. (lines 166 to 170).

- It is unclear to me why paralogous genes on the identical chromosomes were excluded from the analysis. This would have been a nice dataset to dissect the mechanisms of compensation based on chromosomal proximity, etc.

This is a limitation of our approach that is now discussed in the discussion on lines 679-685. Two genes which are chromosomally proximal will typically be deleted / amplified together and so it is difficult to assess how copy number changes of one will impact the abundance of another. However, the HAP1 analysis lets us test two pairs, DDX42-DDX5 and KATNAL2-VPS4B, which are on the same chromosome. Neither of these pairs are hits.

Our analysis focussed on copy number loss events, which often affect multiple genes. As such, to avoid identifying spurious collateral loss pairs which are actually explained by the same deletion event, we excluded 1,400 paralog pairs which met all other filtering criteria, but were on the same chromosome. However, since we generated HAP1 knockouts using CRISPR-Cas9, this was not a concern with the HAP1 analysis. We tested two pairs,

KATNAL1-VPS4B and DDX42-DDX5 which are on the same chromosome, and did not identify a significant effect for either pair (lines 754 to 760).

- Line 80: what are the differences between phenotypic and molecular compensation? Protein abundance is also a phenotype. Many be more precise?

We have clarified our use of this terminology in the text as follows-

This 'need-based upregulation' means that the compensatory response only occurs in environmental conditions where the function of the gene is required for optimal growth or survival, and disappears in conditions where the function becomes dispensable. Intriguingly, the pairs that exhibited this upregulation were enriched for pairs known to be synthetic lethal, suggesting a relationship between phenotypic compensation (i.e. cases where the presence of a paralog enables cells to survive the deletion of an essential gene) and molecular compensation (i.e. cases where the paralogous protein is upregulated in the context of essential gene loss) (lines 80 to 87).

- Line 111: cite the papers from yeast here as well

We have added these citations to the text.

- Line 138: what does unbiased mean here?

Here, we used unbiased to mean that the whole proteome was quantified without the use of antibodies targeting specific proteins. We have removed 'unbiased' from this sentence as it might be redundant and confusing.

- Line 140: Is non-heterozygous the same as homozygous?

We have replaced this phrase with “homozygous” for clarity.

- Line 248: the end of the paper makes it clear that compensation or duplication has not evolved to provide robustness to mutations. But, some of the writing is done in a way that may suggest it is the case. For instance, here, "be a means" seems to indicate that there is an intent.

We have modified the sentence as shown below to avoid any assertions about the evolution of the compensation mechanism.

We hypothesised that proteomic compensation might provide a mechanism to stabilise the protein-protein interaction network against variation in protein abundance (either due to stochastic variation or external perturbations) (lines 274 to 276).

- Line 451: Maybe the authors could specify better what they mean by active compensation. Is this a nomenclature accepted upon or published before? Same for collateral loss.

We have replaced references to ‘active compensation’ with ‘proteomic compensation’ for clarity. We have also included a discussion of the existing terminology in the field in the introduction as follows-

We note that there has been a diversity of terms used in the literature to refer to cases where the abundance of one paralog is increased in response to the deletion of another – need-based upregulation, transcriptional reprogramming, active compensation – and here we simply use the term ‘proteomic compensation’ (Kafri et al, 2005; DeLuna et al, 2010; Diss et al, 2014). The opposite effect, where the abundance of one paralog is decreased in response to deletion of another has also been observed (Diss et al, 2017; Dandage & Landry, 2019), and reported using distinct terms, including negative responsiveness, and

dependency. We here use the term collateral loss to indicate that a reduction in the protein abundance of one paralog happens in response to the genetic perturbation of another (i.e. the protein is not itself the target of the genetic event, but nonetheless it displays decreased abundance) (lines 99 to 109).

- Line 466: twice the same gene name?

The first use of STAG1 here refers to the protein, i.e. we see an increase in the protein abundance of STAG1 upon STAG2 loss. STAG1 then refers to the gene itself, i.e. we also see an increase in the genetic dependency upon this gene upon STAG2 loss, measured as the fitness defect associated with STAG1 perturbation.

- Line 501: Suggest a similar model or support the same model.

We have edited this sentence for clarity- *Our finding that paralog compensation pairs, especially those that are post-transcriptionally regulated, are enriched in protein complex subunits and more highly ubiquitinated is in line with this model (lines 674 to 676).*

- Line 513: I would need references here. Same at line 543.

We have added references to these lines in the text (lines 656, 672).

- Line 565: problem with parentheses. Also, why 0 and 50mM TCEP?

We have fixed the parentheses in this sentence. This was an error and we have amended the text (1.5 mg/ml TCEP was used).

- Line 575: space before 200

We have amended the text.

- Line 586: why remove reverse sequences?

Reverse sequences are "decoy" sequences used to estimate the False Discovery Rate and as such their removal is a routine step in proteomic processing.

- Line 631: I assume sequence identity is for AA sequences?

We have amended the text to clarify that we are referring to amino acid sequence identity here.

- Line 632: what measures?

We have amended the text to clarify that we are referring to measurements of protein abundance in the CPTAC dataset here.

- Line 650: p-values is not always written the same way

We have amended the text.

- Line 729: Biogrid is not always written the same way

We have amended the text.

- Line 730: The STRING database also includes predicted PPIs based on gene features that could create false association in the analyses

We believe this is mitigated by our use of the STRING physical sub-network, which does not include functional associations between proteins. Further, we show the same results with the BioGRID dataset, which relies solely on experimental evidence.

- Figure 1: Expected results could be shown using schematics rather than "fake" boxplots in panel b.

We have replaced the boxplots in Figures 1 and 2 with cartoon boxplots to make it clear that these do not reflect real data.

- Figure 1C: collateral loss appears almost as frequent as compensation. It would be good to know if they co-occur in the same paralog family or if these are exclusive.

We have replaced the boxplots in Figures 1 and 2 with cartoon boxplots to make it clear that these do not reflect real data.

- Figure 6: some legend elements are inverted, g, and f.

We have amended the figure legend.

Reviewer 2

- Thank you for the opportunity to review "Proteomic compensation by paralogs preserves protein interaction networks after gene loss in cancer" by Venkatesh et al. This manuscript discusses how proteins respond to their paralog deletions. Authors use HAP1 loss of functional mutant cell lines which they profile with proteomics as well as analyze CPTAC proteomic data based on samples with copy number variation in paralogs. They show that proteins that compensate for the loss of their paralog are more conserved, central in the PPI network and belong to essential protein complexes compared to those that do not compensate or depend on their paralog.

This study investigates how proteins respond to their paralog perturbation and implications for tumor biology. This addresses an important question in the field of paralog evolution and protein interactions and provides insight about the mechanisms of paralog buffering. This is an interesting study although I have a few comments.

We thank the reviewer for the careful reading of our manuscript and for the positive feedback.

- The authors mention that analysis presented in Figure 2 used cancer type as a covariate. However, context specific paralog relations were not reported in the manuscript resulting from this analysis, but this is important since not all genes are expressed in all tissues and represent real biological paralog relationships. These data should be presented in the results and discussed. In other words if both paralogs are detectable in a given tumor type but not all tumor types, there is no biological basis for remove these data from the analysis.

We agree with the reviewer that context-specific relationships between paralogs are of significant interest and we are exploring them in other work (e.g. context-specific synthetic lethality between paralogs). The reason we have not included them in this analysis is statistical rather than biological. With the full pan-cancer CPTAC dataset we have 1023 samples, while with individual cancer types, we have only 108 samples on average (median). This means that we test far fewer pairs when analysing individual cancer types (median 535). The discordance in hits across cancer types (e.g. compensation observed in one but not another) is likely primarily attributable to differences in our ability to detect them rather than cancer-type specific biology. We note that when we aggregate the ‘cancer type-specific’ hits across all cancer types, for a total of 5,474 pairs tested (with 4%, i.e. 219 compensation hits and 6% i.e. 332 collateral loss hits), we observe similar patterns to the ones we describe in the manuscript. Compensation pairs are enriched for neighbour essentiality (Response Fig. 1a), degree centrality (1b), Jaccard index (1c), essential CORUM complex membership (1d), synthetic lethality (1e), physical interaction (1f), and paralog length (1g, 1h). Collateral loss pairs have significantly lower neighbour essentiality (1a), are less likely to be closest pairs (1i), have a family size of 2 (1j), and have significantly lower conservation scores (1k) and higher lost paralog abundance in GTEx (1l). We have not included this analysis in the paper as we do not feel it adds significantly to the results, but present the relevant figures below:

Response Figure 1 (not included in revised text): Compensation and collateral loss hits detected using data from individual CPTAC studies show similar patterns to hits detected using the aggregated CPTAC dataset. (a) Box plot showing the distribution of DepMap CERES scores for the deleted gene in paralog pairs that are compensation hits in at least one CPTAC study, pairs that are never hits in any study, and pairs that are collateral loss hits in at least one study. **(b)** Similar box plot showing the distribution of BioGRID degree centrality values for the deleted gene in paralog pairs belonging to the three groups. **(c)** Similar box plot showing the distribution of BioGRID Jaccard indices for the three groups. **(d)** Bar chart showing the percentage of pairs in each group that are members of a CORUM complex with at least one broadly essential member. **(e)** Bar chart showing the percentage of pairs in each group found to be synthetic lethal in at least two contexts in multiplexed (combinatorial) CRISPR screens. **(f)** Similar bar chart showing the percentage of pairs in each group that directly physically interact with each other. **(g)** Box plot showing the distribution of lost paralog lengths for the three groups. **(h)** Similar box plot showing the distribution of lengths for the other (not lost) paralog for the three groups. **(i)** Bar chart showing the percentage of pairs in each group that are each other's closest paralog. **(j)** Bar chart showing the percentage of pairs in each of the three groups that belong to a family with size 2, i.e. no other paralogs in the family. **(k)** Box plot showing the distribution of conservation scores for paralog pairs in the three groups. **(l)** Box plot showing the

distributions of mean GTEx protein abundances for the lost paralog, for pairs in each of the three groups. Sample sizes are shown in parentheses and all p-values are from two-sided t-tests assuming equal variance.

We have added a paragraph to the discussion as follows:

Combinatorial CRISPR screens have revealed that many paralog pairs are synthetic lethal in a context-specific fashion, with some pairs synthetic lethal in one cell line but not others (Thompson et al, 2021; Ito et al, 2021; Parrish et al, 2021). Work in yeast has also established that at least some proteomic compensation relationships are only evident in specific contexts where such compensation is necessary for survival, e.g. for glucose metabolism in environments where glucose is the primary carbon source (DeLuna et al, 2010). In order to gain statistical power, we have here focussed on compensatory relationships that can be observed across samples from all cancer types combined. It is still unclear to what extent there may also be compensatory relationships evident within specific contexts or cancer types. This may be possible to address as the number of samples available for individual cancer types increases (lines 764 to 774).

- Authors use the DepMap to identify cases of essentiality of paralogs. These are useful data but should also be complemented with the existing multiplexed screens such as Dede et al Genome Biology 2020, Gonatopoulos-Pournatzis et al Nature Biotech 2020 and Esmaeili Anvar et al Nat Com 2024 which specifically probed paralog pairs.

We note that in our original submission we assessed this enrichment using both the DepMap synthetic lethal dataset and additional datasets derived from combinatorial screens and found that it was significant in all cases. This may not have been clear and so we have amended the text to state ‘multiplexed combinatorial CRISPR screens’ on line 413. For this analysis of combinatorial screens we assembled a consensus dataset based on the results of five multiplexed screens- Thompson et al 2021, Parrish et al 2021, Gonatopolous-Pournatzis et al 2020, Dede et al 2020, and Ito

et al 2021, and used this to assess the overlap between compensation pairs and gene pairs that are synthetic lethal in at least one (Appendix Figure S6) or two (Figure 5B) contexts in these screens.

- The term collateral loss is confusing and a meaning of an unrelated consequence comes to mind, which is not the case here involving the paralogous protein rather than any random protein. The community previously proposed the term to describe the loss of protein abundance of one protein in response to the deletion of its paralog as "dependency" (Diss et al Science 2017, Dandage et al bioRxiv 2023). Perhaps it would be useful use this term instead of introducing yet another term for the same phenomenon.

We struggled with terminology as related phenomena have been discussed using distinct terminology (e.g. need based upregulation and active compensation) while distinct phenomena have also been discussed using the same terminology (e.g. dependency for proteomic relationships and genetic vulnerability). We wished to avoid the term dependency because in our other work, and in the cancer community in general, the term is more widely used to describe genetic vulnerabilities (most notably by the Cancer Dependency Map). Often when one gene is lost, as in the cases here, this creates a dependency on another. We have added some text to the introduction to address known terminology in the field--

We note that there has been a diversity of terms used in the literature to refer to cases where the abundance of one paralog is increased in response to the deletion of another – need-based upregulation, transcriptional reprogramming, active compensation – and here we simply use the term ‘proteomic compensation’ (Kafri et al, 2005; DeLuna et al, 2010; Diss et al, 2014). The opposite effect, where the abundance of one paralog is decreased in response to deletion of another has also been observed (Diss et al, 2017; Dandage & Landry, 2019), and reported using distinct terms, including negative responsiveness, and dependency. We here use the term collateral loss to indicate that a reduction in the protein abundance of one paralog happens in response to the genetic perturbation of another (i.e.

the protein is not itself the target of the genetic event, but nonetheless it displays decreased abundance) (lines 99 to 109).

- To improve readability e should be displayed as $\times 10$ to the power of exponent in the main text and all the figures.

We have amended the text and figures for readability.

- The y-axis on figure 5c-d and 6c-e should be clearly labeled to contain the word "level." Otherwise, it is not clear what is actually shown.

We have amended the axes of these and other boxplots shown in the manuscript to state 'abundance' or 'corrected abundance' to maintain clarity and consistency with the terminology used in the text.

Reviewer 3

- In this manuscript, Venkatesh and colleagues investigated how the copy number loss of paralog genes is tolerated in tumours and human cell lines with a focus on proteomic response. They find that the loss of one paralog can lead to either an increase, active compensation, or a decrease, collateral loss, in the protein abundance of its paralog, with active compensation being more prevalent. Further, the authors show that compensation paralog pairs, unlike collateral loss relationships, are more sequence similar, more central in protein-protein interaction network, more likely to be synthetic lethal, and involved in essential functions. By using an integrative approach that combines genomic, transcriptomic and proteomic data, the authors shed light on underlying regulatory mechanisms of this compensation, and provide evidence that both transcriptional and post-transcriptional mechanisms are under control. By taking advantage of large cancer datasets, the authors characterise proteomic response to paralog loss in tumours and address the gap left by previous studies which have mainly focused on non-human models such as yeast. This work advances our understanding of molecular consequences of paralog loss, with potential clinical relevance for synthetic lethality approaches in

cancer therapy.

Overall, this manuscript is really interesting and makes a great contribution to the field. I have a few specific questions and suggestions that I believe could help improve and strengthen the work.

We thank the reviewer for the careful reading of our manuscript and for the positive feedback.

- 1. I found the analysis in lines 208-219 somewhat difficult to follow, particularly the lines 231-215. From my understanding, the authors tested the overlap between paralog pairs showing reciprocal relationships separately for compensation pairs and collateral pairs, finding that reciprocal relationships are enriched among compensation pairs. If so, the results could be presented more clearly (e.g. indicating the number of pairs have reciprocal relationships out of 4,568 paralogs tested - line 198). I assume the total number should be the sum of all groups in the supplementary figure 3a and 3b? Also, line 214, "overlap among reciprocal compensation pairs..", could be simplified to "... among compensation pairs". Clarification of this section would improve readability.

Since paralog pairs in our analysis are directional, i.e. A1-A2 is distinct from A2-A1, the filters we employ before running our tests often result in a paralog pair only being tested in one direction. For example, A2-A1 would not be tested if A1 is hemizygotously deleted in fewer than 20 samples. As such, to study whether a pair being a hit in one direction made it more likely to be a hit in the other direction, we restricted the analysis to only pairs that have been tested in both directions. We have amended the text for clarity as shown below. We have also replaced the use of 'reciprocal' with 'symmetric', with pairs that are tested in both directions but only a hit in one direction annotated as 'asymmetric' pairs.

In our analysis of the HAP1 data, we found that some pairs were observed as compensatory in a symmetric fashion – A2 loss was associated with an increased abundance of A1, while loss of A1 was associated with increased abundance of A2. With the CPTAC analysis, most paralog pairs were tested in an asymmetric fashion – i.e. we assessed the impact of A2 loss on A1's protein abundance but not A1's loss on A2's protein abundance. This was due to the

filtering steps we applied, which excluded some pairs from being tested in both directions. However, we found that, for the subset of pairs that were tested in both directions, there was a significant enrichment for symmetric compensation pairs (OR = 2.7, $p = 2 \times 10^{-6}$; Appendix Fig. S1a) and for symmetric collateral loss pairs (OR = 1.7, $p = 0.02$; Appendix Fig. S1b). In other words, paralogs that were a hit in one direction were significantly more likely to be a hit in the other direction (lines 231 to 241).

- 1.a. Given that active compensation pairs can have collateral loss relationships when tested in the other direction, it would also be valuable to investigate how the characteristics (sequence similarity, degree centrality etc) differ when compensation pairs are grouped as those with symmetrical and asymmetrical relationships.

This is an interesting idea that we had not considered. We have performed the analysis and identified symmetric compensation pairs to be more likely to be CORUM complex members (including when restricting to essential complexes). The lost paralogs in these pairs are more essential and have higher degree centrality. Symmetric pairs are also more likely to physically interact. We have added plots showing this to Appendix Figure S3 and we discuss it in the text as follows-

Interestingly, symmetric compensation pairs, i.e. pairs exhibiting a compensatory relationship in both directions, were significantly more likely to be in CORUM complexes, both generally and when restricting to essential complexes, compared to asymmetric pairs (OR = 2.3, $p = 1.6 \times 10^{-2}$) (Appendix Figs. S3a, S3b). The lost paralogs in symmetric compensation pairs were also more essential (Cohen's $d = 0.4$, $p = 7.3 \times 10^{-4}$) and had significantly higher degree centrality values (Cohen's $d = 0.5$, $p = 4.6 \times 10^{-3}$) (Appendix Figs. S3c, S3d). Symmetric compensation pairs were also more likely to have a direct physical interaction than asymmetric pairs (OR = 2.8, $p = 0.02$) (Appendix Fig. S3e). These symmetric cases might correspond to paralog pairs that have a stronger functional overlap (lines 367 to 375).

- 2. Based on lines 80-83, "deletion of one gene was associated with reduced protein abundance of its paralog ... due to stoichiometric requirements of protein complexes containing both paralogs.", as well as lines 90-92, "Further analysis suggested that ... dependency are enriched among paralogs that form heteromers ...", and lines 201-206, where the authors provide evidence that they observed similar pattern like dependency - collateral loss- between a paralog pair forms heterodimer. Considering the percentage of post-transcriptionally regulated paralog pairs does not differ much between compensation and collateral loss pairs (Fig 6a) and the hypothesis in lines 402-405 that "...post-transcriptionally regulated pairs might be more likely to involve members of protein complexes ..", it would be interesting to test if this is the case also for collateral pairs (considering the loss of a paralog would leave its paralog orphan and target for degradation)? Alternatively, can the authors justify/clarify why this analysis (number of ubiquitination sites) was performed only for compensation pairs, but not for collateral loss pairs?

We have performed the ubiquitination analysis for collateral loss pairs and found that similarly to compensation pairs, post-transcriptional collateral loss pairs have greater numbers of ubiquitination sites than transcriptional compensation pairs, comparable to CORUM complex members. We have added a new supplementary figure (Appendix Figure S10) showing this. We have discussed this in the text as follows-

We observed a similar pattern for collateral loss— post-transcriptional hits had significantly greater numbers of ubiquitination sites than transcriptional hits (Cliff's Delta = 0.3, $p = 1.5 \times 10^{-6}$), while transcriptional hits had significantly lower numbers of ubiquitination sites than non-hits (Cliff's Delta = -0.2, $p = 1.3 \times 10^{-5}$) (Appendix Fig. S10) (lines 594 to 598).

We note that we do not see any enrichment for protein complex subunits among post-transcriptionally regulated collateral loss pairs, as evident from Appendix Figure S9.

- 1. Line 165-166, "For example, we identified between VPS4B and VPS4B" should be VPS4A and VPS4B.

We have amended this in the text (lines 187 to 188).

- 2. It would be helpful to specify the cancer types used from the CPTAC dataset in the Method section. This addition would improve the transparency and reproducibility.

We have included this in the methods (lines 876 to 880).

- 3. (Important but not-critical) It would be a good addition to include additional discussion on non-hit paralog pairs, particularly since, for CPTAC tumour samples, the majority of paralog pairs (based on line 199: 12% and 10% of paralog pairs are compensation and collateral loss pairs, respectively) fall under non-hit category. Some discussion on possible explanations for this observation would be helpful.

We have added a discussion of this as follows-

We note that the majority of paralog pairs do not exhibit compensatory relationships, suggesting that the loss of many genes can be tolerated without increasing the abundance of a paralog. This may be because the function of the lost gene is completely dispensable (i.e. there is no need to compensate for it), because the compensating paralog is already sufficiently abundant (i.e. compensation can happen without increased abundance), or because compensation happens via an alternative non-paralog mechanism (lines 619 to 632).

References

- Adane B, Alexe G, Seong BKA, Lu D, Hwang EE, Hnisz D, Lareau CA, Ross L, Lin S, Dela Cruz FS, *et al* (2021) STAG2 loss rewires oncogenic and developmental programs to promote metastasis in Ewing sarcoma. *Cancer Cell* 39: 827–844.e10
- Dandage R & Landry CR (2019) Paralog dependency indirectly affects the robustness of human cells. *Mol Syst Biol* 15: e8871
- DeLuna A, Springer M, Kirschner MW & Kishony R (2010) Need-based up-regulation of protein levels in response to deletion of their duplicate genes. *PLoS Biol* 8: e1000347
- Diss G, Ascencio D, DeLuna A & Landry CR (2014) Molecular mechanisms of paralogous compensation and the robustness of cellular networks. *J Exp Zool B Mol Dev Evol* 322: 488–499

- Diss G, Gagnon-Arsenault I, Dion-Coté A-M, Vignaud H, Ascencio DI, Berger CM & Landry CR (2017) Gene duplication can impart fragility, not robustness, in the yeast protein interaction network. *Science* 355: 630–634
- Drew K, Wallingford JB & Marcotte EM (2021) hu.MAP 2.0: integration of over 15,000 proteomic experiments builds a global compendium of human multiprotein assemblies. *Mol Syst Biol* 17: e10016
- Emert BL, Cote CJ, Torre EA, Dardani IP, Jiang CL, Jain N, Shaffer SM & Raj A (2021) Variability within rare cell states enables multiple paths toward drug resistance. *Nat Biotechnol* 39: 865–876
- Garcia-Alonso L, Holland CH, Ibrahim MM, Turei D & Saez-Rodriguez J (2019) Benchmark and integration of resources for the estimation of human transcription factor activities. *Genome Res* 29: 1363–1375
- Gerlinger M, Rowan AJ, Horswell S, Math M, Larkin J, Endesfelder D, Gronroos E, Martinez P, Matthews N, Stewart A, *et al* (2012) Intratumor heterogeneity and branched evolution revealed by multiregion sequencing. *N Engl J Med* 366: 883–892
- Ito T, Young MJ, Li R, Jain S, Wernitznig A, Krill-Burger JM, Lemke CT, Monducci D, Rodriguez DJ, Chang L, *et al* (2021) Paralog knockout profiling identifies DUSP4 and DUSP6 as a digenic dependence in MAPK pathway-driven cancers. *Nat Genet* 53: 1664–1672
- Jacobo Jacobo M, Donnella HJ, Sobti S, Kaushik S, Goga A & Bandyopadhyay S (2024) An inflamed tumor cell subpopulation promotes chemotherapy resistance in triple negative breast cancer. *Sci Rep* 14: 3694
- Jänes J, Müller M, Selvaraj S, Manoel D, Stephenson J, Gonçalves C, Lafita A, Polacco B, Obernier K, Alasoo K, *et al* (2024) Predicted mechanistic impacts of human protein missense variants. *bioRxiv*: 2024.05.29.596373
- Kafri R, Bar-Even A & Pilpel Y (2005) Transcription control reprogramming in genetic backup circuits. *Nat Genet* 37: 295–299
- van der Lelij P, Lieb S, Jude J, Wutz G, Santos CP, Falkenberg K, Schlattl A, Ban J, Schwentner R, Hoffmann T, *et al* (2017) Synthetic lethality between the cohesin subunits STAG1 and STAG2 in diverse cancer contexts. *Elife* 6
- Li Y, Dou Y, Da Veiga Leprevost F, Geffen Y, Calinawan AP, Aguet F, Akiyama Y, Anand S, Birger C, Cao S, *et al* (2023) Proteogenomic data and resources for pan-cancer analysis. *Cancer Cell* 41: 1397–1406
- Meldal BHM, Bye-A-Jee H, Gajdoš L, Hammerová Z, Horácková A, Melicher F, Perfetto L, Pokorný D, Lopez MR, Türková A, *et al* (2019) Complex Portal 2018: extended content and enhanced visualization tools for macromolecular complexes. *Nucleic Acids Res* 47: D550–D558
- Parrish PCR, Thomas JD, Gabel AM, Kamlapurkar S, Bradley RK & Berger AH (2021) Discovery of synthetic lethal and tumor suppressor paralog pairs in the human genome. *Cell Rep* 36: 109597
- Prieto-Vila M, Usuba W, Takahashi R-U, Shimomura I, Sasaki H, Ochiya T & Yamamoto Y (2019) Single-cell analysis reveals a preexisting drug-resistant subpopulation in the luminal breast cancer subtype. *Cancer Res* 79: 4412–4425

Replogle JM, Saunders RA, Pogson AN, Hussmann JA, Lenail A, Guna A, Mascibroda L, Wagner EJ, Adelman K, Lithwick-Yanai G, *et al* (2022) Mapping information-rich genotype-phenotype landscapes with genome-scale Perturb-seq. *Cell* 185: 2559–2575.e28

Smits AH, Ziebell F, Joberty G, Zinn N, Mueller WF, Clauder-Münster S, Eberhard D, Fälth Savitski M, Grandi P, Jakob P, *et al* (2019) Biological plasticity rescues target activity in CRISPR knock outs. *Nat Methods* 16: 1087–1093

Szkarczyk D, Kirsch R, Koutrouli M, Nastou K, Mehryary F, Hachilif R, Gable AL, Fang T, Doncheva NT, Pyysalo S, *et al* (2023) The STRING database in 2023: protein-protein association networks and functional enrichment analyses for any sequenced genome of interest. *Nucleic Acids Res* 51: D638–D646

Thompson NA, Ranzani M, van der Weyden L, Iyer V, Offord V, Droop A, Behan F, Gonçalves E, Speak A, Iorio F, *et al* (2021) Combinatorial CRISPR screen identifies fitness effects of gene paralogues. *Nat Commun* 12: 1302

Tsitsiridis G, Steinkamp R, Giurgiu M, Brauner B, Fobo G, Frishman G, Montrone C & Ruepp A (2023) CORUM: the comprehensive resource of mammalian protein complexes-2022. *Nucleic Acids Res* 51: D539–D545

UniProt Consortium (2025) UniProt: The universal protein knowledgebase in 2025. *Nucleic Acids Res* 53: D609–D617

Zecha J, Satpathy S, Kanashova T, Avanesian SC, Kane MH, Clauser KR, Mertins P, Carr SA & Kuster B (2019) TMT labeling for the masses: A robust and cost-efficient, in-solution labeling approach. *Mol Cell Proteomics* 18: 1468–1478

2nd May 2025

Manuscript Number: MSB-2024-12686R

Title: Proteomic compensation by paralogs preserves protein interaction networks after gene loss in cancer

Author: Anjan Venkatesh

Niall Quinn

Swathi Ramachandra Upadhy

Barbara De Kegel

Alfonso Bolado-Carrancio

Thomas Lefeuvre

Olivier Dennler

Kieran Wynne

Alexander von Kriegsheim

Colm Ryan

Dear Dr Ryan,

Thank you for submitting your revised manuscript to Molecular Systems Biology. We have now received the enclosed report from two Reviewers who agreed to re-assess your work. As you will see below, both reviewers are satisfied with the revisions. Reviewer #3 is currently unavailable. However, given that the points raised by this reviewer were relatively minor and that you appear to have responded to them carefully, we are pleased to inform you that your manuscript can be accepted pending the following minor amendments:

1. Please provide up to five keywords.

2. Remove the "Authors' contribution" section.

3. Data availability:

- The "Code Availability" content should be incorporated into the Data Availability section; please remove the separate "Code Availability" heading.

- Remove referee access codes from the Data Availability section. Please ensure that all referenced datasets will be made publicly available upon acceptance of the manuscript.

4. Supplementary tables:

- Source file names, titles, legends and manuscript callouts all need to be updated to Dataset EV1-EV10 instead of Supplementary/Appendix Tables 1-10.

- Each dataset should be uploaded as individual excel file, with its legend in a separate tab/sheet within the file.

- The section " Appendix table legends" should be removed from the Appendix PDF.

5. Table EV1 in the manuscript should be renamed to "Table 1" and the corresponding callout should be updated accordingly.

5. Please address the following issues related to figure legends:

- Please indicate the statistical test used for data analysis in the legend of figure 2B.

- Please note that the box plots need to be defined in terms of minima, maxima, centre, bounds of box and whiskers, and percentile in the legends of figures 1D-I; 3A, C, D, E, F; 5C, D; 6C, D, E, G; EV1 B-G; EV3 A-D; EV4 A-F.

- Please note that the box plots need to be defined in terms of minima, maxima, percentile in the legends of figures 2D-I.

- Please note that information related to n is missing in the legends of figures 2B, 4B, 5E, EV1 A.

- Please note that the error bars are not defined in the legends of figures 4B, 5E.

4. The section order should be revised as follows: Title page - Abstract & Keywords - Introduction - Results - Discussion -

Methods - Data Availability - Acknowledgements - Disclosure and Competing Interests Statement - References - Figure Legends

- Table(s) -Expanded View Figure Legends.

When you resubmit your manuscript, please download our CHECKLIST (<https://bit.ly/EMBOPressAuthorChecklist>) and include the completed form in your submission. *Please note* that the Author Checklist will be published alongside the paper as part of the transparent process (<https://www.embopress.org/page/journal/17444292/authorguide#transparentprocess>)

Kind regards,
Jingyi

Jingyi Hou, PhD
Senior Editor
Molecular Systems Biology

*** PLEASE NOTE *** As part of the EMBO Press transparent editorial process initiative (see our Editorial at <https://dx.doi.org/10.1038/msb.2010.72> , Molecular Systems Biology will publish online a Review Process File to accompany accepted manuscripts. When preparing your letter of response, please be aware that in the event of acceptance, your cover letter/point-by-point document will be included as part of this File, which will be available to the scientific community. More information about this initiative is available in our Instructions to Authors. If you have any questions about this initiative, please contact the editorial office (msb@embo.org).

Reviewer #1:

I was happy to see that the authors addressed all of my comments.

Reviewer #2:

Authors sufficiently addressed most comments.

All editorial and formatting issues were resolved by the authors.

13th May 2025

Manuscript number: MSB-2024-12686RR

Title: Proteomic compensation by paralogs preserves protein interaction networks after gene loss in cancer

Dear Dr Ryan,

Thank you again for sending us your revised manuscript. We are now satisfied with the modifications made and I am pleased to inform you that your paper has been accepted for publication.

Yours sincerely,
Jingyi

Jingyi Hou, PhD
Senior Editor
Molecular Systems Biology
